# Intra-ripple frequency accommodation in an inhibitory network model for hippocampal ripple oscillations

Natalie Schieferstein[1,2]*, Tilo Schwalger[2,3], Benjamin Lindner[2,4], Richard Kempter[1,2,5]*

**1** Institute for Theoretical Biology, Department of Biology, Humboldt-Universität zu Berlin, Berlin, Germany, **2** Bernstein Center for Computational Neuroscience, Berlin, Germany, **3** Institute for Mathematics, Technische Universität Berlin, Berlin, Germany, **4** Department of Physics, Humboldt-Universität zu Berlin, Berlin, Germany, **5** Einstein Center for Neurosciences, Berlin, Germany

* natalie.schieferstein@freenet.de (NS); r.kempter@biologie.hu-berlin.de (RK)

**Data Availability Statement:** All code written in support of this publication is publicly available at https://github.com/NatalieSchieferstein/interneuron_ripples_with_ifa.git (https://doi.org/10.

## Abstract

Hippocampal ripple oscillations have been implicated in important cognitive functions such as memory consolidation and planning. Multiple computational models have been proposed to explain the emergence of ripple oscillations, relying either on excitation or inhibition as the main pacemaker. Nevertheless, the generating mechanism of ripples remains unclear. An interesting dynamical feature of experimentally measured ripples, which may advance model selection, is intra-ripple frequency accommodation (IFA): a decay of the instantaneous ripple frequency over the course of a ripple event. So far, only a feedback-based inhibition-first model, which relies on delayed inhibitory synaptic coupling, has been shown to reproduce IFA. Here we use an analytical mean-field approach and numerical simulations of a leaky integrate-and-fire spiking network to explain the mechanism of IFA. We develop a drift-based approximation for the oscillation dynamics of the population rate and the mean membrane potential of interneurons under strong excitatory drive and strong inhibitory coupling. For IFA, the speed at which the excitatory drive changes is critical. We demonstrate that IFA arises due to a speed-dependent hysteresis effect in the dynamics of the mean membrane potential, when the interneurons receive transient, sharp wave-associated excitation. We thus predict that the IFA asymmetry vanishes in the limit of slowly changing drive, but is otherwise a robust feature of the feedback-based inhibition-first ripple model.

## Author summary

The hippocampus plays a central role in the acquisition and consolidation of explicit memories. During sleep or rest, hippocampal neurons replay recently acquired memories, while the neuronal network exhibits high-frequency oscillations, so-called ripples. To study the potential function of ripples, their generation mechanism needs to be clarified. Here we analyze a model network of inhibitory interneurons as a potential ripple pacemaker. We derive an analytical approximation of the ripple dynamics that explains why

5281/zenodo.10602552). An exemplary simulation result file is available from https://zenodo.org/doi/10.5281/zenodo.10018496.

**Funding:** This work was supported by the German Research Foundation (Deutsche Forschungsgemeinschaft [DFG], SFB 1315, project-ID 327654276 to BL and RK, and GRK 1589/2 to RK and NS; https://www.dfg.de/de). The funders had no role in study design, data collection and analysis, decision to publish, or preparation of the manuscript.

**Competing interests:** The authors have declared that no competing interests exist.

the ripple frequency tends to decay over the course of an event (intra-ripple frequency accommodation, IFA). Studying dynamical phenomena such as IFA advances model selection and enables an understanding of how rhythmic brain activity contributes to cognitive functions.

## Introduction

Sharp wave-ripples (SPW-R) are a prominent rhythmic signature of neuronal activity in the hippocampus. Hippocampal SPW-Rs are embedded in a larger system of thalamo-cortical rhythms [1, 2] and have been associated with the replay of behaviorally relevant neuronal activity [3–5]. SPW-Rs may thus be important for cognitive functions such as memory consolidation [6–11] and planning [4, 12–14]. To probe their potential functional role, we first need to understand the generation mechanism of SPW-Rs.

SPW-Rs are brief (50–100 ms) periods of elevated, highly synchronized neuronal activity, which can be measured in the local field potential (LFP) of the hippocampus *in vivo* [15–17], as well as *in vitro* [18–22]. In CA1 the LFP sharp wave (SPW) is most prominent in layer *stratum radiatum* and is thought to reflect a current sink due to elevated excitatory synaptic transmission from the CA3 Schaffer collaterals. The LFP ripple oscillation (150–250 Hz *in vivo*, [16]; 210 ± 16 Hz *in vitro*, [19]) is strongest in *stratum pyramidale* of CA1 and is thought to reflect inhibitory synaptic currents, and potentially rhythmic excitatory action potentials [23, 24]. It is believed that the local CA1 network can generate ripples in isolation [19, 25, 26].

Various computational models have been put forward to explain ripple generation, relying either on excitation or inhibition as the main pacemaker (see Discussion for *A note on mixed models*). Excitation-first models assume that SPW and ripple oscillations are generated jointly by the sparsely connected pyramidal cell network, either via axonal gap junctions and antidromic spike propagation [27, 28] or via supralinear dendritic integration [29].

Inhibition-first models posit that the interneuron network (*e.g.* in CA1) produces ripples in response to transient excitatory input due to a SPW event, which is generated by a separate mechanism upstream (*e.g.* in CA3) [30–32]. Inhibition-first models can be further subdivided into two classes, which we will call here *feedback-based* and *perturbation-based*, hinting at the mechanism by which ripples emerge: In the feedback-based models [16, 33–40] the recurrent synaptic coupling between interneurons is strong such that for sufficiently strong excitatory drive the network undergoes a bifurcation and can, in theory, produce *sustained* ripple oscillations as long as the drive remains strong enough. The perturbation-based model [41], on the other hand, assumes weak coupling and cannot produce sustained oscillations. Here ripples emerge as a transient ringing effect in response to a *perturbation* in the external drive.

Experiments remain inconclusive as to which of the proposed mechanisms is the most plausible for ripple generation [17, 20, 42]. Previous analyses have focused on the average frequency and duration of ripple events as well as their dependency on pharmacological or optogenetic manipulation [17, 21, 43]. We propose that taking into account the *transient* dynamical features of spontaneous ripples can advance model selection. The instantaneous ripple frequency typically decays over the course of a ripple event. This *intra-ripple frequency accommodation* (IFA) has been observed *in vivo* and *in vitro* and across different animals, brain states, and measurements [9, 25, 37, 43–46]. It is therefore an interesting question which of the proposed ripple models can account for IFA, and under which assumptions.

So far, only the feedback-based inhibition-first ripple model has been shown to reproduce IFA in exemplary numerical simulations [37]. Here we use a theoretical mean-field approach

and extensive numerical simulations to explain the mechanism of IFA, show that IFA is robust with respect to parameter variation, and predict that IFA depends on the time course of the external, SPW-associated drive. These insights about IFA can be used to distinguish between the feedback-based and the perturbation-based inhibition-first ripple mechanisms.

## Results

The feedback-based inhibitory ripple model assumes that a network of synaptically coupled CA1 interneurons, such as parvalbumin-expressing (PV[+]) basket cells, acts as a delayed negative feedback loop and thus creates fast ripple oscillations when stimulated with excitatory, sharp wave-associated drive from CA3. This idea was first expressed by [16, 33], and has been formally studied in computational models of varying complexity [34–39]. Simulations of a biophysically detailed version of this model by [37] revealed that it can reproduce intra-ripple frequency accommodation (IFA) in response to time-dependent, sharp wave-like drive.

To explain the mechanism of IFA analytically, we first demonstrate that IFA is preserved in a spiking network model of reduced complexity (the network used in [34] with all-to-all coupling). We then turn to a mean-field approximation of the network dynamics, which enables us to study the ripple dynamics and IFA as a function of the time course of the excitatory sharp wave-like drive.

### Ripples and IFA in a spiking neural network model

We model the CA1 interneuron network as a homogeneous network of $N$ leaky integrate-and-fire (LIF) neurons with membrane time constant $\tau_m$, capacitance $C$, and resting potential $E_{\text{leak}}$. The membrane potential $\nu_i$ of a unit $i$ is given by the following stochastic differential equation:

$$\tau_m \dot{\nu}_i = -\nu_i + E_{\text{leak}} + \frac{\tau_m}{C} I_{\text{ext}}(t) - \tau_m \frac{J}{N} \sum_{j=1}^{N} \sum_k \delta\left(t - t_j^k - \Delta\right) + \sqrt{2\tau_m} \sigma_V \xi_i(t) \tag{1}$$

where the derivative with respect to time $t$ is abbreviated by a dot: $\dot{\nu}_i = \frac{d\nu_i}{dt}$. Whenever the membrane potential crosses a spike threshold $V_{\text{thr}}$, a spike is emitted and the membrane potential is reset instantaneously to a reset potential $V_{\text{reset}}$. For simplicity there is no absolute refractory period. All interneurons receive the same external, excitatory drive $I_{\text{ext}}(t)$ and an independent Gaussian white noise input $\xi_i(t)$, with zero mean $\langle \xi_i(t) \rangle = 0$ and unit noise intensity $\langle \xi_i(t)\xi_j(t') \rangle = \delta_{ij}\delta(t - t')$, scaled by the noise strength parameter $\sigma_V$. The network is fully connected via inhibitory pulse coupling of strength $J$ and with a synaptic delay $\Delta$ (see Methods for details and default values of parameters).

The empirical population activity $r_N(t)$ in a small time interval $[t, t + \Delta t)$ is defined as the number of spikes $n_{\text{spk}}(t, t + \Delta t)$ emitted by the population (Methods, Eq (15)), divided by the size of the population and the time step $\Delta t$:

$$r_N(t) := \frac{n_{\text{spk}}(t, t + \Delta t)}{N\Delta t}. \tag{2}$$

For plotting purposes, $r_N$ is smoothed with a narrow Gaussian kernel (Methods, Eq (16)). In the following, we will illustrate that the key network dynamics in response to constant drive $I_{\text{ext}}$ and time-dependent, sharp wave-like drive $I_{\text{ext}}(t)$, as presented by [37], are preserved in this model, *i.e.*, there are fast oscillations in the ripple range and there is IFA.

**Dynamics for constant drive.** The network dynamics for constant drive $I_{\text{ext}}$ are illustrated in Fig 1. At low $I_{\text{ext}}$, the network is in a steady-state with units firing asynchronously and irregularly at an overall low rate $f_{\text{unit}}$ (Fig 1A, left). As $I_{\text{ext}}$ increases, the network activity begins to

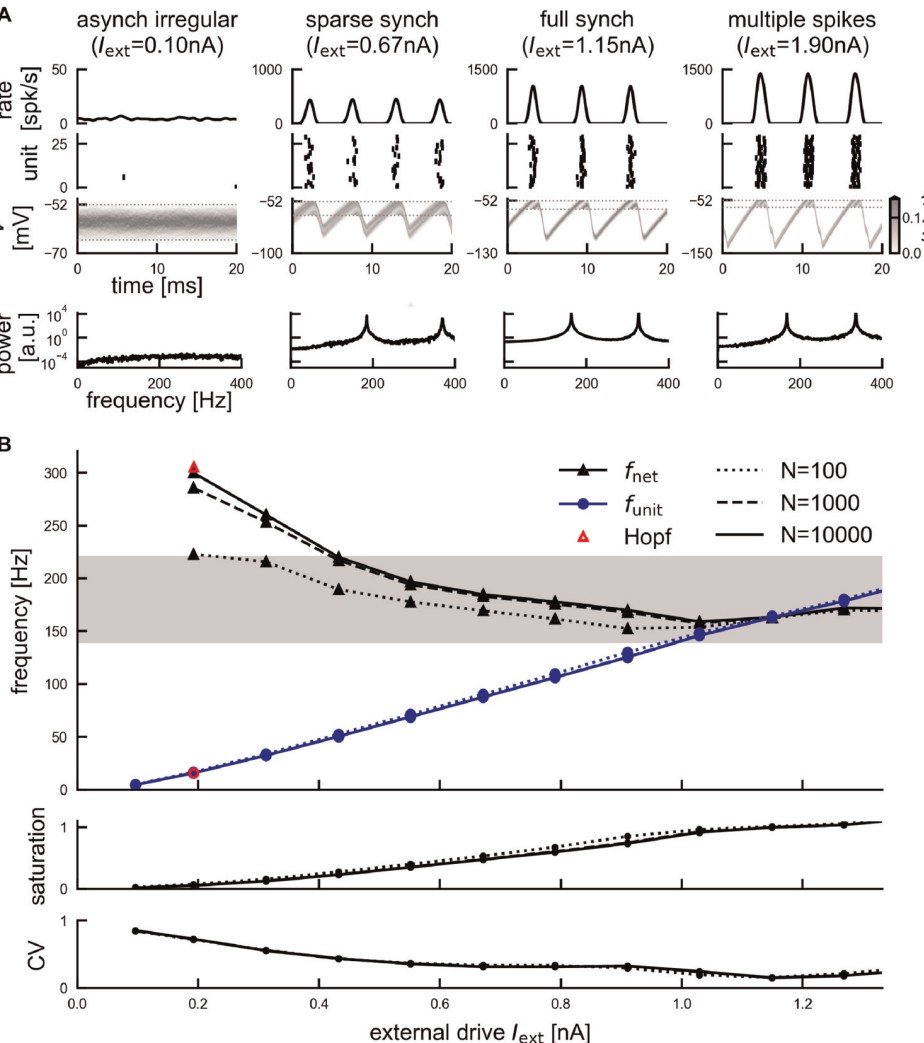

**Fig 1. Constant-drive dynamics of the spiking network.** (A) Four dynamical regimes depending on the external drive: asynchronous irregular state, sparse synchrony, full synchrony, multiple spikes (left to right, $N = 10,000$). Top: population rate $r_N$, middle: raster plot showing spikes of 30 example units, and histogram of membrane potentials $v$ (normalized as density, threshold and reset marked by horizontal dashed lines), bottom: power spectral density of the population rate. (B) Top: network frequency $f_{net}$ (black) and mean unit firing rate $f_{unit}$ (blue) for a range of constant external drives $I_{ext}$. Grey band marks approximate ripple frequency range (140–220 Hz). Red markers indicate the critical input level $I_{ext}^{crit}$ (Hopf bifurcation) and the associated network and unit frequency, as resulting from linear stability analysis, see section *Linear Stability Analysis* in Methods, Eq (70), and [34]. Linestyle indicates network size ($N \in [10^2, 10^3, 10^4]$). Middle: saturation $s = f_{unit}/f_{net}$. Bottom: coefficient of variation of interspike intervals, averaged across units.

exhibit coherent oscillations (Fig 1A, middle). In the following, we will refer to the dominant frequency of this population oscillation as the network frequency $f_{net}$ (Methods, black triangle markers in Fig 1B). For a biologically reasonable set of parameters (Methods) the network frequency lies within the ripple range for a large range of external drives (Fig 1B, gray band). For $0.3\,\text{nA} \lesssim I_{ext} \lesssim 0.9\,\text{nA}$, the unit activity underlying the network ripple oscillation is sparse and irregular, *i.e.* the average unit firing rate $f_{unit}$ is lower than the frequency $f_{net}$ of the network oscillation (*saturation* $s = f_{unit}/f_{net} < 1$) and the coefficient of variation (CV) of interspike intervals is around 0.5 (Fig 1B, bottom panels). We hence refer to this state as *sparse synchrony* [37, 47].

In the mean-field limit $N \to \infty$, the transition to this oscillatory state is sharp (a supercritical Hopf bifurcation) and the transition point can be determined by a linear stability analysis of the stationary state [34]. In a simulated spiking network of finite size, the transition to the oscillatory state is not perfectly sharp because of inevitable fluctuations. Nevertheless, at the bifurcation point, both the network frequency and the unit firing rate are well predicted by the mean-field theory if the network is large enough (Fig 1B, red markers: $I_{ext}^{crit} = 0.19$ nA with $f_{net}$ = 305 Hz, $f_{unit}$ = 16 Hz, $s = 0.05$).

For some sufficiently strong drive, the network reaches a state of full synchrony ($s \approx 1$), with units firing regularly and at the same average frequency as the population rhythm (Fig 1A, "full synch"). If the drive increases beyond this level, units spike several times per cycle ($s > 1$, Fig 1A, "multiple spikes")), which increases the CV of interspike intervals. Since the firing rates in that regime are too high to be biologically plausible, we focus in the following on the dynamical regime between the onset of oscillations (bifurcation point) and the point of full synchrony (Fig 1).

We note that in this reduced model (and also in [34]) the network frequency is a decreasing function of the external drive (Fig 1B). It is generally not straight-forward to infer how the network frequency changes with the external drive $I_{ext}$: On the one hand, individual neurons reach the spiking threshold faster when the drive is stronger. On the other hand, the resulting increase in spiking activity leads to a stronger delayed feedback inhibition that pushes all membrane potentials further away from the threshold. The network frequency thus results from a self-consistency condition. In fact, the shape of the network frequency curve can be different (increasing or even non-monotonic) depending on the details of the model network architecture (see also Discussion and Fig A in S2 Appendix). The mechanism of IFA that we demonstrate in the following is independent of the shape of the network frequencies under constant drive.

**Dynamics for time-dependent drive.** During a sharp wave (SPW) event, which is thought here to be generated in CA3 (cf. models introduced by [30–32]), pyramidal cells in CA3 transiently increase their firing rates [43, 48]. Here we model the resulting feedforward input $I_{ext}(t)$ to the CA1 interneuron network in a simplified form: as a symmetric, piecewise linear double ramp (Fig 2A, bottom; see also Eq (17) in Methods, and Discussion). This SPW-like drive elicits a transient ripple event in the network (Fig 2A). We measure the instantaneous frequency of the population activity, either by taking a windowed Fourier transform (wavelet spectrogram) and finding the peak in power in each time step, or by taking the inverse of the distances between consecutive peaks in the population activity providing frequencies for a discrete set of time points (Fig 2A, top, solid line vs white dots).

There is no established definition of IFA. The central aspect of IFA is the decrease (*accommodation*) of the instantaneous frequency. Over the course of an experimentally measured ripple event (20–100 ms) the instantaneous ripple frequency typically decreases by about 10–60 Hz (rough visual estimate of peak-to-trough distances seen in [9, 25, 37, 43–46]), corresponding to linear slopes of about −0.1 to −3 Hz/ms. We observe such a decreasing trend of the instantaneous frequency in the model even though the external stimulus is perfectly symmetric (Fig 2A, top, see also [37]), *i.e.* the network exhibits intra-ripple frequency accommodation (IFA). We quantify IFA by linear regression over the discrete instantaneous frequency estimates (Fig 2A, white dots). This quantification of IFA is applied to results from many simulations with different noise realizations and the same SPW-like drive (Fig 2B; Methods, Eq (18)). A negative regression slope (here $\chi_{IFA} = -3.01$ Hz/ms) indicates IFA.

In the model, the decrease in instantaneous frequency is not strictly linear over the entire event. In the central portion of the simulated ripple event, where power in the ripple band is large, the ripple frequency decreases monotonically. At the end of the event, however, we

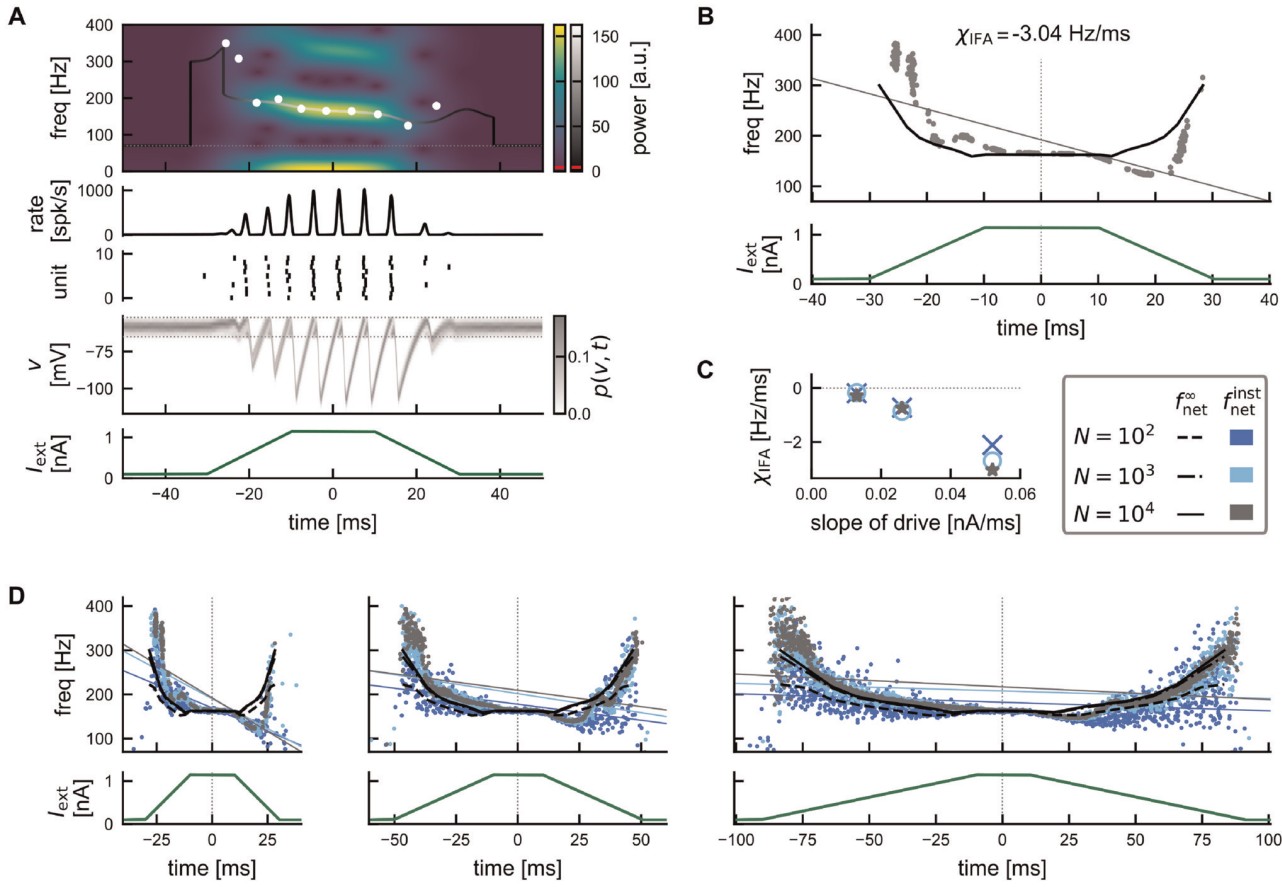

**Fig 2. Transient dynamics of the spiking network and IFA.** (A) Example simulation showing a transient ripple with IFA. From bottom to top: SPW-like drive (Eq (17)); histogram of membrane potentials (normalized), horizontal dotted lines: threshold and reset potential; raster plot showing spike times of 10 example units; population rate exhibiting transient ripple oscillation; wavelet spectrogram indicating instantaneous power (blue-yellow colorbar) for a frequency range of 0–400 Hz. Solid curve: continuous estimate of instantaneous frequency based on wavelet spectrogram with gray scale indicating maximal instantaneous power. Dotted line: cutoff frequency for peak detection $f_{min}$ = 70Hz. Red lines in scalebars: power threshold (see Methods). White dots: discrete estimate of instantaneous frequency based on peak-to-peak distance in population rate. Network size $N$ = 10, 000. (B) Quantification of IFA. Top: Grey dots: discrete instantaneous frequency estimates from 50 repetitions of the simulation shown in (A) with different noise realizations. Grey line: linear regression line with negative slope ($\chi_{IFA}$ = −3.04 Hz/ms) indicating IFA (see Methods, Eq (18)). Black line: asymptotic frequencies (cf. Fig 1B, top). Bottom: The same SPW-like drive was applied in all 50 simulations. Network size $N$ = 10, 000. (C) Dependency of IFA slope $\chi_{IFA}$ on the slope of the external drive for different network sizes (color coded). (D) Instantaneous (dots) vs asymptotic (black lines) network frequencies (top) for piecewise linear drives (bottom) of decreasing slopes (left to right). Color/linestyle indicates network size $N$. Thin, colored linear regression lines illustrate decreasing strength of IFA for shallower drive (regression slopes summarized in C). Asymptotic network frequencies are derived via interpolation of the constant-drive results shown in Fig 1B, top. The asymptotic frequencies $f_{net}^{\infty}$ for $N$ = $10^3$ and $N$ = $10^4$ are nearly identical (dash-dotted and solid lines). Note that the drive is identical in panels A, B, and in the left panel of D.

observe a small *in*crease of the frequency, albeit with low power (the underlying peaks in the population rate are small) (Fig 2A and 2B top). Similar non-monotonic trajectories of the ripple frequency have been observed in experimentally measured ripples and still been classified as exhibiting IFA [44].

The transient dynamics of the instantaneous frequency in this model can be understood best by comparing it to the asymptotic frequency that the network would settle into if the drive remained constant at the instantaneous value of the drive (Fig 2B, grey dots vs black line). Naturally, this asymptotic frequency follows the same symmetry as the external drive and thus provides a useful reference frame. For our double ramp drive (Methods, Eq (17)), we observe that during the plateau phase (*i.e.* constant drive) the instantaneous frequency quickly

approaches the asymptotic frequency. However, during the rising phase of the drive the instantaneous frequency tends to be higher than the asymptotic reference. During the falling phase it is lower, thus creating the overall IFA asymmetry.

Varying the slope of the external double ramp drive we see that the IFA asymmetry is speed-dependent (Fig 2C and 2D): If the external drive changes more slowly (smaller slope), the network frequency response becomes more symmetric; for very small slopes the instantaneous frequencies approach the symmetric, asymptotic reference frequencies, and IFA vanishes.

In what follows, we aim at understanding the mechanism behind the observations illustrated for constant drive in Fig 1 and time-dependent drive in Fig 2. Further simulations indicate that the network dynamics varies only little when the network size is varied by two orders of magnitude (Figs 1B, 2C and 2D). We hence hypothesize that IFA is preserved in the mean-field limit of an infinitely large network, and will use a mean-field approach to explain the generating mechanism of IFA.

## Gaussian-drift approximation of ripple dynamics in the mean-field limit

To facilitate notation in the mathematical analysis, we rescale all voltages to units of the distance between threshold and rest such that the new spiking threshold is at $V_T = 1$ and the resting potential is at $E_L = 0$. The single unit stochastic differential equation (previously Eq (1)) then reads

$$\tau_m \dot{V}_i = -V_i + I_E(t) - \frac{K\tau_m}{N}\sum_{j=1}^{N}\sum_{k} \delta(t - t_j^k - \Delta) + \sqrt{2D\tau_m}\xi_i(t)\,, \tag{3}$$

with rescaled external excitatory current $I_E$, inhibitory synaptic strength $K$, and noise intensity $D$; see also section *Dimensionless equations* in the Methods, Eqs (20)–(22); there we provide details on the rescaling and values of default parameters.

In the mean-field limit $N \to \infty$, the dynamics of the density of membrane potentials $p(V, t)$ is described by the Fokker-Planck equation (FPE)

$$\tau_m \partial_t p(V, t) = -\partial_V\left[(I(t) - V)p(V, t)\right] + D\partial_V^2 p(V, t) \tag{4}$$

$$I(t) = I_E(t) - I_I(t) = I_E(t) - K\tau_m r(t - \Delta) \tag{5}$$

$$r(t) = -\frac{D}{\tau_m}\partial_V p(V_T^-, t) \tag{6}$$

(see Methods, Eq (23) for details). The population rate $r(t)$ defined in Eq (6) represents the mean-field limit of the population activity $r_N(t)$ in Eq (2) of the spiking neural network. Compared to the classical application of the FPE [49, 50] there are two essential differences: First, the FPE is nonlocal because of the resetting rule. This rule imposes an absorbing boundary condition at the threshold (Eq (23e)) and a source of probability at the reset point; the latter source can be imposed by a jump condition for the derivative of the density at the reset point (Eq (23g)) that matches the derivative at the threshold point, see *e.g.* [51–55]. Secondly, the FPE (4) is nonlinear because the current $I(t)$ depends on the probability density $p(V, t)$ through the population rate $r(t)$ in Eq (6).

Stable stationary solutions of this nonlocal, nonlinear FPE correspond to asynchronous irregular spiking [34, 51]. It has been shown that an oscillatory (*i.e.* periodic) solution $r(t)$ emerges via a supercritical Hopf bifurcation when the external drive $I_E$ exceeds a critical value

[34, 51]. This network oscillation well reproduces the coherent stochastic oscillation of the population activity $r_N(t)$ in the finite-size spiking neural network (Fig 1). The network frequency at the onset of oscillations (*i.e.* at the Hopf bifurcation, where the stationary solution looses stability) is well predicted analytically by a linear stability analysis [34] (see also Methods, Eq (70) and Fig 1B, red markers). However, this analytical prediction quickly breaks down further away from the bifurcation where the oscillation dynamics becomes strongly nonlinear (see power spectral densities with higher harmonics in Fig 1A). In this regime neither an exact periodic solution of the FPE nor an approximation of its frequency is known. We will thus introduce a simplified approach that allows us to approximate the network frequencies at strong drive beyond the bifurcation.

Our approach can be motivated by two observations from the spiking network simulations: (a) In the relevant regime between sparse and full synchrony, units spike at most *once* per cycle of the population rhythm ($f_{unit} \leq f_{net}$, Fig 1B). This property will allow us to approximate the time course of a population spike in $r_N(t)$ using a first-passage-time ansatz, neglecting the reset mechanism. (b) In between population spikes, the population rate $r_N(t)$ is close to zero, and the strong inhibitory feedback pushes the bulk of the membrane potential distribution significantly below threshold (Fig 1A). In those periods the absorbing boundary condition at threshold does not have any significant impact on the dynamics of $p(V, t)$. Observation b) is illustrated in more detail in Fig 3A, which shows the spiking network simulations from Fig 1 ($I_{ext} = 0.55$ nA) with rescaled voltages.

These two observations motivate a considerable simplification of the mean-field dynamics: Without the boundary conditions at threshold and reset, the FPE (4) can be solved analytically. In the long-time limit its solution becomes a simple Gaussian—independent of the initial condition (Methods, Eq (24)). We thus approximate the density of membrane potentials as a Gaussian

$$p(V, t) \approx \frac{1}{\sqrt{2\pi D}} \exp\left[-\frac{(V - \mu(t))^2}{2D}\right] \tag{7}$$

(see Fig 3B), and the only time-dependent quantity is the mean membrane potential $\mu(t)$, which evolves according to

$$\dot{\mu}(t) = \frac{1}{\tau_m}(I(t) - \mu(t)) \overset{(5)}{=} \frac{1}{\tau_m}(I_E(t) - K\tau_m r(t - \Delta) - \mu(t)) \tag{8a}$$

(see Fig 3Bi and 3Bii, black lines, cf. Methods Eq (25)). Because in the considered regime spikes are mainly driven by the mean input rather than membrane potential fluctuations, the population rate can be well approximated by the drift part of the probability current across the threshold, while diffusion-mediated spiking is ignored (Methods, Eqs (28) and (29); see also [56–59]):

$$r(t) = [\dot{\mu}(t)]_+ p(V_T, t) = \frac{[\dot{\mu}(t)]_+}{\sqrt{2\pi D}} \exp\left[-\frac{(V_T - \mu(t))^2}{2D}\right] . \tag{8b}$$

Thus, in our approximate dynamics without reset, the population rate $r(t)$ is given by the membrane potential density at threshold, scaled by the speed $\dot{\mu}$ at which the mean membrane potential approaches the threshold. Whenever the mean membrane potential is *de*creasing ($\dot{\mu} < 0$), the drift current at the threshold is negative, and hence, the rate is clipped to 0 ($[x]_+ := (x + |x|)/2$, see Fig 3Bi and 3Bii, top).

Because the single-neuron dynamics of integrate-and-fire neurons includes a reset mechanism after the spike-threshold has been reached, we add a phenomenological account for such

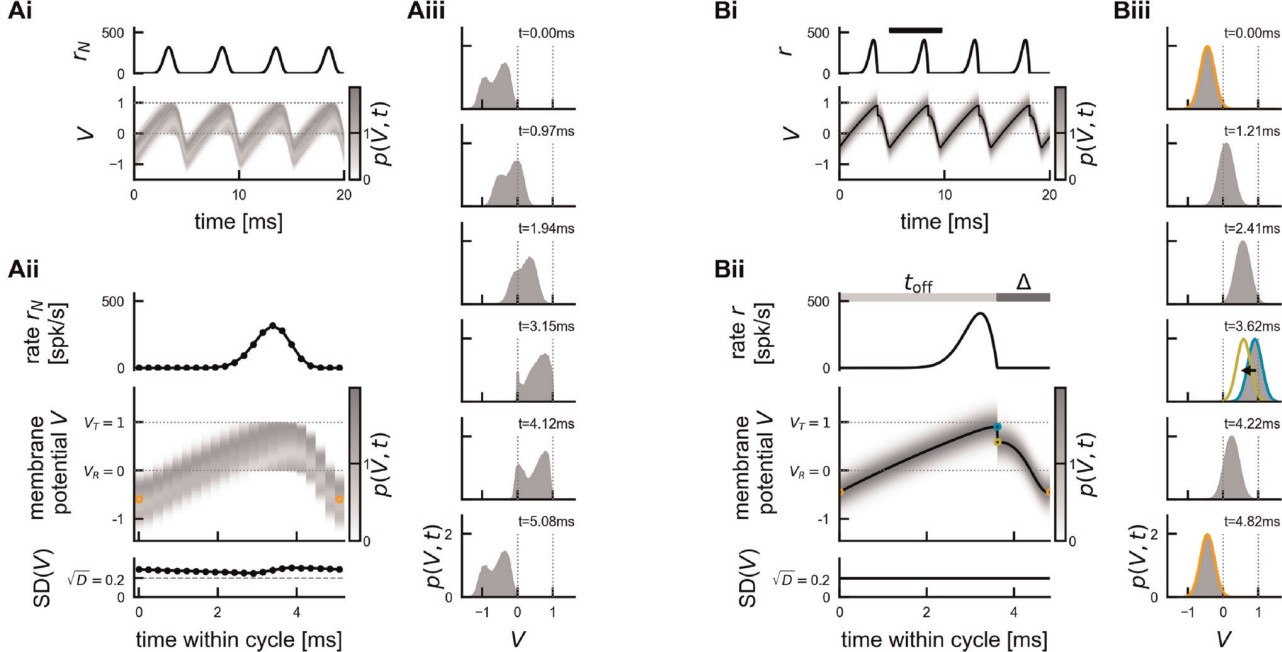

**Fig 3. Illustration of the Gaussian-drift approximation.** Comparison of oscillation dynamics in the spiking network simulation (A) and the Gaussian-drift approximation (B) at constant drive ($I_E = 4.24$). (Ai) Spiking network simulation. Empirical population rate $r_N$ (top), and density of membrane potentials (bottom, dimensionless voltage), exhibiting coherent stochastic oscillations with (weak) finite size fluctuations. (Aii) The average cycle of the oscillation dynamics in Ai (computed for 21 bins of 0.24 ms each). Top: population rate; middle: density of membrane potentials, orange marker: local minimum of mean membrane potential; bottom: standard deviation of membrane potential distribution, dashed line: theoretical asymptote $\sqrt{D}$ in the absence of boundary conditions (Eq (26)). (Aiii) Snapshots of the membrane potential density over the course of the average cycle shown in (Aii). (Bi) Gaussian-drift approximation (numerical integration of DDE (8) with reset condition). Top: population rate, bottom: mean membrane potential $\mu(t)$ (black line) with Gaussian density of membrane potentials $p(V, t)$ painted on in the background for better comparison with Ai. (Bii) Zoom into one oscillation cycle (black bar in Bi). Top: population rate $r$, middle: density of membrane potentials $p(V, t)$ with mean $\mu(t)$, bottom: constant standard deviation $\sqrt{D}$. The mean membrane potential $\mu(t)$ (black line) starts in each cycle at $\mu_{\min}$ (orange) and rises up until $\mu_{\max}$ (cyan) at time $t_{\text{off}}$, at which point the population spike ends. In a phenomenological account for the single unit reset, $\mu$ is reset instantaneously to $\mu_{\text{reset}}$ (yellow). From there $\mu$ declines back towards $\mu_{\min}$. (Biii) Snapshots of the membrane potential density $p(V, t)$ over the course of one cycle (Bii). Colors mark $t = 0 \sim T$ (orange), and $t = t_{\text{off}}$ (cyan/yellow). Dotted lines in all voltage panels mark spike threshold $V_T = 1$ and reset potential $V_R = 0$. Note that in the theoretical approximation the spike threshold $V_T = 1$ is no longer an absorbing boundary.

a reset to our population-level description: At the end of a population spike ($r \sim [\dot{\mu}]_+ = 0, \ddot{\mu} < 0$) the mean membrane potential is pushed down ("reset") by an amount that is proportional to the fraction of neurons that have participated in the population spike (Fig 3Biii, $t = 3.62$ ms; see Methods Eq (49) for details).

The two coupled Eqs (8a) and (8b) are equivalent to a single delay differential equation (DDE; Methods, Eq (31)). This DDE (including the reset condition) governs the dynamics of the mean of the Gaussian membrane potential distribution, and the resulting drift-based population rate. In the following, we will therefore refer to the DDE dynamics (with the phenomenological reset condition) as the *Gaussian-drift approximation*.

The main differences between the spiking network model (or the exact mean-field dynamics given by the FPE (4)) and the Gaussian-drift approximation are illustrated in Fig 3 for the case of constant drive. In the spiking network model, the membrane potential density $p(V, t)$ is not strictly Gaussian (Fig 3Aiii). Because of the fire-and-reset mechanism, $p(V, t)$ changes in shape during the oscillation cycle, becoming at times even bimodal (Fig 3Aiii vs Fig 3Biii). Still, we see that our Gaussian assumption (b) is justified: Whenever the membrane potential

density is subthreshold in between population spikes, it becomes more Gaussian, and its standard deviation approaches $\sqrt{D}$ (Fig 3Aii, bottom, and Fig 3Aiii, first 3 snapshots).

The advantage of our Gaussian-drift approximation is that it reduces the FPE with complex boundary conditions to a simpler DDE with oscillatory solutions that can be studied analytically. In the following, we will first consider the dynamics for constant drive, then extend the Gaussian-drift approximation to understand the transient response to time-dependent drive, and hence explain the emergence of IFA.

**Analysis of oscillation dynamics for constant drive.** A $T$-periodic solution of the Gaussian-drift model, Eq (8), must have a mean membrane potential $\mu(t) = \mu(t + T)$ that oscillates between two local extrema $\mu_{\min}$ and $\mu_{\max}$ (Fig 3Bi). Whenever the mean membrane potential *increases, a positive population rate $r$ is produced; when the mean membrane potential *de*creases the population rate is clipped to 0 (Eq (8b)). Let us consider a single cycle of this oscillatory dynamics (Fig 3Bii): The moment when $\mu$ reaches its local maximum $\mu_{\max}$ (Fig 3Bii, cyan circle) is of special importance as it marks the end of the population spike. We will refer to this time as $t_{\mathrm{off}}$:

$$\mu(t_{\mathrm{off}}) = \mu_{\max}, \quad r(t_{\mathrm{off}}) = 0 .$$

Since the inhibitory feedback $I_{\mathrm{I}}(t) = -K\tau_m r(t - \Delta)$ is proportional to the population rate with a delay of $\Delta$ (Eq (8a)), it follows that this feedback ceases exactly $\Delta$ after the end of the population spike, *i.e.* $I_{\mathrm{I}}(t_{\mathit{off}} + \Delta) = 0$. It is convenient to define this time as the end of a cycle, *i.e.* the beginning of the next one. The mean membrane potential at this time is close to its local minimum (see Methods, Step 2) and will be denoted as $\mu_{\min} := \mu(t_{\mathrm{off}} + \Delta)$. The period $T$ can then be split into the *upstroke* time $t_{\mathrm{off}}$ needed for the mean membrane potential to rise from $\mu(t = 0) = \mu_{\min}$ towards $\mu(t_{\mathrm{off}}) = \mu_{\max}$, and the *downstroke* time $\Delta$ during which the mean membrane potential is pushed back down to $\mu(T) = \mu_{\min}$ due to the delayed inhibitory feedback (Fig 3Bii). In Methods we derive, through a series of heuristic approximations, analytical expressions for the local extrema $\mu_{\max}$ (Eq (38)) and $\mu_{\min}$ (Eq (50)) of the mean membrane potential oscillation as a function of the external drive $I_E$. Using these expressions, we obtain an analytical formula for $t_{\mathrm{off}}$ (Eq (47)) and hence for the network frequency:

$$f_{\mathrm{net}} = T^{-1} = (t_{\mathrm{off}} + \Delta)^{-1} . \tag{9}$$

Apart from the network frequency, the Gaussian-drift approximation also allows an intuitive understanding of the mean unit firing rate. When the population spike ends at time $t_{\mathrm{off}}$, the suprathreshold portion of the Gaussian density corresponds to the fraction of units that have spiked in the given cycle (the *saturation s*, Methods, Eq (48)). The mean unit firing rate can thus be inferred as:

$$f_{\mathrm{unit}} = sf_{\mathrm{net}} . \tag{10}$$

In the following, we compare our analytical Gaussian-drift approximations for the network frequency and mean membrane potential dynamics to the spiking network simulations for a range of external drives $I_{\mathrm{E}}$.

**Performance evaluation of the Gaussian-drift approximation.** Our theory captures the dependence of the network frequency $f_{\mathrm{net}}$ and the mean unit firing rate $f_{\mathrm{unit}}$ on the external drive $I_{\mathrm{E}}$, including the transition from sparse to full synchrony for increasing external drive (Fig 4, top). Our analytically derived expression for the saturation $s$ predicts this transition (Fig 4, middle), since $s$ is a monotonically increasing function of $\mu_{\max}$ (Eq (48)), which monotonically increases as a function of the external drive $I_{\mathrm{E}}$ (Eq (38), Fig 4, bottom). We can also estimate analytically the point of full synchrony, *i.e.* the amount of external drive $I_{\mathrm{E}}^{\mathrm{full}}$ that is

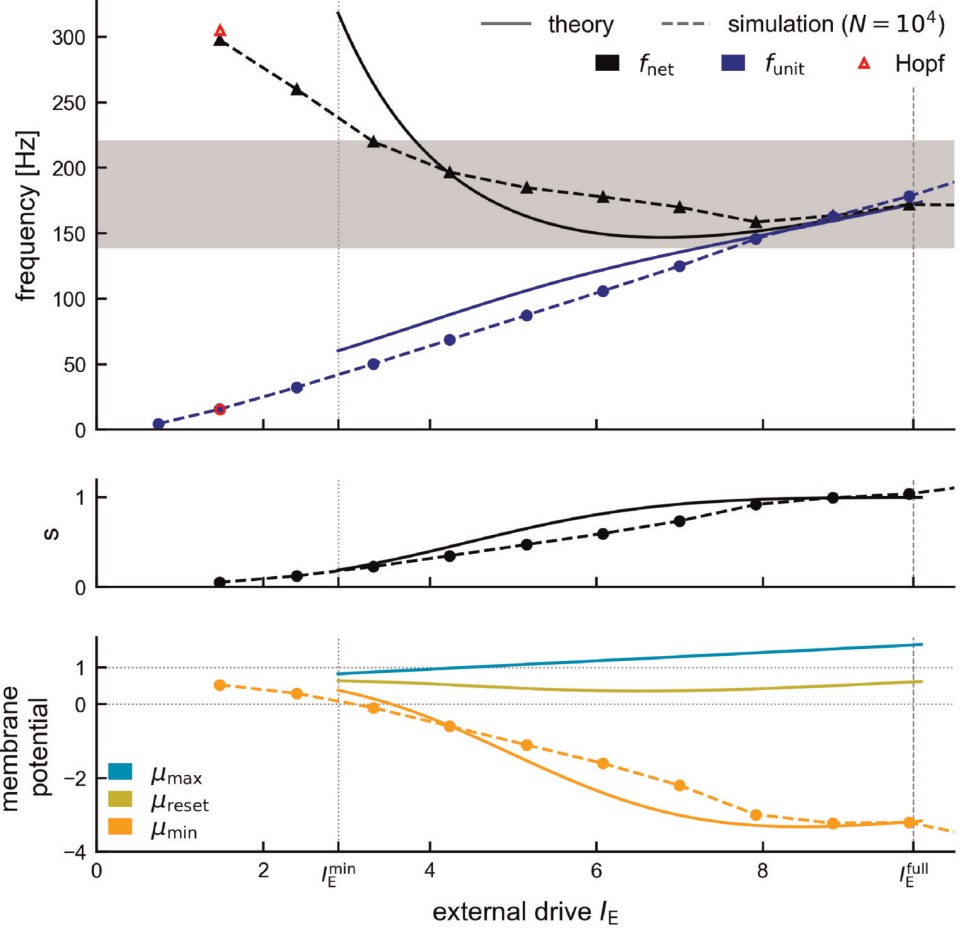

**Fig 4. Analytical approximation of the oscillation dynamics for constant drive.** Comparison of dynamics in theory (full lines) and spiking network simulation (dashed lines, $N = 10,000$). Top: Network frequency (black triangles) and mean unit frequency (blue circles). Red markers: Hopf bifurcation. Vertical lines indicate the range $[I_E^{min}, I_E^{max} = I_E^{full}]$ for which the theory applies (see Methods, Eq (52)). Middle: Saturation $s$ increases monotonically with the drive (Eq (48)). Bottom: characterization of the underlying mean membrane potential dynamics via local maximum $\mu_{max}$ (cyan, Eq (38)), local minimum $\mu_{min}$ (orange, Eq (50)) and population reset $\mu_{reset}$ (yellow, Eq (49)). Default parameters (see Methods).

required for single units to fire approximately at the frequency of the network rhythm ($s(I_E^{full}) \approx 1$; Methods, Eq (51); Fig 4, vertical dashed line). The Gaussian-drift approximation slightly overestimates the point of full synchrony, but correctly predicts its parameter dependencies: For stronger coupling and/or larger noise, stronger external drive is required to reach full synchrony (Fig B in S1 Appendix).

The theory shows that the amplitude $\mu_{max} - \mu_{min}$ of the oscillatory mean membrane potential grows with increasing drive (Fig 4, bottom). This is mainly due to a strong decrease of the periodic minimum $\mu_{min}$ (Fig 4, bottom, solid orange line), which we also observe in the spiking network simulation (Fig 4, bottom, dashed orange line): stronger external drive increases the amplitude of the population spikes (Fig 1A), resulting in stronger recurrent inhibitory feedback and thus a stronger hyperpolarization of the membrane potentials. The quantities $\mu_{max}$ and $\mu_{reset}$ are pertinent to the Gaussian-drift approximation and have no direct counterpart in the spiking neural network model.

The range of applicability of our theory is defined by our two assumptions (see Methods for details, Eq (52)): (a) units should spike at most once per cycle (*i.e.* $I_\mathrm{E} \leq I_\mathrm{E}^\mathrm{full}$), (b) in between population spikes the bulk of the membrane potential distribution should be subthreshold (*i.e.* $\mu_\mathrm{min}(I_\mathrm{E}) + 3\sqrt{D} \leq V_T$). The resulting range $[I_\mathrm{E}^\mathrm{min}, I_\mathrm{E}^\mathrm{max}]$ for the external current $I_\mathrm{E}$ covers the large part of the regime of sparse synchrony up to the point of full synchrony (Fig 4, here $I_\mathrm{E}^\mathrm{max} = I_\mathrm{E}^\mathrm{full}$). We confirmed with numerical simulations that for strong enough drive the Gaussian-drift approximation works for a wide parameter regime *w.r.t.* noise, coupling strength, and synaptic delay (S1 Appendix).

At low drive ($I_\mathrm{E} < I_\mathrm{E}^\mathrm{min}$) this theory breaks down (see also Methods, section *Numerical analysis of oscillation dynamics*). This is to be expected from a purely drift-based approximation. The dynamics of the spiking network close to its supercritical Hopf bifurcation is largely fluctuation-driven. Such dynamics cannot be captured by focusing only on the oscillation of the mean membrane potential, which has vanishing amplitude as the drive approaches its critical value. This limitation of the theory does not pose a problem, since (a) the fluctuation-driven dynamics around the Hopf bifurcation has already been studied in depth by [34] and (b) our main goal here is to explain the IFA dynamics, which happens in the strongly mean-driven regime $I_\mathrm{E}(t) \gg I_\mathrm{E}^\mathrm{min}$.

So far we have studied the case of constant drive describing the *asymptotic* dynamics of sustained ripple oscillations observed in the long-time limit, after initial conditions have been forgotten. This will be emphasized from here on by adding a superscript "$\infty$". The simulations in Fig 2 already suggested that IFA emerges due to a deviation of the transient dynamics from these asymptotic dynamics. We will now use the established Gaussian-drift approximation to understand why. In a first step we will maintain the assumption of constant drive and study the transient dynamics that can be induced by perturbations of the initial condition (Fig 5A and 5B). With *piecewise* constant drive as a simplistic model of a SPW-like drive we then demonstrate the core mechanism of IFA: a hysteresis of the mean membrane potential dynamics (Fig 5B and 5C). We then extend the same formalism to piecewise linear drive, as a more realistic approximation of SPW-like drive, and in order to demonstrate that the IFA asymmetry is preserved even when the drive is fully symmetric (Fig 6). The Gaussian-drift approximation for linear drives allows us to study the dependence of IFA on the slope of the drive. We demonstrate that the hysteresis effect causing IFA is *speed-dependent* (Fig 7)—a novel, testable prediction of the feedback-based inhibition-first ripple model matching the simulation results from the spiking network (Fig 2D).

**Analysis of oscillation dynamics for piecewise constant, sharp wave-like drive.** Even for constant drive, there is *transient* dynamics if the initial mean membrane potential deviates from the asymptotic minimum $\mu_\mathrm{min}^\infty$. Let us assume that a cycle starts with an initial mean membrane potential $\mu_\mathrm{min} \neq \mu_\mathrm{min}^\infty(I_\mathrm{E})$. We only require that $\mu_\mathrm{min}$ be sufficiently subthreshold, such that the initial population rate is close to zero. What will be the period of the first cycle and how long does it take until the asymptotic dynamics is reached?

First we note that, independent of $\mu_\mathrm{min}$, the mean membrane potential will rise towards the asymptotic $\mu_\mathrm{max}^\infty(I_\mathrm{E})$, which can be shown to be independent of the initial condition (Methods, Eq (38)). Thus, only the duration of the first upstroke will be influenced by the initial condition, and the asymptotic dynamics is reached immediately thereafter (Fig 5Ai and 5Aii).

The duration of the first upstroke depends on the distance that the mean membrane potential has to travel, from its initial value $\mu_\mathrm{min}$ to the next peak $\mu_\mathrm{max}^\infty(I_\mathrm{E})$. For $\mu_\mathrm{min} = \mu_\mathrm{min}^\infty(I_\mathrm{E})$ the upstroke has length $t_\mathrm{off}^\infty(I_\mathrm{E})$ and hence the length of the first cycle is equal to the asymptotic period $T^\infty$. Correspondingly, the instantaneous frequency is equal to the asymptotic network frequency, $f_\mathrm{net}^\mathrm{inst} = f_\mathrm{net}^\infty = 1/T^\infty$ (Fig 5Aiii, middle). The upstroke takes *less* time, if the mean

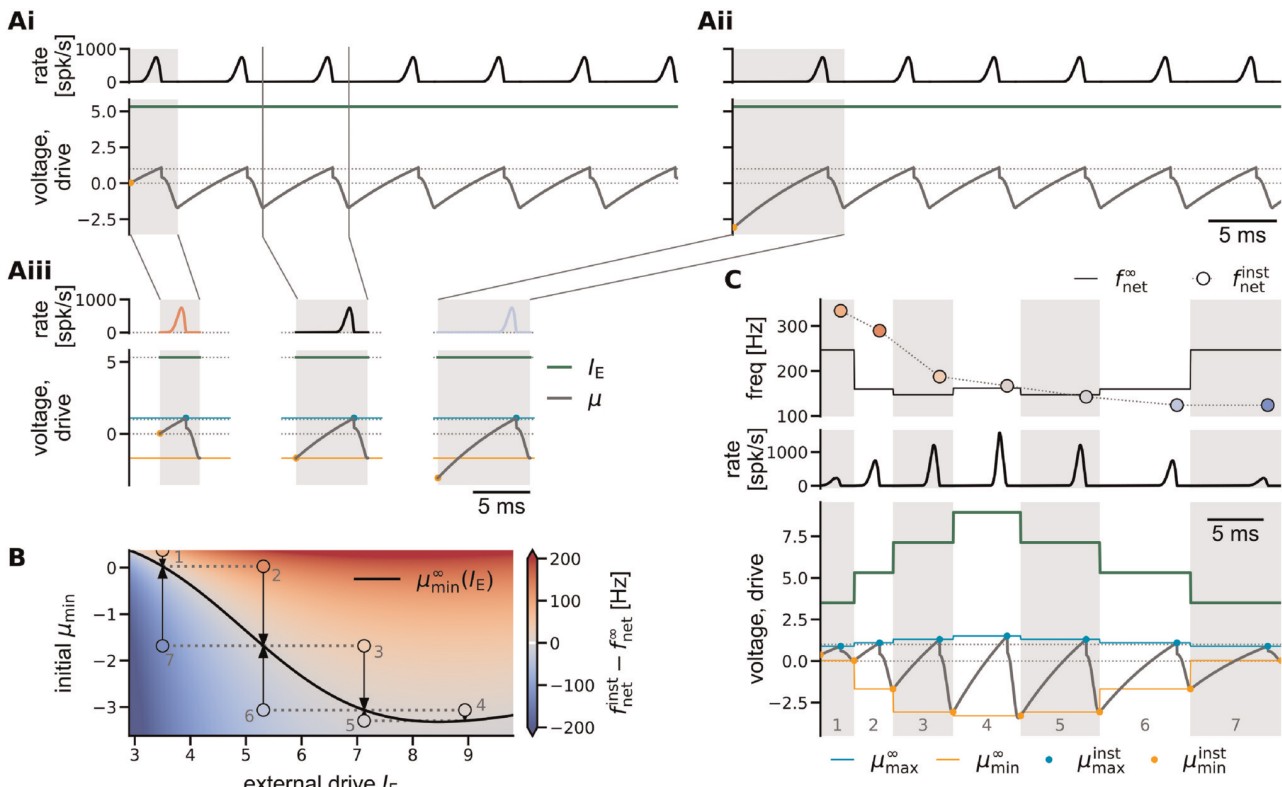

**Fig 5. Transient dynamics and IFA for piecewise constant external drive.** (Ai, Aii) Dynamics under constant drive depending on the initial mean membrane potential. Top: Population rate. Bottom: Constant external drive (green) and mean membrane potential (gray) with initial value $\mu_{\min}$ (orange marker). Dotted horizontal lines mark spike threshold $V_T = 1$ and reset potential $V_R = 0$. (Aiii) Direct comparison of the first oscillation cycles in Ai/Aii (gray shaded area) with the asymptotic cycle dynamics. Orange and cyan horizontal lines mark the asymptotic values for $\mu_{\min}^\infty(I_E)$ and $\mu_{\max}^\infty(I_E)$, respectively. Left: shorter cycle for $\mu_{\min} > \mu_{\min}^\infty(I_E)$. Middle: asymptotic period for $\mu_{\min} = \mu_{\min}^\infty(I_E)$. Right: longer cycle for $\mu_{\min} < \mu_{\min}^\infty(I_E)$. The color of the population rate curve (left, right) expresses the difference in cycle length as a difference in instantaneous frequency (colorbar in B). (B) Difference between the instantaneous frequency of a cycle with constant drive $I_E$ and initial condition $\mu_{\min}$, and the asymptotic frequency $f_{\text{net}}^\infty(I_E)$ for a range of external drives $I_E$ and initial mean membrane potentials $\mu_{\min}$. Black line: asymptotic $\mu_{\min}^\infty(I_E)$ (cf. Fig 4, bottom, orange line). Markers indicate example cycles shown in C. Arrows indicate convergence to the asymptotic dynamics after one cycle. If the drive changes after each cycle (dotted lines), the seven examples lead to the trajectory shown in C. Cycles 2 and 6 are also shown in Aiii (left, right), together with their common asymptotic reference dynamics. IFA for piecewise constant drive with symmetric step heights. Shaded areas mark oscillation cycles. Bottom: The external drive is increased step-wise, up to the point of full synchrony $I_E^{\text{full}} \approx 8.9$ (green staircase). As in A, lines in all panels indicate the asymptotic dynamics associated to the external drive of the respective cycle. Circular markers indicate transient behavior. Cyan: $\mu_{\max}$. Orange: $\mu_{\min}$. Reset not marked to enhance readability. Gray line: trajectory of the mean membrane potential. Middle: Population rate. Top: the instantaneous network frequency (markers) is first above and then below the respective asymptotic network frequencies (black line). Same colorbar as B. All quantities are derived analytically from the Gaussian-drift approximation. Vertical axis labeled "voltage, drive" in panels A and C applies to membrane potential and external drive.

membrane potential starts at a *higher* value ($\mu_{\min} > \mu_{\min}^\infty(I_E)$, Fig 5Ai and 5Aiii, left), and *more* time otherwise (Fig 5Aii and 5Aiii, right). Hence the period of the first cycle will be either shorter or longer, which can be rephrased as an instantaneous frequency $f_{\text{net}}^{\text{inst}}$ that is higher or lower than the asymptotic $f_{\text{net}}^\infty(I_E)$. Fig 5B illustrates the instantaneous frequency of the first cycle for different combinations of (constant) drive $I_E$ and initial condition $\mu_{\min}$ (red: instantaneous frequency is higher than asymptotic frequency; blue: instantaneous frequency is lower).

Once the mean membrane potential has reached its first peak, it will follow the asymptotic dynamics, settling into $\mu_{\min}^\infty(I_E)$ at the end of the first cycle, and all subsequent cycles will come at the asymptotic frequency $f_{\text{net}}^\infty(I_E)$ associated to the external drive $I_E$ (Fig 5Ai and 5Aii, convergence indicated by arrows in Fig 5B). A change of initial condition can thus only introduce

a transient deviation from the asymptotic dynamics in a single cycle. In particular, this implies that stimulation with a square pulse cannot induce IFA in this model (see also Discussion and Fig A in S2 Appendix, panels a5–d5).

What if we change the external drive after each cycle (green line in Fig 5C)? Then the initial mean membrane potential of each cycle $i$ will be the asymptotic minimum associated to the drive of the *previous* cycle:

$$\mu^i_{min} = \mu^\infty_{min}(I^{i-1}_E)$$

*i.e.* the mean membrane potential dynamics exhibits a history dependence (or *hysteresis*). Now recall that the asymptotic minimum $\mu^\infty_{min}(I_E)$ is a monotonically decaying function of the drive (except for very strong drive close to $I^{full}_E$, Fig 5B, black line). Thus, if the external drive *in*creases stepwise, each cycle starts with an initial mean membrane potential *above* the asymptotic minimum associated to that cycle's drive, hence the instantaneous frequency is *above* its asymptotic value (Fig 5B, trajectory through red area: $f^{inst,i}_{net} > f^\infty_{net}(I^i_E) \, \forall i$). Vice versa, if the external drive *de*creases stepwise, each cycle starts with an initial mean membrane potential *below* the asymptotic minimum associated to that cycle's drive, hence the instantaneous frequency is *below* its asymptotic value (Fig 5B, trajectory through blue area: $f^{inst,i}_{net} < f^\infty_{net}(I^i_E) \, \forall i$). In summary:

$$I^i_E > I^{i-1}_E \quad \Rightarrow \quad \mu^i_{min} = \mu^\infty_{min}\left(I^{i-1}_E\right) > \mu^\infty_{min}\left(I^i_E\right) \quad \Rightarrow \quad f^{inst,i}_{net} > f^\infty_{net}\left(I^i_E\right)$$

$$I^i_E < I^{i-1}_E \quad \Rightarrow \quad \mu^i_{min} = \mu^\infty_{min}\left(I^{i-1}_E\right) < \mu^\infty_{min}\left(I^i_E\right) \quad \Rightarrow \quad f^{inst,i}_{net} < f^\infty_{net}\left(I^i_E\right)$$

Thus, if we approximate the transient change in drive during a sharp wave as a simple, piecewise constant function that first increases after each cycle, and then decreases (Fig 5C, green line), we observe IFA: The asymptotic network frequency associated to the drive in each cycle describes a reference curve that follows the same symmetry as the drive (Fig 5C, top, solid black line). However, the instantaneous network frequency (Fig 5C, top, round markers) is asymmetric over time, as it is *above* the asymptotic network frequencies during the rising phase of the external drive, and *below* during the falling phase. The theory thus describes the relationship between instantaneous and asymptotic frequencies that was already described for the spiking network simulations in Fig 2D.

The piecewise constant shape of the drive may not be realistic, but serves to illustrate the core mechanism of IFA: a hysteresis in the oscillation amplitude of the mean membrane potential. A drawback is that this simple model for SPW-like drive is not symmetric in *time*, since the drive changes after each cycle, and the cycle length increases due to IFA. To show that the IFA asymmetry does not rely on an asymmetry in the drive, we adapted the Gaussian-drift approximation to incorporate time-dependent linear drive.

**Analysis of oscillation dynamics for piecewise linear, sharp wave-like drive.** Following the same approach as before, we approximate the *transient* dynamics of the mean membrane potential in a cycle $i$ with initial value $\mu^i_{min}$ and a drive that changes linearly around a level $\hat{I}^i_E$ with slope $m$:

$$\left(\hat{I}^i_E, \mu^i_{min}, m\right) \mapsto \left(f^{inst,i}_{net}, f^i_{unit}, t^i_{off}, \mu^i_{max}, \mu^i_{reset}, \mu^{i+1}_{min}\right) \tag{11}$$

The dynamics is quantified in terms of the peak of the mean membrane potential $\mu^i_{max}$, its reset value $\mu^i_{reset}$, and the value $\mu^{i+1}_{min}$ that is reached at the end of cycle $i$ (and may thus be the initial membrane potential of the next cycle $i + 1$). Most importantly, the duration of the upstroke $t^i_{off}$

is inferred, and from that the instantaneous network frequency $f_{\text{net}}^{\text{inst},i} = (t_{\text{off}}^i + \Delta)^{-1}$ (see Methods, Eqs (56)–(66)). In agreement with our theoretical approximation, which is anchored to the end of the population spike, the reference drive $\hat{I}_E^i$ is chosen such that $\hat{I}_E^i = I_E(t_{\text{off}}^i)$, *i.e.*

$$I_E(t) = \hat{I}_E^i + m(t - t_{\text{off}}^i) \ . \tag{12}$$

The analysis is now a little more complex, since each cycle depends on three parameters $(\hat{I}_E^i, \mu_{\min}^i, m)$, in contrast to only two parameters $(I_E^i, \mu_{\min}^i)$ for piecewise constant drive. We will demonstrate, however, that the essential findings from the basic case of piecewise constant drive still hold, *i.e.* that IFA is generated by the same hysteresis in the transient dynamics of the mean membrane potential that we have uncovered before (Fig 5 vs Fig 6). To illustrate this by an example, we fix the slope of the linear drive to $m = \pm0.4$/ms. We then compare how the transient dynamics deviates from the asymptotic dynamics, depending on whether the drive is increasing ($m = +0.4$/ms) or decreasing ($m = -0.4$/ms) (Fig 6).

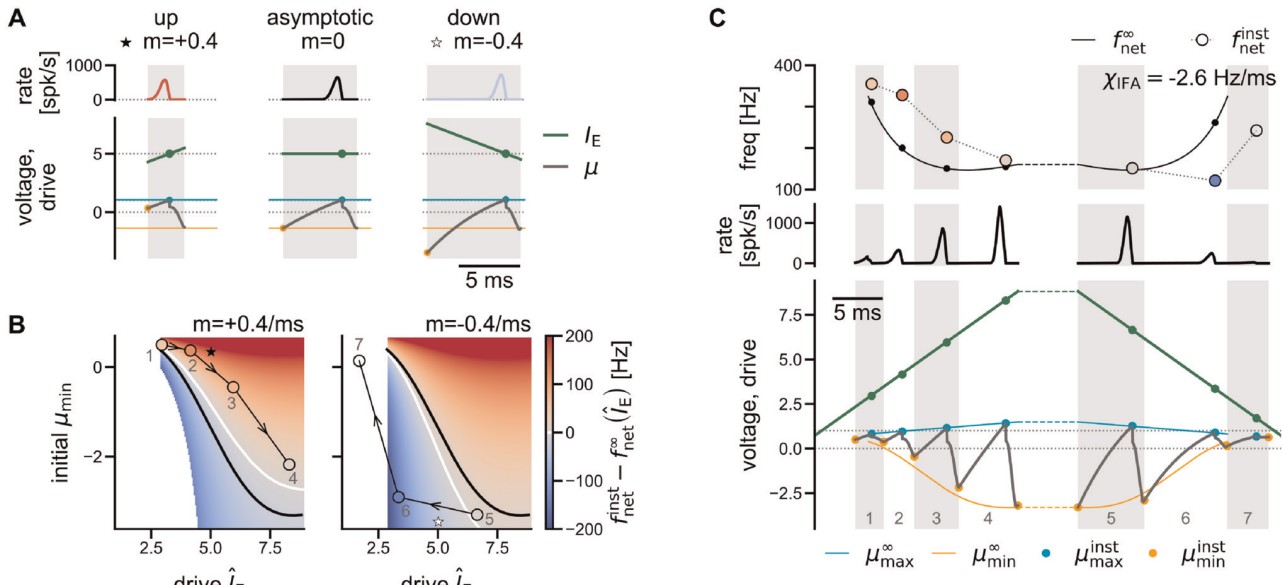

**Fig 6. Transient dynamics and IFA for piecewise linear external drive.** (A) Exemplary transient dynamics during rising vs falling phase of the external drive ("up" vs "down"), given fixed $\hat{I}_E = I_E(t_{\text{off}}) = 5$ (green dot). Bottom: external drive (green lines), and trajectories of mean membrane potential (gray lines) depending on initial conditions (orange dots). Dotted horizontal lines mark reference drive $\hat{I}_E$, spike threshold $V_T = 1$, and reset potential $V_R = 0$. Top: Population rate. Color (colorbar in B) indicates the difference in cycle length (shaded area) compared to the asymptotic reference (middle panel). Left: shorter cycle for $m > 0$ and initial $\mu_{\min} \gg \mu_{\min}^\infty(\hat{I}_E)$ (orange dot vs orange line). Middle: asymptotic period for constant drive ($I_E \equiv \hat{I}_E$, $m = 0$) and initial $\mu_{\min} = \mu_{\min}^\infty(\hat{I}_E)$. Right: longer cycle for $m < 0$ and initial $\mu_{\min} < \mu_{\min}^\infty(\hat{I}_E)$. (B) Difference between instantaneous and asymptotic frequency for a range of reference drives $\hat{I}_E$ and initial mean membrane potentials $\mu_{\min}$. Left: linearly increasing drive with slope $m = +0.4$/ms. Right: linearly decreasing drive with slope $m = -0.4$/ms. Black line: asymptotic $\mu_{\min}^\infty(\hat{I}_E)$ for constant drive. White line: initial membrane potential $\mu_{\min}$ for which $f_{\text{net}}^{\text{inst}} = f_{\text{net}}^\infty(\hat{I}_E)$. Stars mark the examples shown in A for $\hat{I}_E = 5$. Round markers and arrows indicate the trajectory shown in C for piecewise linear drive, numbered by cycle. White space where either: no asymptotic oscillations occur ($I_E < I_E^{\min}$, $\mu_{\min} > V_T - 3\sqrt{D}$), or (bottom left): no transient solution exists (see Methods, Eq (64)). (C) IFA for symmetric, piecewise linear (SPW-like) drive. Shaded areas mark oscillation cycles. Bottom: The external drive is increased up to the point of full synchrony $I_E^{\text{full}} \approx 8.9$ (green line). Colored lines indicate asymptotic dynamics. Gray line: mean membrane potential trajectory $\mu(t)$. Markers quantify transient behavior. Cyan: $\mu_{\max}$. Orange: $\mu_{\min}$. Reset not marked to enhance readability. Middle: Population rate. Top: the instantaneous network frequency (markers) is first above, then below the resp. asymptotic network frequencies (black line). Same colorbar as B. Dashed lines: plateau phase of variable length with $I_E \equiv I_E^{\text{full}}$, during which the network settles into the asymptotic dynamics. The IFA slope $\chi_{\text{IFA}}$ was derived for an assumed plateau length of 20 ms (as in Fig 2). All quantities are derived analytically. Vertical axis labeled "voltage, drive" in panels A and C applies to membrane potential and external drive.

The principal insight from the case of constant drive still holds for the case of time-dependent drive, except for a small range of initial values $\mu_{\min}^i$: If the initial mean membrane potential $\mu_{\min}^i$ is (sufficiently) larger than the asymptotic reference $\mu_{\min}^\infty(\hat{I}_E^i)$, the period is shorter than the asymptotic reference $T^\infty$, hence $f_{\text{net}}^{\text{inst},i} > f_{\text{net}}^\infty(\hat{I}_E^i)$ (Fig 6A, left; Fig 6B, red area). In contrast, if $\mu_{\min}^i$ is smaller than the asymptotic reference $\mu_{\min}^\infty(\hat{I}_E^i)$, the period is longer than the asymptotic reference $T^\infty$, hence $f_{\text{net}}^{\text{inst},i} < f_{\text{net}}^\infty(\hat{I}_E^i)$ (Fig 6A, right; Fig 6B, blue area). Exceptions from this "rule" (Fig 6B, area between black and white lines) occur because for time-dependent drive an asymptotic initial value $\mu_{\min}^i = \mu_{\min}^\infty(\hat{I}_E^i)$ (black line) no longer implies the asymptotic period $T^\infty$ and frequency $f_{\text{net}}^{\text{inst},i} = f_{\text{net}}^\infty(\hat{I}_E^i)$ (white line, see Methods for details). We note, however, that these exceptions from the rule occur only in a small portion of the state space that is rarely visited in a given ripple event, as we will see in the following.

What can we say about the dynamics of consecutive cycles $i$, $i + 1$, . . . that occur if the drive rises or falls continuously with slope $m$? At the end of each cycle the mean membrane potential is close to the asymptotic reference $\mu_{\min}^\infty(\hat{I}_E^i)$ (Methods, Eq (66)). Thus we observe the same hysteresis as before: if the drive *increases* ($m > 0$, Fig 6B, left), trajectories of consecutive cycles will lie in the upper right half of the parameter space, where every cycle starts with an initial mean membrane potential that is higher than its asymptotic reference $(\mu_{\min}^i \gtrsim \mu_{\min}^\infty(\hat{I}_E^{i-1}) > \mu_{\min}^\infty(\hat{I}_E^i))$, and hence has an instantaneous frequency that is (mostly) *higher* than the asymptotic reference (red color code). Vice versa, as the drive *decreases* ($m < 0$, Fig 6B, right), trajectories will lie in the lower left half $(\mu_{\min}^i \lesssim \mu_{\min}^\infty(\hat{I}_E^{i-1}) < \mu_{\min}^\infty(\hat{I}_E^i))$ where instantaneous frequencies are mostly *lower* than their asymptotic reference (blue color code). Hence even a *symmetric*, piecewise linear double ramp drive (Methods, Eq (17)), induces the IFA asymmetry (Fig 6C): During the rising phase of the drive the instantaneous frequencies are above the asymptotic reference, and during the falling phase they lie below (Fig 6C, top: markers vs black line; note cycle 5 as the only exception from the above "rule"). The IFA asymmetry thus does *not* rely on asymmetry in the input. Linear regression over the (semi-)analytically estimated instantaneous frequencies yields an IFA slope of −2.6 Hz/ms which is close to the spiking network simulation (Fig 2B).

Interestingly, the last cycle $i = 7$ in Fig 6C has a reference drive for which the constant-drive theory no longer applies ($\hat{I}_E^7 = 1.7 < 2.85 = I_E^{\min}$), hence there is no asymptotic reference for the network frequency (empty marker, Fig 6C, top). This means that at the end of the sharp wave the network can sustain one more ripple cycle at a level of drive that in the beginning would be insufficient to trigger ripples (see cycle 1 in Fig 6C). The transient ripple is thus not only asymmetric in its instantaneous frequency (IFA), but also with respect to the level of drive at which it starts and ends.

We have established that IFA occurs in response to transient, sharp wave-like drive, independent of its symmetry, due to a hysteresis effect in the amplitude of the oscillatory mean membrane potential. Varying the slope $m$ in our double ramp model for SPW-like drive (Methods, Eq (17)) we find that this hysteresis is *speed-dependent* (Fig 7): If the drive changes more slowly, the transient dynamics approaches the asymptotic dynamics, and the IFA asymmetry is reduced ($\chi_{\text{IFA}} \rightarrow 0$ for $|m| \rightarrow 0$, Fig 7A–7C; Methods, Eqs (56)–(66)). The theory thus explains the speed-dependence of IFA that we already observed in the spiking network simulations (Fig 2D). The theoretically predicted instantaneous frequencies and IFA slopes are in excellent agreement with the corresponding observations in the spiking neural network simulation if the slope is sufficiently strong (Fig 7Aii and 7Bii, colored vs grey markers, average relative error $\epsilon$: 11–13%, Table 1). The discrepancies between theory and simulation for slow drive

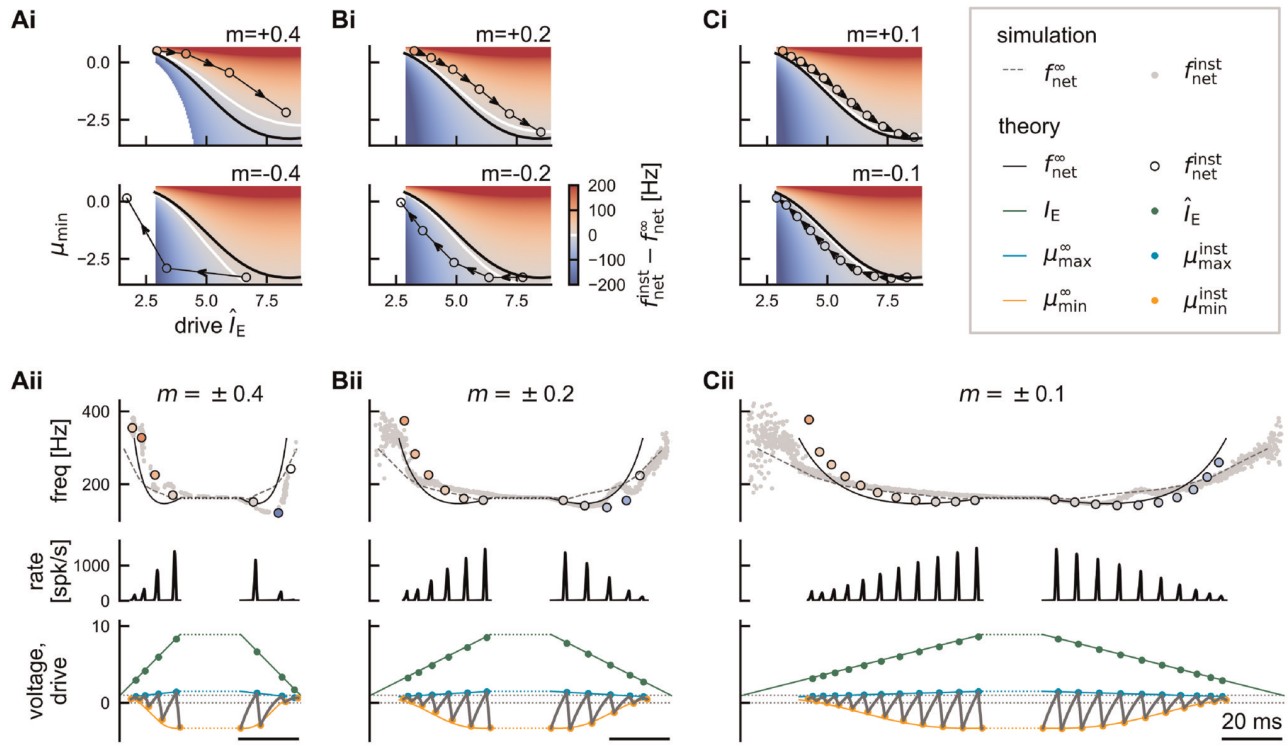

**Fig 7. IFA is speed-dependent.** Transient dynamics for SPW-like drive with slope $m = \pm 0.4$/ms (**A**, cf. Fig 6), $m = \pm 0.2$/ms (**B**), or $m = \pm 0.1$/ms (**C**). Top panels (**i**): difference between instantaneous and asymptotic network frequency for the possible combinations of external reference drive $\hat{I}_E$ and initial mean membrane potential $\mu_{min}$, shown separately for positive (top) and negative (bottom) slope of the drive. Trajectories shown in (ii) are overlaid. Bottom panels (**ii**): Example trajectories through the space shown in (i) for consecutive cycle under SPW-like drive with slope $\pm m$. Top: instantaneous (colored markers) vs asymptotic (black solid line) network frequency as predicted by the theory. Grey dots indicate instantaneous frequencies in spiking network simulations (cf. Fig 2D, $N = 10,000$). Black dashed line illustrates asymptotic network frequencies from spiking network simulations (cf. Fig 1B, $N = 10,000$). See Table 1 for a quantitative comparison of simulation and theory. Middle: population rate as predicted by the Gaussian-drift approximation. Bottom: instantaneous vs asymptotic $\mu_{min}$ (orange) and $\mu_{max}$ (cyan). Gray line: trajectory of mean membrane potential $\mu(t)$. Green lines show SPW-like external drive (Eq (17)), green dots mark reference drive $\hat{I}_E$ of each cycle. The difference between instantaneous and asymptotic network frequencies (IFA) becomes less pronounced for smaller slope (C vs A, see also Table 1).

(Fig 7Cii, colored vs grey markers) are mainly due to the discrepancies in the estimate of the asymptotic reference frequencies for constant drive (Fig 4).

The speed-dependence of IFA is an important prediction of the feedback-based inhibitory ripple model that can be tested in experiments: Optogenetic stimulation of PV$^+$ basket cells can trigger ripple oscillations in the LFP [21] or induce ripple-modulated spiking activity [43]. Increasing and decreasing the intensity of the light pulse could mimic the piecewise linear,

**Table 1. IFA in theory and simulation.** Quantification of the IFA slope $\chi_{IFA}$ in the spiking network simulations and the theoretical approximations shown in Fig 7Aii–7Cii for different slopes $m$ of the external SPW-like drive. The error $\epsilon$ (Eq (67)) quantifies the average relative deviation of the theoretically predicted instantaneous network frequencies (colored markers in Fig 7) from the simulation results (grey dots in Fig 7).

| slope of SPW-like drive [1/ms] | $m = \pm 0.4$ | $m = \pm 0.2$ | $m = \pm 0.1$ |
|---|---|---|---|
| IFA slope $\chi_{IFA}$ (Hz/ms, simulation) | − 3.04 | − 0.74 | − 0.29 |
| IFA slope $\chi_{IFA}$ (Hz/ms, theory) | − 2.60 | − 1.45 | − 0.51 |
| error $\epsilon$ in instantaneous frequency | 11% | 13% | 13% |

double ramp drive studied here. The model predicts that the IFA asymmetry is reduced if the optogenetic light stimulus changes more slowly.

## Discussion

We studied the dynamics of hippocampal ripple oscillations in a feedback-based inhibition-first model. By reducing the model complexity we derived an analytical approximation of the mean-field dynamics in the regime of strong drive and strong coupling. For constant drive, our theory (i) yields an approximation of the asymptotic network frequencies and mean unit firing rates far beyond the Hopf bifurcation, (ii) captures the transition from sparse to full synchrony, and (iii) reveals an increase of the mean membrane potential oscillation amplitude for increasing levels of external drive. For a fast changing, sharp-wave like drive we then show that a speed-dependent hysteresis effect in the trajectory of the mean membrane potential produces an IFA-like asymmetry of the instantaneous ripple frequency compared to the asymptotic frequencies. Our analytical approach presents a substantial advancement over previous, exemplary, numerical demonstrations of IFA [37] as it allows a mechanistic understanding of the phenomenon and its parameter dependencies. Our derivation shows that IFA is an intrinsic feature of the feedback-based inhibitory ripple model and that IFA occurs for any fast-enough sharp wave-like drive, independent of other parameter choices. The speed-dependence of IFA is a new prediction that can be tested experimentally.

### Simplifying assumptions

To achieve an analytical treatment of the spiking network dynamics, we have made a number of simplifying assumptions. Our resulting *reduced model* with full recurrent connectivity is a special case of the network analyzed in [34] with sparse recurrent connectivity. We demonstrate in Figs 1 and 2 that the reduced model exhibits ripples and IFA similar to a biologically more realistic model [37] that includes random sparse connectivity, correlations in the background noise, synaptic filtering, absolute refractoriness, local pyramidal cells, and conductance-based synapses. Let us discuss these differences in more detail:

*Pyramidal cells*. Inhibition-first models posit that the interneuron network generates ripple oscillations and entrains the spiking of local CA1 pyramidal cells. Thus we neglect pyramidal cells in our reduced model—we regard them as a non-essential ingredient. We have no *a priori* reason to believe that the dynamics of IFA would be qualitatively altered by the presence of pyramidal cells: Numerical simulations of the feedback-based model including pyramidal cells have shown that (i) the inhibitory network can entrain the local pyramidal cell network, and (ii) IFA can occur in the spiking activity of both the excitatory and inhibitory population [37]. The influence of pyramidal cells is most relevant when it comes to determining the boundaries between the parameter spaces for ripple and gamma oscillations [37], or the relation between neuronal spiking activity and LFP [24, 38].

*Absolute refractoriness*. If an absolute refractory period is added to the reduced model in its default parameter setting (Table 2), the membrane potential distribution becomes bimodal and the network activity oscillates with alternating clusters of active neurons [60]—a state that is beyond the scope of our theory. Unimodality of the membrane potential distribution can easily be recovered by increasing the noise intensity. In the regime of sparse synchrony the network frequency is then virtually independent of the refractory period [60]. The refractory period only leads to a slight delay of the point of full synchrony, which also occurs at a lower network frequency due to the decrease in mean unit firing rate.

*Conductance-based synapses*. A noteworthy *qualitative* difference in the network dynamics of our reduced model compared to the model in [37] comes from the use of current-based

**Table 2. Default parameters for the spiking network.**

| Parameter | Value | Definition |
|---|---|---|
| $N$ | 10,000 | Number of interneurons |
| $\tau_m$ | 10 ms | Membrane time constant |
| $C$ | 100 pF | Membrane capacitance |
| $E_{\text{leak}}$ | -65 mV | Resting potential |
| $V_{\text{thr}}$ | -52 mV | Spike threshold |
| $V_{\text{reset}}$ | - 65 mV | Reset potential |
| $J$ | 65 mV | Inhibitory coupling strength |
| $\Delta$ | 1.2 ms | Synaptic delay |
| $\sigma_V$ | 2.62 mV | Standard deviation of Gaussian white noise input |
| $V_T$ | 1 | Spike threshold, dimensionless |
| $V_R$ | 0 | Reset potential, dimensionless |
| $K$ | 5 | Inhibitory coupling strength, dimensionless |
| $D$ | 0.04 | Variance of Gaussian white noise input, dimensionless |

instead of conductance-based synapses. In our network with current-based synapses, an increase in external drive, and thus recurrent feedback, leads to an increase of the hyperpolarization of the membrane potentials, and the achieved membrane voltages can be quite low ($< -100$ mV in Fig 1A; $\mu_{\text{min}}$ in Fig 4). In contrast, in a network with conductance-based synapses, the decrease in $\mu_{\text{min}}$ would still be monotonic, but bounded by the inhibitory reversal potential, and thus confined to a biologically more realistic range. Moreover, inhibitory feedback in a conductance-based model *compresses* the membrane potential distribution around the reversal potential. To accurately capture the resulting oscillation dynamics, it would thus be necessary to capture not only the first, but also the *second* moment of the moving membrane potential distribution. We developed such an extension of the Gaussian-drift approximation for a toy model of pulse-coupled oscillators with linear phase-response curve, which mimicks the effect of an inhibitory reversal potential [60]. We find again that IFA is caused by a hysteresis effect, but now with respect to two variables: the mean and the variance of the membrane potential distribution. However, a direct analytical treatment of the conductance-based LIF network of [37] is difficult and may be an interesting subject for future investigations.

*Short term plasticity and adaptation*. There are many more biological complexities that are neglected not only in our reduced model, but also in the reference simulations by [37]. There is evidence that PV$^+$ interneurons in CA3 exhibit short-term synaptic depression [61]. In principle, short term depression or facilitation could easily be incorporated in our Gaussian-drift approximation by making the synaptic strength $K$ cycle-dependent. Spike frequency adaptation would be another factor relevant for ripple dynamics. However, PV$^+$ interneurons in CA1 —some of the prime candidates for a potential inhibition-first ripple generator—do not exhibit strong adaptation [62].

## Performance and limitations of the Gaussian-drift approximation

Our reduced network model allows a Gaussian-drift approximation of its oscillation dynamics that can be treated analytically. We added a new phenomenological account for the reset mechanism on the population level, which improves the accuracy and range of applicability of the Gaussian-drift approximation (Fig 9A vs 9B). We would like to stress that this phenomenological reset is not *necessary* to capture the most important *qualitative* features of the network,

namely the transition from sparse to full synchrony for constant drive and the hysteresis effect leading to IFA for time-dependent drive.

The differences between the simulations of spiking units and the theory have been analyzed in detail for the case of constant drive (S1 Appendix): there was a good quantitative match for a large parameter range, and a slight decrease in performance towards high noise intensities. For a slow-enough time-dependent drive, any approximation error in the asymptotic frequencies at constant drive (Fig 7Cii, top, solid vs dashed line) also contributes to an error in the instantaneous frequencies (Fig 7Cii, top, colored vs gray markers). Furthermore, the trajectory of the instantaneous frequencies in the spiking network simulations is influenced by the non-gaussian shape of the membrane potential distribution, which cannot be captured by the theory. An example is the "wiggle" (local maximum) in the simulated instantaneous network frequencies during the falling phase of the drive as seen e.g. in Fig 7Bii and 7Cii (top, gray markers): It reflects a single cycle of reduced period that occurs, with some phase jitter, in each of the 50 simulations pooled together in the plot. Inspection of the membrane potential distribution during this cycle revealed a significant reset-induced bimodality (S1 Fig). Such details of the transient dynamics cannot be captured by the theory, but are also not crucial to the IFA phenomenon.

Since our approximation is purely drift based, it is not well behaved in the limit of low drive (called "pathological oscillation" in Fig 8). In that sense our approach is truly complementary

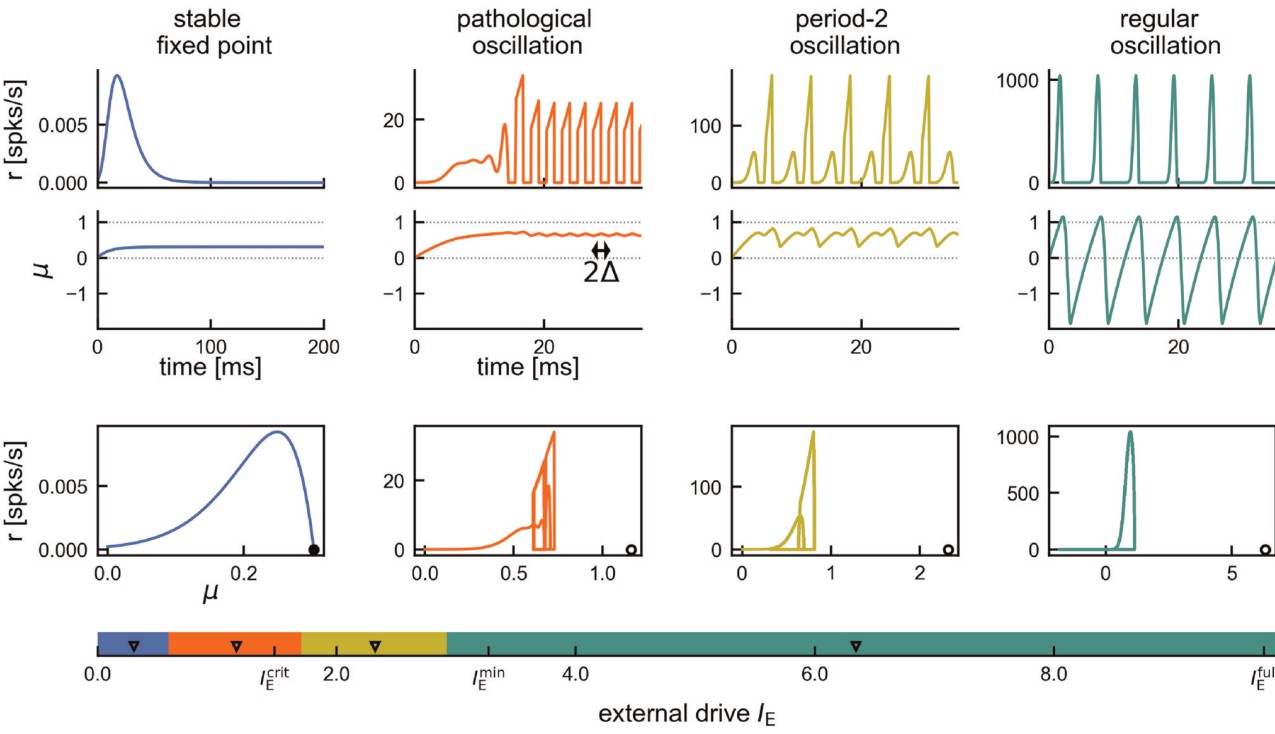

**Fig 8. Dynamical states of the DDE system.** Numerical integration of the DDE system (without phenomenological reset) demonstrating 4 distinct dynamical regimes for increasing external drive $I_E$ (bottom axis). Blue: Stable fixed point with zero rate. Red: Pathological, fast oscillations. Yellow: Period-2 oscillations. Green: Regular period-1 oscillations. Top: Numerically integrated trajectories of population rate $r$ and mean membrane potential $\mu$ over time. Note the changes in scale for the population rate. Bottom: Phase space showing the trajectory $(\mu(t), r(t))$ and the fixed point $(I_E, 0)$, which is only stable in the first case (black circle) and unstable otherwise (empty circle). The horizontal colored bar shows the full range of (relevant) drives from 0 to the point of full synchrony $I_E^{\text{full}}$. The boundaries between the four dynamical regimes were approximated numerically. Black triangles mark the levels, for which the above example dynamics are shown. Extra ticks indicate: critical drive $I_E^{\text{crit}} \approx 1.48$, for which the spiking network undergoes a Hopf bifurcation (see section *Linear Stability Analysis*); lower bound $I_E^{\text{min}}$ introduced for our theory to ensure that we only consider non-pathological dynamics (see paragraph *Range of applicability*).

to previous linear (or weakly nonlinear) analyses around the bifurcation, which break down at strong drive [34]. A theory that can describe the dynamics of recurrent spiking networks both in the drift- and diffusion-dominated regimes, as well as capture the transitions in between, would be desirable. In the context of escape noise models, various approaches have been put forward to approximate the combined effect of drift- and diffusion-mediated spiking [56–59, 63–65]. The rate expression in our Gaussian-drift approximation (Eq (8b)) is equivalent to the drift-based hazard function derived by [56, 58].

Our approach for analyzing the population rate dynamics in response to strong voltage transients may also be relevant in other contexts, such as gamma oscillations [66, 67], or propagation of synchronous neural activity in synfire-chains [29, 68]. A hysteresis effect similar to IFA was recently observed in optogenetically evoked gamma oscillations in cat visual cortex [69]. It may be an interesting subject for future research to extend the present Gaussian-drift approximation to gamma rhythms in E-I networks.

## The shape of the asymptotic network frequency curve as a function of the external drive

A hallmark of the feedback-based inhibition-first ripple model is its state of sparse synchrony, where, for a wide range of external drives, the collective population activity exhibits oscillations in the ripple range, while individual neurons fire sparsely and irregular at lower firing rates (Fig 1). In this state the asymptotic network frequency is not necessarily proportional to the external, excitatory drive, as is the case in networks at full synchrony or in clustered states [67, 70, 71].

In our reduced model, the asymptotic network frequency decreases with increasing external drive (Fig 1B). In the vicinity of the Hopf bifurcation and in the limit of small synaptic delay such a frequency decrease was shown analytically in [34]. In biologically more realistic model networks with an absolute refractory period and/or conductance-based synaptic coupling, the shape of the asymptotic network frequency curve can be quite different (see Fig A in S2 Appendix, panels a1–d1). Qualitatively, the hysteresis mechanism of IFA that we have uncovered here is preserved in all network architectures (Fig A in S2 Appendix, panels 2–4).

## The shape of IFA

Our theory predicts that instantaneous ripple frequencies for a changing drive are different from the asymptotic frequencies for constant drive: If the drive increases the instantaneous frequencies are higher than the asymptotic frequencies, and if the drive decreases the instantaneous frequencies are lower. For a symmetric time course of the drive, this asymmetry immediately implies IFA with a negative regression slope $\chi_{IFA}$ approximately equal to the regression slope over the *deviations* between instantaneous and asymptotic frequencies (see Methods, Eq (19)). For *asymmetric* drive, the IFA slope $\chi_{IFA}$ may be slightly stronger or weaker, depending on the covariance of the asymptotic reference frequencies with time (see Methods, Eq (19)). In general, the shape of the instantaneous ripple frequency curve over time thus depends both on the shape of the drive, and on the shape of the asymptotic frequency as a function of the external drive (Fig A in S2 Appendix). In our reduced model (Fig A in S2 Appendix, panel d) the asymptotic frequency decreases for increasing values of the constant drive. Thus it is possible for the instantaneous frequencies in the second half of the ripple event (when the external drive decreases) to lie *below* the asymptotic reference while actually *in*creasing as a function of time (Figs 2D and 7Aii–7Cii, Fig A in S2 Appendix, panels d2, d3). Interestingly, such a switch from decreasing to increasing frequency at the end of the ripple event was also found in some experimental data [44]. Others have reported an almost

monotonic decrease or even a small peak in the beginning of the event [25, 37, 43, 45]. In the reduced model, IFA gets stronger (more negative $\chi_{\text{IFA}}$) if the peak of the drive is asymmetrically shifted to the right (Fig A in S2 Appendix, panel d4). If we incorporate into the model an absolute refractory period (Fig A in S2 Appendix, panel c) or conductance-based synaptic coupling (Fig A in S2 Appendix, panels a,b) the shape of the asymptotic frequency curve changes such that the strongest IFA is achieved with a left-shifted stimulus (Fig A in S2 Appendix, panels a3, b3). It is only for rare combinations of model architecture and stimulus profile that the described asymmetry between instantaneous and asymptotic network frequencies does *not* imply IFA in the sense of $\chi_{\text{IFA}} < 0$: If a network with asymptotic frequencies that increase strictly monotonically with the drive is stimulated with a strongly right-shifted stimulus (Fig A in S2 Appendix, panel b4), the resulting ripple does not exhibit IFA. Our theory still correctly predicts this response: the instantaneous frequencies are first above, then below the asymptotic reference.

## Modeling sharp wave-like drive to CA1 interneurons

*Square pulse.* One may argue that a square pulse is the simplest possible model for the sharp wave-associated momentary increase in drive to CA1 interneurons (SPW-like drive). Our analysis, however, showed that transients under constant drive due to perturbations of initial conditions are short-lived ($\sim$ one cycle in theory, Fig 5). This implies in particular that a square pulse cannot account for IFA (Fig A in S2 Appendix).

Note that there is a subtlety when comparing the simulation of the reduced model under a square pulse (panel d5 of Fig A in S2 Appendix) with the theoretical analysis for piecewise constant drive in Fig 5: in Fig 5 we analyze the network response to stepwise changes in the drive that occur when the network is already in an oscillatory state (*i.e.* $\mu_{\text{min}} < V_T - 3\sqrt{D}$ is sufficiently subthreshold, otherwise the drift-based theory does not apply). In that case, an upwards step in the drive always corresponds to the red regime of Fig 5B, where the instantaneous frequency is higher than the asymptotic frequency. In Fig A in S2 Appendix, on the other hand, a square pulse is delivered to the network in the non-oscillatory steady-state. The stationary mean membrane potential in that state hovers just below the threshold (*i.e.* $\mu_{\text{min}} > V_T - 3\sqrt{D}$). If we apply the Gaussian-drift approximation nevertheless, this corresponds to a starting distribution of membrane potentials with a significant portion already above threshold. In that case a square pulse triggers a first population spike starting with a high, non-zero rate, that silences itself as soon as the inhibitory feedback starts ($t_{\text{off}} = \Delta$). Due to the strong inhibitory feedback this cycle can end with a mean membrane potential *below* the asymptotic $\mu_{\text{min}}^\infty$ associated to the amplitude of the square pulse. The next cycle thus starts in the blue regime of Fig 5B and is *longer* than the asymptotic period, before the system settles into the asymptotic dynamics. This explains the slight tendency for positive IFA slopes $\chi_{\text{IFA}}$ (Fig A in S2 Appendix, panel d5).

Most importantly, however, the network settles into the asymptotic dynamics very quickly ($|\chi_{\text{IFA}}| \sim 0$), so stimulation with a square pulse cannot account for realistic IFA in the feedback-based inhibition-first model.

In experiments, an optogenetic square pulse stimulation of CA1 pyramidal cells has been found to elicit IFA in the local field potential [9, 43]. An extension of the present model, including pyramidal cells and a proxy for the LFP, would be needed to determine the compatibility of the feedback-based inhibition-first model with these findings. It is plausible that a square pulse stimulation of pyramidal cells yields a non-square, double ramp-like drive to locally connected interneurons (see the ramp-like evolution of the evoked firing rates of pyramidal cells in Fig 2E of [9]). As our analysis emphasizes, any such fast-changing double ramp

drive to CA1 interneurons is likely to induce IFA. For a direct square-pulse stimulation of interneurons, on the other hand, the present reduced model predicts *no* IFA in the inhibitory population activity. This prediction could be tested in future experiments.

*Double ramp*. For most of the manuscript we used a double ramp drive as the simplest model of SPW-like drive that can account for IFA. We chose a symmetric double ramp to highlight that the asymmetric phenomenon of IFA does not depend on asymmetry in the input. Our derivation emphasizes that the only necessary requirement for IFA is an external drive that changes sufficiently fast and first rises, then decays. Thus we predict that IFA in the feedback-based model will occur for a wide range of such *SPW-like* external drives, even when their time courses are weakly asymmetric (Fig A in S2 Appendix, panels 2–4).

## IFA in other ripple models

In the introduction we motivated IFA as a means of dissociating the present feedback-based model from the *perturbation-based*, inhibition-first model. Indeed, the *perturbation-based* inhibitory model [41] makes different predictions with respect to IFA: In the *perturbation-based* model, ripples can emerge only transiently, when a sudden increase in drive synchronizes the interneurons; furthermore, there is no sparse synchrony but all neurons fire in all ripple cycles until the neurons desynchronize: The network frequency during this transient ripple event reflects the momentary unit firing rate and is thus set directly by the strength of the external drive; a symmetric "up-down" drive thus creates a symmetric instantaneous frequency response that first rises and then decays (no IFA). Since the oscillation power in this *perturbation-based* model decays monotonically over the course of the ripple event, the initial phase of rising frequency even dominates over the subsequent decay ("anti-IFA"). The perturbation-based model thus predicts that IFA only occurs in response to *asymmetric* drive (*e.g.* a sudden step up, followed by a ramp down). Excitatory currents measured during spontaneous SPW-Rs can exhibit some asymmetry due to synaptic filtering [26], but generally have a non-zero rise time (CA3 PV+ BC: [21, 72]; CA1 pyramidal cell: [26, 37]). Alternatively, the *perturbation-based* model could account for IFA under a step-current input, by assuming strong spike frequency adaptation. However, experimental evidence suggests that PV+ interneurons —likely candidates for inhibition-first ripple generation in CA1 [73–78]—do not exhibit strong spike frequency adaptation [62]. This speaks more in favor of the feedback-based inhibitory model, which can account for IFA occuring independently of the exact (a)symmetry of the SPW-associated drive (Fig A in S2 Appendix), and without assuming high interneuron firing rates with strong spike frequency adaptation.

In both excitation-first ripple models, the ripple frequency depends on the average spike propagation delay among pyramidal cells—either orthodromically via supralinear dendrites [29, 79] or antidromically via axo-axonal gap junctions [27, 28]. These models may be able to account for IFA by assuming an increase in the spike propagation delay over the course of a ripple event. Such an increase in latency might occur due to increasing somatic depolarization which has been shown to decrease the action potential amplitude [80, 81]. Future work should investigate in more depth whether and under which conditions excitation-first models can generate IFA.

## A note on mixed excitatory-inhibitory models for ripple generation

Computational models for ripple generation that combine excitatory and inhibitory populations (mixed, excitatory-inhibitory 'E-I' network models) are sometimes listed as a third class of ripple models in the literature [36, 38, 43]. Because CA1 contains both excitatory and inhibitory neurons, any "complete" *in silico* ripple model would need to incorporate both. In the

present work, however, we use the term *model* to distinguish different generation mechanisms of ripple oscillations. To the best of our knowledge, there is no evidence that an E-I network model can produce oscillations in the ripple range in the absence of recurrent inhibitory coupling if biologically realistic parameters of delays and time constants are used. On the contrary, both theoretical [35] and numerical [37, 39] analyses suggest that the presence of a disynaptic E-I loop promotes network frequencies that are well below the ripple range, which is why E-I network models are typically used in the context of gamma oscillations [66, 82, 83]. We thus posit that, mechanistically, all currently known ripple models can be categorized into either inhibition-first or excitation-first (a potential special case is the model by [84] relying on an excitatory effect of GABAergic synaptic transmission). The respective other cell type can have a modulatory influence, and such modulations have been studied (interneurons in excitation-first models: [28, 29]; pyramidal cells in inhibition-first models: [35–41, 43]): Changes in the recruitment of CA1 interneurons (feedforward via CA3, or local via CA1) or in the balance of feedforward drive to CA1 interneurons and pyramidal cells have been suggested to mediate transitions between ripple and gamma network states [37, 39]. Furthermore, there is evidence that ripple phase-locked action potentials of pyramidal cells may increase the ripple power in the local field potential [24].

## Measurement and quantification of ripple oscillations: Population activity or local field potential?

Understanding ripple oscillations requires not only understanding their generating mechanism but also the origin of the signal that is used to detect them. Typically, ripple oscillations are quantified using the local field potential (LFP). The LFP is traditionally believed to reflect synaptic currents [23, 85], but modeling studies have suggested that action potentials of pyramidal cells can also contribute when occurring locked to a ripple rhythm [24, 38]. In fact, the participation of pyramidal cells in the ripple rhythm may be crucial for the detectability of ripple power in the LFP [43].

In the present work, we used the population activity of interneurons as a proxy for the LFP signal. The relation between interneuron population activity and LFP in the feedback-based inhibition-first model was partially explored by [37] using numerical simulations: They found that for bandpass-filtered (120–300 Hz) average inhibitory currents as a simple model of the CA1 LFP, IFA in the inhibitory population activity translates into IFA in the LFP. Simulations of the feedback-based inhibition-first model including pyramidal cells [36–38] have shown that the inhibitory network can pace pyramidal cell spikes to occur phase-locked to the ripple rhythm. Furthermore, IFA can occur in the spiking activity of both excitatory and inhibitiory populations [37]. It may therefore be reasonable to assume that even in a more complex model of the LFP, taking into account both inhibitory synaptic currents and excitatory action potentials, IFA in the inhibitory population activity may translate into IFA in the LFP.

Note that all proposed ripple models focus on explaining ripple generation in the spiking activity of the respective pacemaker population (excitatory or inhibitory). A detailed model of the CA1 LFP and additional numerical simulations will be needed to confirm, for each of the models, whether they yield a ripple signature in the LFP that is consistent with experimental data.

## Conclusion

In conclusion: The feedback-based inhibition-first ripple model can account naturally for IFA without adding further parameter constraints. A deepened understanding of the transient ripple dynamics in each of the proposed models together with extensive experimental testing of

the various predictions will hopefully advance our understanding of the generating mechanism of ripple oscillations and enable us to study their potential role in memory consolidation.

## Methods

The following section on the Methods of this paper is written to be self-contained. Thus, some of the equations shown in the Results will be reprinted here in the Methods for better readability.

### Spiking neural network simulations

**Network architecture.** To model ripples in a spiking neural network, we consider a fully-connected inhibitory network of noisy leaky integrate-and-fire (LIF) neurons. Each neuron's membrane potential $v_i$ (for $i = 1, \ldots, N$) is given by the following stochastic differential equation (SDE):

$$\tau_m \dot{v}_i = -v_i + E_{\text{leak}} + \frac{\tau_m}{C} I_{\text{ext}}(t) - \tau_m \frac{J}{N} \sum_{j=1}^{N} \sum_{k} \delta(t - \Delta - t_j^k) + \sqrt{2\tau_m}\sigma_V \xi_i(t) \; ; \qquad (13)$$

see also Results, Eq (1). Whenever the membrane potential reaches a threshold of $V_{\text{thr}} = -52$ mV, a spike is recorded and the membrane potential is reset to $V_{\text{reset}} = -65$ mV. For simplicity there is no absolute refractory period. We choose a membrane time constant $\tau_m = 10$ ms, resting membrane potential $E_{\text{leak}} = -65$ mV and membrane capacitance $C = 100$ pF as used by [37]. All default parameter values are summarized in Table 2.

In our model, Eq (13), each unit receives a common excitatory drive $I_{\text{ext}}(t)$ and a common inhibitory synaptic input in form of a sum of presynaptic Dirac-delta spikes from all neurons in the network. Spikes occur at times $\{t_j^k\}_{j=1,\ldots,N}^{k=1,2,\ldots}$, where $t_j^k$ denotes the $k$-th spike time of neuron $j$. They are transmitted to all neurons in the network and arrive at the postsynaptic neurons after a synaptic delay of $\Delta = 1.2$ ms. The amplitude of the inhibitory postsynaptic potential elicited by one input spike is determined by the synaptic strength $J = 65$ mV. Our default network has $N = 10, 000$ units.

To account for noisy background activity, every unit receives independent Gaussian white noise $\xi_i(t)$ with $\langle \xi_i(t) \rangle = 0$, $\langle \xi_i(t)\xi_j(t') \rangle = \delta_{i,j}\delta(t - t')$. The strength of the noise is determined by the parameter $\sigma_V = 2.62$ mV, which can be interpreted as the long-time standard deviation of the membrane potential in the absence of a threshold.

In simulations of the spiking network with a finite temporal resolution ($\Delta t = 0.01$ ms), we define the *empirical* population activity as

$$r_N(t) = \frac{n_{\text{spk}}(t, t + \Delta t)}{N\Delta t} \qquad (14)$$

(see also Results, Eq (2)), where

$$n_{\text{spk}}(t, t + \Delta t) = \int_t^{t+\Delta t} \sum_{i=1}^{N} \sum_{k} \delta\left(s - t_i^k\right) \mathrm{d}s \qquad (15)$$

denotes the total number of spikes that were emitted from the population in the time interval $[t, t + \Delta t]$. The empirical population activity has units of spikes per second and can also be interpreted as the instantaneous firing rate of single neurons in our homogeneous network [86].

For plotting purposes the empirical population activity is smoothed with a Gaussian window $g$ of standard deviation $\sigma_t = 0.3$ ms:

$$r_N^{\text{plot}}(t) = (g * r_N)(t)\,, \quad g(t) = \frac{1}{\sqrt{2\pi}\sigma_t}\exp\left[-\frac{t^2}{2\sigma_t^2}\right]\,. \tag{16}$$

To facilitate notation we omit the superscript "plot" in all Figures.

For constant drive, we simulate 5.05 s (time step $\Delta t = 0.01$ ms). The initial membrane potentials are drawn randomly from a uniform distribution between the reset and threshold potentials. The initial 50 ms are excluded from analysis, because we are not interested in the initial transient but in the asymptotic network dynamics. The remaining 5 s are sufficient for a basic spectral analysis.

For time-dependent drive, we first simulate the network for 200 ms with a constant baseline drive of $I_{\text{E}}^{\text{crit}}/2$ (see section *Linear stability analysis*), followed by the time-dependent, *sharp wave-like* stimulus, which we model as a piecewise linear double ramp of slope $\pm m$ with a plateau phase of arbitrary length in between (here 20 ms):

$$I_{\text{ext}}(t) \;=\; \begin{cases} I_{\text{ext}}^{\text{crit}}/2, & 0 \leq t \leq t_1 = 200\,\text{ms} \\[2mm] I_{\text{ext}}^{\text{crit}}/2 + m(t - t_1), & t_1 < t \leq t_2 = t_1 + \frac{I_{\text{ext}}^{\text{full}} - I_{\text{ext}}^{\text{crit}}/2}{m} \\[2mm] I_{\text{ext}}^{\text{full}}, & t_2 < t \leq t_3 = t_2 + 20\,\text{ms} \\[2mm] I_{\text{ext}}^{\text{full}} - m(t - t_3), & t_3 < t \leq t_4 = t_3 + \frac{I_{\text{ext}}^{\text{full}} - I_{\text{ext}}^{\text{crit}}/2}{m} \end{cases} \tag{17}$$

During the plateau phase the drive is at the approximate point of full synchrony $I_{\text{ext}}^{\text{full}}$ for the respective spiking network, as determined by a range of constant-drive simulations.

Simulations of the LIF network model have been performed using the Brian2 simulator [87]. Data storage and parallelization of simulations for large parameter explorations were done using the Python toolkit *pypet* [88].

**Frequency analysis.** The network frequency at constant drive is defined as the location of the dominant peak in the power spectral density of the population activity $r_N(t)$. The saturation (average fraction of neurons firing in one cycle of the population rhythm) is computed by dividing the mean unit firing rate by the network frequency.

To define the instantaneous network frequency in response to time-dependent drive, we use frequency estimates both in continuous time and for a discrete set of time points. The continuous estimate is derived from the wavelet spectrogram (windowed Fourier transform) of the population activity, which indicates instantaneous power in the frequency band from 0 to 350 Hz over time; the instantaneous frequency at each point in time is defined as the frequency above 70 Hz, that has maximal instantaneous power. The lower limit is introduced to exclude the low-frequency contribution due to the sharp wave. The instantaneous frequency is considered significant, whenever the corresponding instantaneous power exceeds a power threshold. The power threshold is chosen as the average instantaneous power at 0 Hz during the initial 200 ms baseline window, plus 4 standard deviations.

An alternative estimate of the instantaneous frequency is given by the inverse of the peak-to-peak distances in the (smoothed) population rate. We consider only peaks that are more than 4 standard deviations above the average rate during baseline stimulation. This procedure delivers a discrete set of frequency estimates associated with a discrete set of time points. In this paper we mostly rely on this "discrete" estimate of instantaneous frequency since its parameter-dependencies (minimal height of oscillation peaks) are more transparent than the ones of the continuous-time estimate (size of time window for windowed

Fourier transform, power threshold). Furthermore it is better suited for comparison with the theory which also describes instantaneous frequency as a discrete measure per cycle (see Eq (65)).

To quantify the network's instantaneous frequency response to a SPW-like drive (Eq (17)), we perform 50 independent simulations of the network model with the same drive but different noise realizations. Linear regression over the discrete instantaneous frequency estimates $(\hat{t}_i, \hat{f}_i)_{i=1,\dots,\ell}$ from $\ell$ discrete oscillation cycles, pooled together from all 50 simulations, yields the average change of the instantaneous frequency over time:

$$f_{\text{net}}^{\text{inst}}(t) \approx \chi_{\text{IFA}} \cdot t + \text{const.} \quad , \qquad \chi_{\text{IFA}} = \frac{\text{Cov}\left(\hat{f}, \hat{t}\right)}{\text{Var}(\hat{t})} \tag{18}$$

A negative slope $\chi_{\text{IFA}} < 0$ indicates IFA.

Note that for our symmetric model of SPW-like drive (Eq (17)), the regression slope $\chi_{\text{IFA}}$ is approximately equal to the slope resulting from linear regression over the *deviations* of the instantaneous network frequencies $\hat{f}_i$ from the symmetric, asymptotic reference frequencies $f_{\text{net}}^\infty(I_{\text{ext}}(\hat{t}_i))$ (except for small effects of asymmetric sampling over time):

$$\chi_{\text{IFA}} = \frac{\text{Cov}\left[\hat{f} - f_{\text{net}}^\infty\left(I_{\text{ext}}(\hat{t})\right), \hat{t}\right]}{\text{Var}(\hat{t})} + \underbrace{\frac{\text{Cov}\left[f_{\text{net}}^\infty\left(I_{\text{ext}}(\hat{t})\right), \hat{t}\right]}{\text{Var}(\hat{t})}}_{\approx 0} \approx \frac{\text{Cov}\left[\hat{f} - f_{\text{net}}^\infty\left(I_{\text{ext}}(\hat{t})\right), \hat{t}\right]}{\text{Var}(\hat{t})} \tag{19}$$

For asymmetric drive, the covariance $\text{Cov}\left[f_{\text{net}}^\infty\left(I_{\text{ext}}(\hat{t})\right), \hat{t}\right]$ needs to be taken into account, when inferring the IFA slope $\chi_{\text{IFA}}$ from our analysis of the asymmetry between instantaneous and asymptotic frequencies.

**The point of full synchrony.** We estimate the point of full synchrony in the spiking network by interpolating the simulated saturation curve and estimating the level of drive for which it becomes 1.

**Extracting the average oscillation cycle.** For constant drive, we split the spiking network simulation into individual cycles based on the Hilbert transform of the mean membrane potential. We take a sufficient number of equally spaced samples from each cycle (here 21) and average them across cycles to derive the average trajectory of the population rate and the membrane potential histogram over the course of a ripple cycle. For each of the 21 sample times we can calculate the average membrane potential, which will be used for comparison with the theory.

## Mean-field approximation

**Dimensionless equations.** To facilitate notation in the theoretical part of this paper, we shift and rescale all voltages, such that the spiking threshold becomes $V_T = 1$ and the resting potential is $E_L = 0$. This corresponds to measuring voltage in units of the distance from the resting potential to the spike threshold. The single unit SDE then reads

$$\tau_m \dot{V}_i = -V_i + I(t) + \sqrt{2D\tau_m}\, \xi_i(t) \tag{20}$$

$$I(t) = I_{\text{E}}(t) - I_{\text{I}}(t) \tag{21}$$

(see also Results Eq (3)), where now

$$V_i = \frac{v_i - E_{\text{leak}}}{V_{\text{thr}} - E_{\text{leak}}} , \qquad\qquad K = \frac{J}{V_{\text{thr}} - E_{\text{leak}}} ,$$

$$I_{\text{E}}(t) = \frac{\tau_m}{C(V_{\text{thr}} - E_{\text{leak}})} I_{\text{ext}}(t) , \qquad\qquad \sqrt{D} = \frac{\sigma_V}{V_{\text{thr}} - E_{\text{leak}}} , \qquad (22)$$

$$I_{\text{I}}(t) = K\tau_m \frac{1}{N} \sum_{i=1}^{N} \sum_{k=1}^{n_j} \delta(t - \Delta - t_j^k) , \qquad\qquad V_R = \frac{V_{\text{reset}} - E_{\text{leak}}}{V_{\text{thr}} - E_{\text{leak}}}$$

are all dimensionless quantities.

**Fokker-Planck equation.** In the mean-field limit of an infinitely large interneuron population ($N \to \infty$) the evolution of the membrane potential density $p(V, t)$ is described by the following Fokker-Planck equation (FPE) (see *e.g.* [34, 51]):

$$\partial_t p(V, t) = -\partial_V \left( \frac{1}{\tau_m} (I(t) - V) p(V, t) \right) + \frac{D}{\tau_m} \partial_V^2 p(V, t) \qquad (23a)$$

$$I(t) = I_{\text{E}}(t) - I_{\text{I}}(t) = I_{\text{E}}(t) - K\tau_m r(t - \Delta) ; \qquad (23b)$$

see also Results Eqs (4)–(5). The FPE can also be written as a continuity equation

$$\partial_t p(V, t) = -\partial_V J(V, t) \qquad (23c)$$

with a probability current

$$J(V, t) = \frac{1}{\tau_m} (I(t) - V) p(V, t) - \frac{D}{\tau_m} \partial_V p(V, t) . \qquad (23d)$$

Since units are reset instantaneously as soon as they reach the spiking threshold, the FPE has an absorbing boundary condition at threshold:

$$p(V_T, t) = 0 . \qquad (23e)$$

The population rate $r$ is given by the probability current through the threshold:

$$r(t) = J(V_T, t) \overset{(23e)}{=} -\frac{D}{\tau_m} \partial_V p(V, t)|_{V_T} ; \qquad (23f)$$

see also Results, Eq (6). The fire-and-reset mechanism introduces a derivative discontinuity at the reset potential $V_R$:

$$\lim_{\epsilon \to 0} [\partial_V p(V_R + \epsilon, t) - \partial_V p(V_R - \epsilon, t)] = -\frac{\tau_m}{D} r(t) , \qquad (23g)$$

see also [51–55]. The membrane potential density must decay to zero fast enough in the limit of $V \to -\infty$:

$$\lim_{V \to -\infty} p(V, t) = \lim_{V \to -\infty} V p(V, t) = 0 . \qquad (23h)$$

As a probability density, $p(V, t)$ obeys the normalization condition:

$$\int_{-\infty}^{V_T} p(V, t) \, \mathrm{d}V = 1 \quad \forall\, t. \qquad (23i)$$

## Derivation of the Gaussian-drift approximation

Without the absorbing boundary at threshold and the source term due to the fire-and-reset rule, the FPE has only natural boundary conditions and can be solved analytically. Its solution $p_\delta$ for an initial Dirac delta distribution $p_\delta(V, 0) = \delta(V - \mu_0)$, is a Gaussian density with time-dependent mean $\mu(t)$ and variance $\sigma^2(t)$ [89]:

$$p_\delta(V, t) = \frac{1}{\sqrt{2\pi}\sigma(t)} \exp\left[-\frac{(V - \mu(t))^2}{2\sigma^2(t)}\right] . \qquad (24)$$

The mean membrane potential $\mu$ evolves according to the single unit ODE (Eq (20)) without the noise term:

$$\dot{\mu}(t) = \frac{1}{\tau_m}(I(t) - \mu(t)), \quad \mu(0) = \mu_0 ; \qquad (25)$$

see also Results (8a). The variance of the membrane potential distribution, $\sigma(t)^2$, approaches $D$ with a time constant of $\tau_m/2$:

$$\sigma^2(t) := D\left(1 - e^{-2t/\tau_m}\right) \xrightarrow{t \to \infty} D . \qquad (26)$$

The solution for an arbitrary initial condition can be found by convolution of the initial condition with $p_\delta$. Hence in the long time limit ($t \to \infty$) *all* solutions of the FPE with natural boundary conditions tend towards a Gaussian with variance $D$—independent of the initial condition (which has to satisfy the boundary conditions Eq (23)).

In our ripple-generating network, with strong drive and strong coupling, the bulk of the membrane potential distribution is strongly *sub*threshold in between population spikes (*i.e.* unaffected by the non-natural boundary conditions), and thus tends towards a Gaussian density. When the next population spike begins, we can hence approximate the membrane potential density as a Gaussian with fixed variance $D$:

$$p(V, t) \approx \frac{1}{\sqrt{2\pi D}} \exp\left[-\frac{(V - \mu(t))^2}{2D}\right] ; \qquad (27)$$

see also Results, Eq (7). This Gaussian approximation allows the derivation of a simple expression for the population rate under strong drive [56–58]:

$$r(t) = J(V_T, t) \overset{(23h)}{=} \int_{-\infty}^{V_T} \partial_V J(V, t) \mathrm{d}V \overset{(23c)}{=} -\partial_t \int_{-\infty}^{V_T} p(V, t) \mathrm{d}V \overset{(27)}{=} \dot{\mu}(t)\, p(V_T, t) . \qquad (28)$$

The rate is given by the value of the Gaussian density at threshold, scaled by the speed at which the mean membrane potential approaches the threshold. Since only *upwards*-threshold-crossings should contribute to the rate, we add a sign-dependence:

$$r(t) = [\dot{\mu}(t)]_+ p(V_T, t) ; \qquad (29)$$

see also Results Eq (8b). The rate is clipped to 0 whenever the mean membrane potential decays: $[\dot{\mu}(t)]_+ := \max(0, \dot{\mu}(t))$ [56–58].

**Numerical analysis of oscillation dynamics.** In its derivation above, we formulated the Gaussian-drift approximation in several equations, describing the membrane potential density $p$ (Eq (27)), mean membrane potential $\mu$ (Eq (25)) and population rate $r$ (Eq (29)) separately. Note however that the mean membrane potential $\mu$ is the only independent variable and thus the Gaussian-drift approximation can be rephrased as a single delay differential equation

(DDE):

$$\dot{\mu}(t) \overset{(25)}{=} \frac{1}{\tau_m}\left(I(t) - \mu(t)\right) \overset{(23b)}{=} \frac{1}{\tau_m}\left(I_{\mathrm{E}} - \tau_m K r(t-\Delta) - \mu(t)\right) \tag{30}$$

$$\overset{(29)}{\underset{(27)}{=}} \frac{1}{\tau_m}\left(I_{\mathrm{E}} - \tau_m K [\dot{\mu}(t-\Delta)]_+ \frac{1}{\sqrt{2\pi D}}\exp\left[-\frac{(V_T - \mu(t-\Delta))^2}{2D}\right] - \mu(t)\right) \tag{31}$$

We will nevertheless illustrate the solutions as two-dimensional trajectories of both $\mu(t)$ and $r(t)$ (Fig 8), since the population rate is our main variable of interest. The DDE can be integrated numerically using a simple forward Euler method. We numerically explored the dynamics for initial conditions $\mu(t \le 0) \ll V_T$. Effectively, we assume that the Gaussian membrane potential distribution is so far subthreshold that no spiking has occured before time 0 ($r(t) \sim p(V_T, t) \approx 0 \,\forall t \le 0$). In that case, the intialization of $\dot{\mu}(t \le 0)$, which only appears as a second factor in the rate expression Eq (29), is irrelevant as long as it is finite. For constant drive $I_{\mathrm{E}}$, we then find a range of potential dynamics (see Fig 8).

There is a large regime of sufficiently strong drive in which the solution $\mu$ exhibits persistent period-1 oscillations (Fig 8, green). This is the regime that we are interested in and the dynamics of which we will approximate in the following.

At lower levels of drive, there are three additional dynamical regimes that we will exclude from analysis: At very low drive, the system has a stable fixed point ($\dot{\mu} = 0$) in $(\mu(t), r(t)) \equiv (I_{\mathrm{E}}, 0)$ (Eq (30), Fig 8, blue). The bifurcation at which the fixed point loses stability can be determined numerically (see Eq (38) for an analytical approximation). Immediately after the bifurcation, the DDE solution exhibits very fast oscillations at $(2\Delta)^{-1} \sim 417$ Hz (Fig 8, red). The oscillation amplitude of the mean membrane potential is very small, and a large portion of the Gaussian potential density is suprathreshold at all times. We refer to this oscillation as "pathological" since it is a direct result of the artificial clipping of the rate to 0 whenever the mean membrane potential decays (Eq (29)). Increasing the drive further brings the system into a state of period-2 oscillations where the Gaussian density gets pushed below threshold only every other cycle (Fig 8, yellow).

These regimes of pathological high-frequency or period-2 oscillations exist due to our simplifying assumptions capturing only the mean-driven aspects of the network dynamics. Both regimes occur either shortly before or after the level of drive $I_{\mathrm{E}}^{\mathrm{crit}}$ at which the original spiking network undergoes a supercritical Hopf bifurcation (see tick mark in Fig 8). In the vicinity of that bifurcation the spiking network dynamics are fluctuation-driven with either no ($I_{\mathrm{E}} < I_{\mathrm{E}}^{\mathrm{crit}}$) or only small-amplitude ($I_{\mathrm{E}} > I_{\mathrm{E}}^{\mathrm{crit}}$) oscillations in the mean membrane potential and population rate. These cannot be captured without taking into account the absorbing boundary at threshold and the non-Gaussian shape of the density of membrane potentials. We will exclude these pathological dynamics from analysis by introducing a lower bound $I_{\mathrm{E}}^{\mathrm{min}}$ for the theoretical approximations that we develop in the following (see tick mark in Fig 8).

## Analytical approximation of oscillation dynamics for constant drive

For strong enough drive the mean membrane potential $\mu$ under the Gaussian-drift approximation oscillates periodically between two local extrema $\mu_{\mathrm{min}}$ and $\mu_{\mathrm{max}}$ (Fig 8, green regime, see also Fig 9A). The population rate $r(t)$ oscillates at the same frequency and is positive when the mean membrane potential increases, *i.e.* $\dot{\mu}(t) \ge 0$, and 0 otherwise, c.f. Eq (29). The time when the mean membrane potential reaches its local maximum $\mu_{\mathrm{max}}$ marks the end of the

population spike and will be denoted as $t_{\text{off}}$. The inhibitory feedback (Eq (23b)) thus ends at time $t_{\text{off}} + \Delta$ and we define $\mu_{\min} := \mu(t_{\text{off}} + \Delta)$ as the end of a cycle. This allows us to approximate the overall period as $T = t_{\text{off}}(\mu_{\min}, \mu_{\max}) + \Delta$. Here, we wrote $t_{\text{off}}(\mu_{\min}, \mu_{\max})$ to emphasize that $t_{\text{off}}$ is the rise time of the mean membrane potential from $\mu_{\min}$ to $\mu_{\max}$ (*upstroke*). Furthermore, $\Delta$ is the duration of the subsequent *downstroke* back to $\mu_{\min}$. In the following, we will approximate $\mu_{\max}$ (Step 1) and $\mu_{\min}$ (Step 2) and thus derive the network frequency as the inverse of the period:

$$f_{\text{net}} = T^{-1} = \left(t_{\text{off}}(\mu_{\min}, \mu_{\max}) + \Delta\right)^{-1} \tag{32}$$

(see also Results, Eq (9)).

**Step 1.** To find the local maximum $\mu_{\max} = \mu(t_{\text{off}})$ we have to solve

$$0 = \dot{\mu}(t_{\text{off}}) \ . \tag{33}$$

Since the dynamics of the mean membrane potential are given by a delay differential equation, the term on the right-hand side is recurrent:

$$\dot{\mu}(t_{\text{off}}) \overset{(25)}{=} \frac{1}{\tau_m}\left(I(t_{\text{off}}) - \mu(t_{\text{off}})\right)$$

From the yet unknown time $t_{\text{off}}$ we have to look back in time (in windows with length of the delay $\Delta$) at the history of the population rate $r(t)$:

$$\overset{(23b)}{=} \frac{1}{\tau_m}\left(I_{\text{E}} - \tau_m K r(t_{\text{off}} - \Delta) - \mu(t_{\text{off}})\right)$$

which in turn depends on $\mu(t)$ and $\dot{\mu}(t)$:

$$\overset{(29)}{=} \frac{1}{\tau_m}\left(I_{\text{E}} - \tau_m K[\dot{\mu}(t_{\text{off}} - \Delta)]_+ \, p(V_T, t_{\text{off}} - \Delta) - \mu(t_{\text{off}})\right)$$

$$\overset{(25)}{\underset{(23b)}{=}} \frac{1}{\tau_m}\left(I_{\text{E}} - K[I_{\text{E}} - \tau_m K r(t_{\text{off}} - 2\Delta) - \mu(t_{\text{off}} - \Delta)]_+ p(V_T, t_{\text{off}} - \Delta) - \mu(t_{\text{off}})\right) \tag{34}$$

$$= \ldots$$

We can resolve the recurrence since there is only a *finite* number of past time windows, during which the population rate, and thus the delayed feedback inhibition $I_{\text{I}}(t) = K\tau_m r(t - \Delta)$, is significantly above 0. In the first time window $[t_{\text{off}} - \Delta, t_{\text{off}}]$, right before the end of the upstroke, we have to take inhibition into account, since this is what stops the upstroke. In the second time window $[t_{\text{off}} - 2\Delta, t_{\text{off}} - \Delta]$, further into the past, we will assume that the inhibitory feedback is negligible, *i.e.*

$$I(t) \approx I_{\text{E}} \quad \forall \, 0 \le t \le t_{\text{off}} - \Delta.$$

This assumption implies that the population rate was negligible in the previous time window, *i.e.*

$$r(t) \approx 0 \quad \forall \, 0 \le t \le t_{\text{off}} - 2\Delta \tag{A1}$$

(Eq (23b), Fig 9A). Note that $t$ here refers to time since the beginning of the cycle ($t = 0$). Since the population spike ends at time $t_{\text{off}}$, (A1) is equivalent to the assumption that the population spike lasts at most $2\Delta$. Adding the subsequent downstroke time of $\Delta$ this amounts to an upper bound for the oscillation period of around $3\Delta$, plus any additional upstroke time with $r \approx 0$, which is a reasonable assumption for a feedback loop with delay $\Delta$ as argued already by [34].

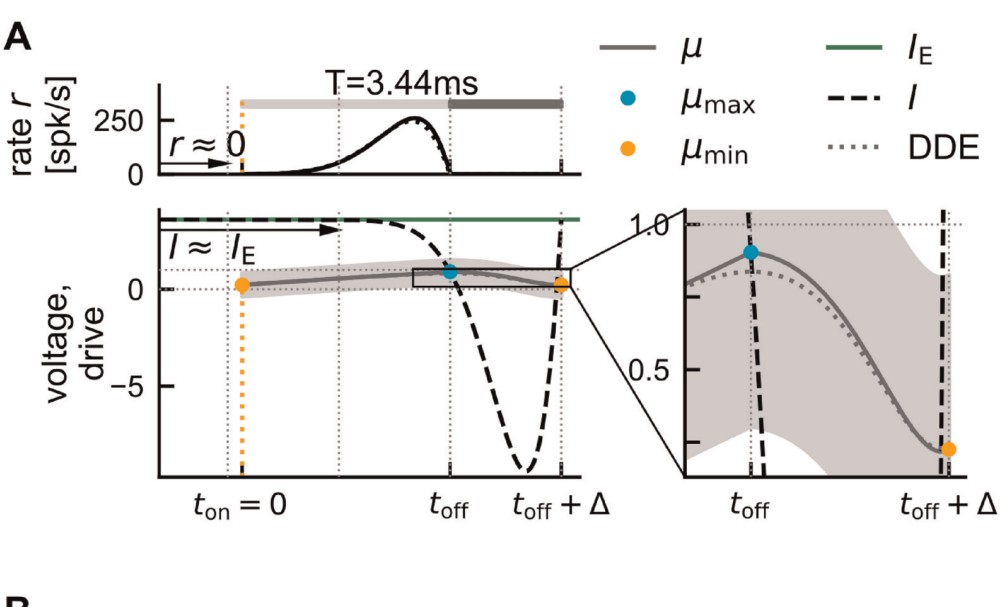

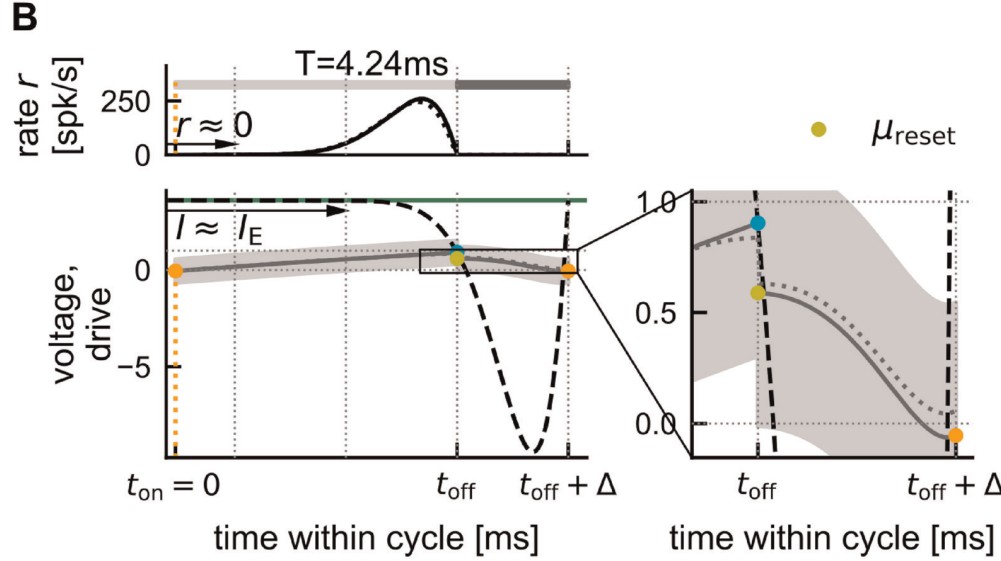

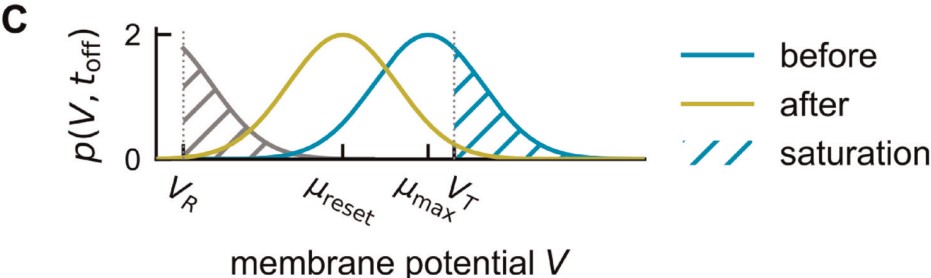

**Fig 9. Illustration of analytical approximation of DDE dynamics.** (A) One cycle of the oscillatory solution of the DDE for $I_E = 3.6$. Dotted lines for rate (top) and mean membrane potential (bottom) are the result of a numerical integration of the DDE (Eq (31)). All other lines illustrate our analytical considerations. Top: population rate $r(t)$ (black). Bottom: mean membrane potential $\mu(t)$ (full grey line: Eq (A2) during upstroke, Eq (39) during downstroke, gray area: $\mu(t) \pm 3\sqrt{D}$ indicating the width of the Gaussian density $p(V, t)$). Constant external drive $I_E$ (green line), total input $I(t) = I_E - I_I(t)$ (dashed line, Eq (42)). Local extrema of the mean membrane potential occur at the intersections of $\mu$ and $I$: Cyan: local maximum $\mu_{max}$ (Eq (38)). Orange: approximate local minimum $\mu_{min}$ (Eq (43)). Vertical dotted lines mark end of population spike $t_{off}$ as well as intervals $t_{off} + k\Delta$, $k \in \{-2, -1, 1\}$. Arrows illustrate

simplifying assumption (A1). The beginning of the cycle ($t_{on}$ = 0) is determined by $\mu_{min}$. Horizontal gray bars mark the length of one cycle (here $T = t_{off} + \Delta = 3.44$ ms (Eq (47)), corresponding to a network frequency of $f_{net}$ = 290.7 Hz, (Eq (32))). Inset: magnification highlighting the differences between numerical solution (dotted) and analytical approximation (full line). Due to assumption (A2) $\mu_{max}$ is slightly overestimated. Note that the second intersection of $\mu$ and $I$ occurs shortly *before* time $t_{off} + \Delta$. Hence $\mu_{min}$ is slightly larger than the true local minimum. (B) Same as A, but with an account for the reset on the population level. At the end of the population spike, $\mu$ is reset instantaneously from $\mu_{max}$ to $\mu_{reset}$ (Eq (49)) (yellow marker). This leads to a lower $\mu_{min}$ (Eq (50)) and hence a slightly longer period ($T$ = 4.24 ms), *i.e.* lower network frequency ($f_{net}$ = 235.8 Hz). (C) Illustration of phenomenological reset. Cyan: density of membrane potentials $p(V, t_{off})$ at the end of the population spike, centered at $\mu_{max}$ (before reset). Cyan hatched area: fraction of active units (*saturation*, Eq (48)). Grey hatched area: resetting the active portion of $p$. Yellow: $p(V, t_{off})$ after reset, centered at $\mu_{reset}$. The reset value $\mu_{reset}$ is calculated as the average of the density that results from summing the grey-dashed (active units) and cyan-non-hatched (silent units) density portions (Eq (49)). Default parameters (see Table 2).

Under this assumption, we set $r(t_{off} - 2\Delta) = 0$ in Eq (34), and only a finite number of terms remains:

$$0 = \dot{\mu}(t_{off}) \stackrel{(A1)}{\approx} \frac{1}{\tau_m}\left[I_E - \mu(t_{off}) - K[I_E - \mu(t_{off} - \Delta)]p(V_T, t_{off} - \Delta)\right]. \qquad (58) \qquad (35)$$

Note that the rectification $[\cdot]_+$ has been dropped in this equation because the mean membrane potential can never be larger than the external drive. Furthermore, we assume that we can approximate the past $\mu(t_{off} - \Delta)$ during the upstroke by only considering the excitatory drive, $I(t) \approx I_E$. Under this assumption, Eq (25) yields the exponential relaxation

$$\mu(t) \approx I_E - (I_E - \mu_{min})e^{-t/\tau_m} \qquad \forall\, 0 \le t \le t_{off} \qquad (36)$$

Looking backwards from time $t_{off}$, we can reformulate this assumption as:

$$\mu(t_{off} - x) \approx I_E - (I_E - \mu_{max})e^{x/\tau_m} \qquad \forall\, x \in [0, t_{off}] \qquad (A2)$$

The resulting error is small since the exponential relaxation of the mean membrane potential towards the total drive $I(t)$ is governed by the membrane time constant ($\tau_m$ = 10 ms). The population spike on the other hand is quite synchronized (we assumed that it lasts less than $2\Delta \ll \tau_m$, (A1)). The time window right before $t_{off}$, during which the units receive inhibitory feedback and the total drive deviates from $I_E$, is thus small compared to the membrane time constant and alters the trajectory of $\mu$ only slightly (Fig 9A, inset).

Eq (35) thus simplifies to

$$\begin{aligned} 0 \quad &\approx \frac{1}{\tau_m}\left(I_E - \mu_{max} - K(I_E - \mu_{max})e^{\Delta/\tau_m}p(V_T, t_{off} - \Delta)\right) \\ \Leftrightarrow \quad 0 \quad &\stackrel{(27)}{\approx} 1 - Ke^{\Delta/\tau_m}\frac{1}{\sqrt{2\pi D}}\exp\left[-\frac{(V_T - I_E - (\mu_{max} - I_E)e^{\Delta/\tau_m})^2}{2D}\right] \end{aligned} \qquad (37)$$

The simplified equation can be readily solved for $\mu_{max}$:

$$\mu_{max} \approx I_E - e^{-\Delta/\tau_m}\left[I_E - V_T + \sqrt{2D\ln\left[\frac{K}{\sqrt{2\pi D}}e^{\Delta/\tau_m}\right]}\right]. \qquad (38)$$

To ensure that the argument of the logarithm in Eq (38) is larger than 1, the coupling must be sufficiently strong: $K \ge \sqrt{2\pi D}e^{-\Delta/\tau_m}$. For weaker coupling the inhibitory feedback is never strong enough to counteract the external drive, hence no local maximum $\mu_{max}$ exists. The existence of a lower bound on synaptic strength for the emergence of oscillations is in line with previous analyses [34]. Furthermore, there is also a lower bound for the drive $I_E$: if the drive is

too weak, i.e. $I_E < V_T - \sqrt{2D \ln\left[\frac{K}{\sqrt{2\pi D}} e^{\Delta/\tau_m}\right]}$, the term inside the square brackets in Eq (38) is

negative and we find $\mu_{\max} > I_E$. But then $\mu_{\max}$ cannot be reached; instead, the mean membrane potential simply settles into its fixed point $I_E$ (Fig 8, blue dynamics). Eq (38) thus yields an analytical approximation of the bifurcation point where the DDE fixed point loses stability and oscillations emerge (numerical, Fig 8, blue to red: $I_E \sim 0.61$; analytical, Eq (38): $I_E \sim 0.56$, for default parameters, Table 2). For strong enough coupling and a strong enough external drive a well-defined local maximum $\mu_{\max} < I_E$ exists. Note that $\mu_{\max}$ does *not* depend on the initial voltage $\mu_{\min}$. This makes sense intuitively for any $\mu_{\min}$ that is sufficiently smaller than $\mu_{\max}$, since the "turning point" of the mean membrane potential depends only on the feedback inhibition resulting from the immediate history shortly before $\mu_{\max}$ is reached ($t \in [t_{off} - 2\Delta, t_{off}]$).

**Step 2.** We will now approximate the trajectory of the mean membrane potential during its inhibition-induced downstroke and infer $\mu_{\min} = \mu(t_{off} + \Delta)$, which corresponds to both the end and the starting value of each cycle of a periodic solution.

Note that while $\mu_{\min}$ is close to the periodic local minimum of the mean membrane potential, it is a slight overestimation: The local extrema of the mean membrane potential $\mu$ occur at its intersections with the total drive $I(t)$ (see Eq (33), Fig 9A). At time $t_{off}$ the mean membrane potential reaches its local maximum and becomes larger than the total drive $I(t)$ (Fig 9A). Since the population spike ends at time $t_{off}$, the delayed inhibitory feedback $I_I(t)$ will stop at time $t_{off} + \Delta$: The total drive at this point will equal the external drive ($I(t_{off} + \Delta) = I_E$); note that the mean membrane potential $\mu$ can never be larger than the external drive $I_E$. Hence, if $\mu$ becomes larger than $I$ at time $t_{off}$ and is smaller than $I$ at time $t_{off} + \Delta$, $\mu$ must intersect with $I(t)$ and reach its local minimum slightly *before* time $t_{off} + \Delta$ (see Fig 9A, inset). What we define as the initial/final membrane potential of a cycle ($\mu_{\min} := \mu(t_{off} + \Delta)$) is thus close to but slightly larger than the periodic local minimum. This does not affect our estimate of the period.

The definition of a fixed downstroke duration $\Delta$ allows us to find $\mu_{\min}$ directly by integrating the mean membrane potential ODE Eq (25) up to time $t_{off} + \Delta$, starting from the initial value $\mu(t_{off}) = \mu_{\max}$:

$$\mu(t) = \mu_{\max}\, e^{-(t - t_{off})/\tau_m} + \frac{1}{\tau_m} \int_{t_{off}}^{t} I(s) e^{-(t-s)/\tau_m}\, ds \quad \text{for} \quad t \geq t_{off}. \tag{39}$$

It follows for $\mu_{\min} = \mu(t_{off} + \Delta)$ with initial condition $\mu(t_{off}) = \mu_{\max}$:

$$\mu_{\min} = \mu_{\max} e^{-\Delta/\tau_m} + \frac{1}{\tau_m} \int_{0}^{\Delta} I(t_{off} + \tau) e^{-(\Delta - \tau)/\tau_m}\, d\tau \tag{40}$$

Again, the total current $I(t_{off} + \tau)$ with $\tau \in [0, \Delta]$, has a recurrent dependency on the past dynamics, which can be seen by repeated application of Eqs (23b), (29), (25):

$$
\begin{aligned}
I(t_{off} + \tau) \;=\; I_E - K\bigg[ & I_E \\
& - K\Big[ I(t_{off} - (2\Delta - \tau)) - \mu(t_{off} - (2\Delta - \tau)) \Big] p(V_T, t_{off} - (2\Delta - \tau)) \\
& - \mu(t_{off} - (\Delta - \tau)) \bigg] p(V_T, t_{off} - (\Delta - \tau))
\end{aligned}
$$

As before we assume that the current $I$ at time $t_{off} - (2\Delta - \tau) \leq t_{off} - \Delta$ is given exclusively by

the excitatory drive (A1), which truncates the infinite recurrent expression above to:

$$I(t_{\text{off}} + \tau) \overset{(A1)}{\approx} I_{\text{E}} - K[I_{\text{E}} - K[I_{\text{E}} - \mu(t_{\text{off}} - (2\Delta - \tau))]p(V_T, t_{\text{off}} - (2\Delta - \tau)) \\ - \mu(t_{\text{off}} - (\Delta - \tau))]p(V_T, t_{\text{off}} - (\Delta - \tau)) \tag{41}$$

We are left with a finite number of terms that depend on the trajectory of $\mu(t)$ during the upstroke. Again, we approximate $\mu(t_{\text{off}} - x)$, $x \geq 0$ assuming exponential relaxation towards only the external drive, *i.e.* ignoring inhibition (A2). This abolishes all dependencies on the yet unknown time $t_{\text{off}}$ of the end of the population spike:

$$I(t_{\text{off}} + \tau) \overset{(A2)}{\approx} I_{\text{E}} - K(I_{\text{E}} - \mu_{\max})e^{(\Delta - \tau)/\tau_m}p(V_T, t_{\text{off}} - (\Delta - \tau)) \\ + K^2(I_{\text{E}} - \mu_{\max})e^{(2\Delta - \tau)/\tau_m}p(V_T, t_{\text{off}} - (2\Delta - \tau))p(V_T, t_{\text{off}} - (\Delta - \tau)) \tag{42}$$

Inserting this approximation (Eq (42)) into the expression for $\mu_{\min}$ (Eq (40)) yields:

$$\mu_{\min} \approx \mu_{\max}\,e^{-\Delta/\tau_m} + \frac{1}{\tau_m}e^{-\Delta/\tau_m}I_{\text{E}}\int_0^\Delta e^{\tau/\tau_m}\,d\tau \\ - \frac{1}{\tau_m}K(I_{\text{E}} - \mu_{\max})\int_0^\Delta p(V_T, t_{\text{off}} - (\Delta - \tau))\,d\tau \\ + \frac{1}{\tau_m}K^2(I_{\text{E}} - \mu_{\max})e^{\Delta/\tau_m}\int_0^\Delta p(V_T, t_{\text{off}} - (2\Delta - \tau))p(V_T, t_{\text{off}} - (\Delta - \tau))\,d\tau$$

The integrals can be solved analytically, if we approximate the past trajectory of $\mu$ linearly within the Gaussian expressions $p$:

$$\mu(t_{\text{off}} - x) \overset{(A2)}{\approx} \mu_{\max} - (I_{\text{E}} - \mu_{\max})\frac{x}{\tau_m}, \qquad 0 \leq x \leq 2\Delta \ll \tau_m \tag{A3}$$

Hence,

$$p(V_T, t_{\text{off}} - (k\Delta - \tau)) \approx \frac{1}{\sqrt{2\pi D}}\exp\left[-\frac{\left(V_T - \mu_{\max} + (I_{\text{E}} - \mu_{\max})\frac{k\Delta - \tau}{\tau_m}\right)^2}{2D}\right].$$

Using this expression, we obtain

$$\mu_{\min} \approx \underbrace{\mu_{\max}e^{-\Delta/\tau_m}}_{\text{initial condition}} + \underbrace{I_{\text{E}}\left(1 - e^{-\frac{\Delta}{\tau_m}}\right)}_{\text{excitatory drive}} \\ \underbrace{-\frac{1}{2}K\left[\left[\text{erf}(\phi(0)) - \text{erf}(\phi(\Delta))\right] - K\frac{1}{\sqrt{2\pi D}}e^{-\frac{\epsilon}{2D}}\frac{e^{\frac{\Delta}{\tau_m}}}{\sqrt{e^{\frac{2\Delta}{\tau_m}} + 1}}\left[\text{erf}(\psi(\Delta)) - \text{erf}(\psi(0))\right]\right]}_{\text{inhibitory feedback}}, \tag{43}$$

where

$$\phi(t) = \frac{V_T - \mu_{\max} + (I_E - \mu_{\max})\frac{\Delta - t}{\tau_m}}{\sqrt{2D}} \tag{44}$$

$$\psi(t) = \frac{-(I_E - \mu_{\max})\left(e^{\frac{2\Delta}{\tau_m}} + 1\right)(\tau_m + \Delta - t) + \tau_m(I_E - V_T)\left(e^{\frac{\Delta}{\tau_m}} + 1\right)}{\sqrt{2D\left(e^{\frac{2\Delta}{\tau_m}} + 1\right)}\tau_m} \tag{45}$$

$$c = \frac{(V_T - I_E)^2\left(1 - e^{\frac{\Delta}{\tau_m}}\right)^2}{e^{\frac{2\Delta}{\tau_m}} + 1}. \tag{46}$$

Although lengthy, this expression can be easily evaluated numerically.

The local "minimum" $\mu_{\min}$ is not only the final but also the initial mean membrane potential of each cycle. The length of the upstroke can thus be approximated as the time it takes for the mean membrane potential to rise from $\mu_{\min}$ to $\mu_{\max}$, based on exponential relaxation towards only the excitatory drive $I_E$ (A2):

$$t_{\text{off}} \stackrel{(A2)}{\approx} \tau_m \ln\left(\frac{I_E - \mu_{\min}}{I_E - \mu_{\max}}\right) \tag{47}$$

Together with the assumed downstroke duration of $\Delta$ we arrive at an analytical estimate for the oscillation period $T$ and hence the network frequency $f_{\text{net}}$ (see Eq (32)). Overall we have derived analytical expressions for $\mu_{\max}$, $\mu_{\min}$, $t_{\text{off}}$, $f_{\text{net}}$ as functions of the external drive $I_E$, which characterize the oscillatory dynamics. To evaluate the accuracy of our analytical approximation we integrate the DDE numerically (Eq (31), Fig 8, green regime of period-1 oscillations) and determine $\mu_{\max}$, $\mu_{\min}$ and $f_{\text{net}}$. We find that the errors introduced by our simplifying assumptions (A1)–(A3) are small (Fig 10, dashed lines vs square markers).

**Accounting for the reset mechanism.** The quantitative accuracy of our Gaussian-drift approximation with respect to the original spiking network can be increased by adding a phenomenological account for the reset on the population level. During the population spike ($t < t_{\text{off}}$) the single unit reset has little influence on the population rate dynamics, since units spike at most once per cycle. At time $t_{\text{off}}$ the population spike ends and the integral over the suprathreshold portion of the Gaussian density of membrane potentials corresponds to the fraction of units that have spiked (the *saturation*):

$$s := \int_{V_T}^{\infty} p(v, t_{\text{off}})\,\mathrm{d}v \stackrel{(27)}{=} \frac{1}{2}\left(1 - \mathrm{erf}\left(\frac{V_T - \mu_{\max}}{\sqrt{2D}}\right)\right) \tag{48}$$

(see Fig 9C, cyan hatched area). Taking into account the reset mechanism at this point would mean shifting the suprathreshold portion of $p(V, t_{\text{off}})$ downwards by an amount $V_T - V_R$ (Fig 9C, gray hatched area), essentially splitting the voltage distribution into two pieces, corresponding to silent units (Fig 9C, non-hatched area under cyan Gauss) and units that have spiked and been reset (Fig 9C, gray hatched area). To preserve our simplified framework of a unimodal, Gaussian voltage distribution, we will instead assume that the Gaussian voltage distribution is reset as a whole, to a new mean membrane potential $\mu_{\text{reset}}$ given by the average of

the two distribution pieces ("silent" and "spike+reset"):

$$
\begin{aligned}
\mu_{\text{reset}} &:= \int_{-\infty}^{V_T} \nu p(\nu, t_{\text{off}}) \mathrm{d}\nu + \int_{V_T}^{\infty} [\nu - (V_T - V_R)] p(\nu, t_{\text{off}}) \mathrm{d}\nu \\
&= \int_{-\infty}^{\infty} \nu p(\nu, t_{\text{off}}) \mathrm{d}\nu - (V_T - V_R) \int_{V_T}^{\infty} p(\nu, t_{\text{off}}) \mathrm{d}\nu \\
&= \mu_{\text{max}} - (V_T - V_R) s
\end{aligned}
\tag{49}
$$

This phenomenological account for the reset contains the implicit assumption that in between population spikes the membrane potential distribution "spends enough time" subthreshold that the bimodality created by the reset mechanism vanishes due to diffusion and the distribution becomes roughly Gaussian again. This assumption is satisfied for a relatively large portion of the parameter space spanned by noise intensity, coupling strength, and reset potential (see S1 Appendix). The introduction of the population-reset requires an adjustment of the definition of $\mu_{\text{min}}$ (Eq (40)): Instead of using $\mu_{\text{max}}$ as the initial value when integrating the feedback inhibition during the downstroke, we will now use the reset potential: $\mu(t_{\text{off}}^+) = \mu_{\text{reset}}$. Therefore, the membrane potential $\mu_{\text{min}} = \mu(t_{\text{off}} + \Delta)$ at the end of the cycle is given by

$$
\mu_{\text{min}} = \underbrace{\mu_{\text{reset}} e^{-\Delta/\tau_m}}_{\text{initial condition}} + \underbrace{I_{\text{E}} \left(1 - e^{-\frac{\Delta}{\tau_m}}\right)}_{\text{excitatory drive}}
$$

$$
\underbrace{- \frac{1}{2} K \left[ \left[ \text{erf}(\phi(0)) - \text{erf}(\phi(\Delta)) \right] - K \frac{1}{\sqrt{2\pi D}} e^{-\frac{c}{2D}} \frac{e^{\frac{\Delta}{\tau_m}}}{\sqrt{e^{\frac{2\Delta}{\tau_m}} + 1}} \left[ \text{erf}(\psi(\Delta)) - \text{erf}(\psi(0)) \right] \right]}_{\text{inhibitory feedback}}.
\tag{50}
$$

Except for the initial condition term, all other terms remain unchanged (cf. Eq (43)). Since $\mu_{\text{reset}} < \mu_{\text{max}}$ (Eq (49)), the introduction of the reset *decreases* our estimate of the local minimum $\mu_{\text{min}}$. This leads to an *increase* of the upstroke time $t_{\text{off}}$ required for the mean membrane potential to rise from $\mu_{\text{min}}$ to $\mu_{\text{max}}$ (Eq (47)), and hence to a *decrease* in the network frequency (Eq (32), see Fig 10).

Note that when the reset is incorporated in the *numerical* integration of the DDE (31) (e.g. Fig 9B, dotted lines), the population rate after time $t_{\text{off}}$ needs an additional artificial clipping to zero, since the slope $\dot{\mu}(t)$ will briefly become positive again after the reset $(\dot{\mu}(t_{\text{off}}^+) \sim I(t_{\text{off}}) - \mu_{\text{reset}} > I(t_{\text{off}}) - \mu_{\text{max}} = 0)$.

**The point of full synchrony.** Our analytical ansatz allows for a straightforward prediction of the point of full synchrony and its parameter dependencies. As mentioned before, the integral over the suprathreshold-portion of the membrane potential density at the end of the population spike corresponds to the fraction $s$ of active units (saturation, Eq (48)). In the strict sense of

$$
s = \int_{V_T}^{\infty} p(\nu, t_{\text{off}}) \mathrm{d}\nu = 1
$$

full synchrony can never be reached, since the Gaussian probability density $p$ approaches zero only in the limit $\nu \to \pm\infty$. We can however define *approximate* full synchrony as the state where only the 0.13th percentile of the distribution remains subthreshold:

$$
s = \int_{V_T}^{\infty} p(\nu, t_{\text{off}}) \mathrm{d}\nu = 0.9987 \quad \Leftrightarrow \quad \mu_{\text{max}} - 3\sqrt{D} \geq V_T
$$

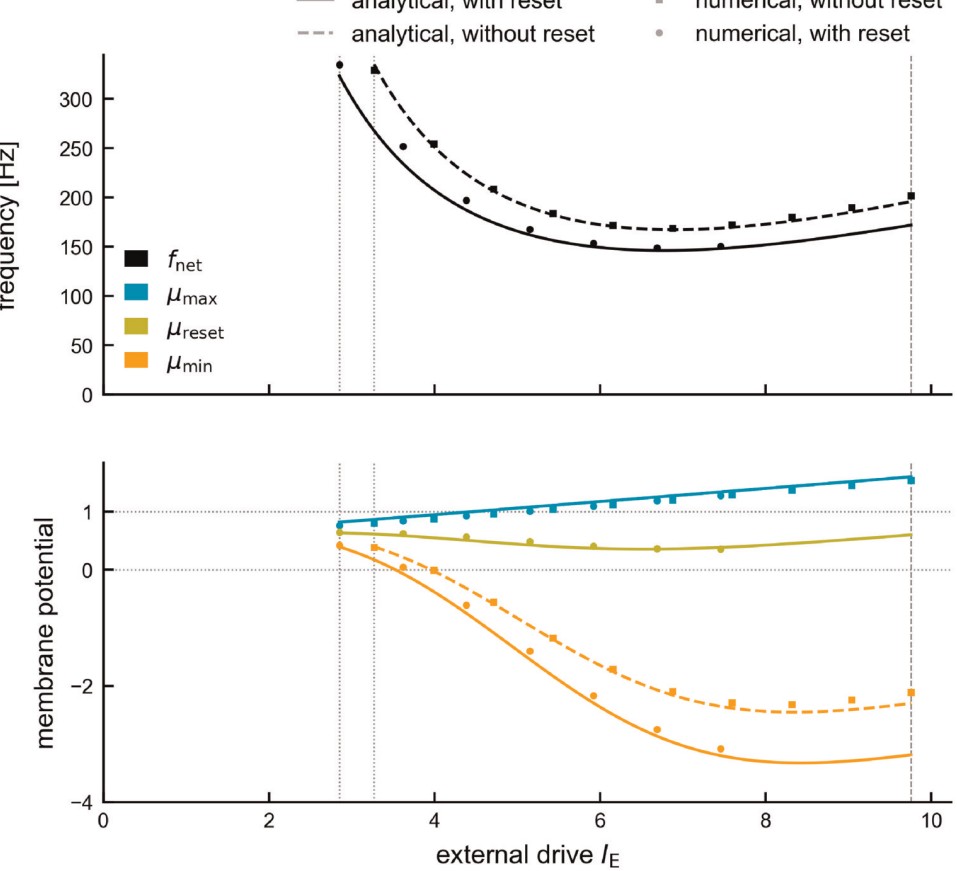

**Fig 10. Analytical vs numerical evaluation of oscillatory solutions in the Gaussian-drift approximation.** Network frequency (top) and dynamics of the mean membrane potential (bottom) quantified in terms of its periodic local minimum $\mu_{\min}$ (orange) and local maximum $\mu_{\max}$ (cyan) for different levels of external drive $I_E$. The analytical approximations (solid lines: with reset, dashed lines: without reset) are very close to the results of numerical integration of the DDE Eq (31) (round markers: with reset, square markers: without reset). Including the reset does not affect $\mu_{\max}$ but leads to a decrease in $\mu_{\min}$ (Eqs (50) vs (43)) and thus a decrease in network frequency. Results are shown in the relevant range of external drives $[I_E^{\min}, I_E^{\text{full}}]$ (vertical dotted lines). For parameters see Table 2.

Using our mapping from external drive to $\mu_{\max}$ (Eq (38)), we can derive a closed-form expression for the external drive that is required to achieve full synchrony:

$$I_E^{\text{full}} = V_T + \sqrt{D}\,\frac{3 + e^{-\Delta/\tau_m}\sqrt{2\ln\left[\dfrac{K}{\sqrt{2\pi D}}e^{\Delta/\tau_m}\right]}}{1 - e^{-\Delta/\tau_m}} \tag{51}$$

**Range of applicability of the Gaussian-drift approximation.** There are two main constraints on the applicability of our theory:

(a) Since we assume that units spike at most once per cycle, the theory is only valid up to the point of full synchrony $I_E^{\text{full}}$ where network frequency and mean unit firing rate coincide.

(b) The assumption of a unimodal distribution of membrane potentials is only valid if, in between population spikes, the bulk of the membrane potential distribution is pushed sufficiently below threshold such that it can diffuse back to approximately Gaussian shape. We will

thus require that at its lowest point the Gaussian density is almost entirely subthreshold:

$$\int_{-\infty}^{V_T} p(v, t_{\text{on}})dv = 0.9987 \quad \Leftrightarrow \quad \mu_{\min}(I_E) + 3\sqrt{D} \overset{!}{\leq} V_T \tag{52}$$

Criteria (a) and (b) yield a finite range $[I_E^{\min}, I_E^{\max}]$ of external drives for which the theory applies. Since for most parameter settings $\mu_{\min}$ is an almost monotonically decaying function of the drive (see Fig 10 and Results), constraint (b) is usually only relevant for the lower boundary $I_E^{\min}$ of the drive, while the upper boundary is determined by constraint (a): $I_E^{\max} = I_E^{\text{full}}$ (Fig 10). For extreme combinations of high noise and weak coupling, however, $\mu_{\min}$ rises again for high drive, so constraint (b) also sets the upper boundary $I_E^{\max} < I_E^{\text{full}}$ (see Fig Ad in S1 Appendix, bottom right panel, corresponding to the three crossed-out parameter settings in Fig B in S1 Appendix: no valid theoretical estimate of the network frequency at full synchrony, since $I_E^{\max} < I_E^{\text{full}}$).

**Quantifying performance.** We quantify the performance of our Gaussian-drift approximation across the range of drives for which

(a). the theory applies ($I_E \in [I_E^{\min}, I_E^{\max}]$)

(b). the spiking network has not crossed the point of full synchrony ($I_E \leq I_E^{\text{full}}$, sim)

The size of this regime varies for different parameter settings. To ensure comparability we interpolate the results of all spiking network simulations to the same fine resolution:

$$I_E^i = I_E^{\min} + 0.1i \quad , \qquad I_E^i \in \left[ I_E^{\min}, \min\left( I_E^{\max}, I_E^{\text{full,sim}} \right) \right] .$$

We then compute the average relative error of the estimated network frequencies for each parameter setting:

$$\chi_{\text{err}} := \frac{1}{n} \sum_i \frac{\left| f_{\text{net}}^{\text{sim}}(I_E^i) - f_{\text{net}}^{\text{theory}}(I_E^i) \right|}{f_{\text{net}}^{\text{sim}}(I_E^i)} \quad \in [0, 1] \tag{53}$$

We introduce a second score to quantify what portion of the relevant range of spiking network dynamics (from the Hopf bifurcation, $I_E^{\text{crit,sim}}$, to the point of full synchrony, $I_E^{\text{full,sim}}$) is covered by the theory:

$$\chi_{\text{appl}} := \frac{\min\left( I_E^{\max}, I_E^{\text{full,sim}} \right) - I_E^{\min}}{I_E^{\text{full,sim}} - I_E^{\text{crit,sim}}} \quad \in [0, 1] \tag{54}$$

We define an overall performance index as

$$\chi_{\text{p}} := \chi_{\text{appl}}(1 - \chi_{\text{err}}) \in [0, 1] \tag{55}$$

The larger the performance index $\chi_{\text{p}}$ the better the Gaussian-drift approximation captures the spiking network dynamics. A systematic evaluation of the performance of the Gaussian-drift approximation for different network parameters and levels of drive can be found in S1 Appendix.

### Analytical approximation of oscillation dynamics for linear drive

We want to characterize the transient dynamics of any cycle $i$ with initial mean membrane potential $\mu^i_{\min}$ and linear drive $I_E(t) = \hat{I}^i_E + m(t - t^i_{\text{off}})$ by deriving functions

$$\left(\hat{I}^i_E, \mu^i_{\min}, m\right) \quad \mapsto \left(f^{\text{inst},i}_{\text{net}}, f^i_{\text{unit}}, t^i_{\text{off}}, \mu^i_{\max}, \mu^i_{\text{reset}}, \mu^{i+1}_{\min}\right) \ . \tag{56}$$

Here $\mu^{i+1}_{\min}$ refers to the mean membrane potential reached at the end of cycle $i$, which potentially serves as the initial condition for the next cycle.

**Step 1.** The local maximum of the mean membrane potential $\mu^i_{\max}$ can be found with the same ansatz that was used for constant drive:

$$\begin{aligned}
0 = \dot{\mu}(t^i_{\text{off}}) \quad &= \frac{1}{\tau_m}\left(\hat{I}^i_E - K\tau_m r(t^i_{\text{off}} - \Delta) - \mu^i_{\max}\right) \\
&\overset{(29)}{=} \frac{1}{\tau_m}\left[\hat{I}^i_E - \mu^i_{\max} - K\tau_m[\dot{\mu}(t^i_{\text{off}} - \Delta)]_+ p(V_T, t^i_{\text{off}} - \Delta)\right] \\
&\overset{(25)}{=} \frac{1}{\tau_m}\bigg[\hat{I}^i_E - \mu^i_{\max} \\
&\quad - K[I_E(t^i_{\text{off}} - \Delta) - K\tau_m \underbrace{r(t^i_{\text{off}} - 2\Delta)}_{\approx 0,(A1)} - \mu(t^i_{\text{off}} - \Delta)]_+ p(V_T, t^i_{\text{off}} - \Delta)\bigg]
\end{aligned}$$

Again we truncate the recurrent expression for the total current two $\Delta$-time windows before the end of the population spike (A1):

$$\overset{(A1)}{\approx} \frac{1}{\tau_m}\left[\hat{I}^i_E - \mu^i_{\max} - K\left[\hat{I}^i_E - m\Delta - \mu(t^i_{\text{off}} - \Delta)\right]p(V_T, t^i_{\text{off}} - \Delta)\right] \ . \tag{57}$$

Again we approximate the trajectory of $\mu$ during the upstroke based on relaxation towards only the excitatory drive, which is now a linear function of time (cf. (A2)). Using the beginning of cycle $i$ as the time origin $t = 0$, we obtain

$$\mu(t) \quad \overset{(25)}{\approx} \mu^i_{\min}e^{-t/\tau_m} + \frac{1}{\tau_m}\int_0^t e^{-(t-\bar{t})/\tau_m}I_E(\bar{t})\bar{t} = I_E(t) - m\tau_m + \left(m\tau_m + \mu^i_{\min} - I_E(0)\right)e^{-t/\tau_m} \tag{58}$$

Under this approximation $\mu^i_{\max}$ can be written as

$$\mu^i_{\max} = \mu(t^i_{\text{off}}) \overset{(58)}{\approx} \hat{I}^i_E - m\tau_m + (m\tau_m + \mu^i_{\min} - I_E(0))e^{-t^i_{\text{off}}/\tau_m} \tag{59}$$

and the trajectory before time $t^i_{\text{off}}$ ($x \geq 0$) can be approximated as:

$$\begin{aligned}
\mu(t^i_{\text{off}} - x) \quad &\overset{(58)}{=} I_E(t^i_{\text{off}} - x) - m\tau_m + \left(m\tau_m + \mu^i_{\min} - I_E(0)\right)e^{-(t^i_{\text{off}} - x)/\tau_m} \\
&\overset{(59)}{=} \left(\mu^i_{\max} - \hat{I}^i_E + m\tau_m\right)e^{x/\tau_m} + \underbrace{I_E(t^i_{\text{off}} - x)}_{=\hat{I}^i_E - mx} - m\tau_m \\
&= \hat{I}^i_E + \left(\mu^i_{\max} - \hat{I}^i_E\right)e^{x/\tau_m} + m\left(\tau_m e^{x/\tau_m} - (x + \tau_m)\right)
\end{aligned} \tag{60}$$

Inserting the above expression in the local maximum condition (Eq (57)) yields:

$$
\begin{aligned}
0 \quad &\approx \hat{I}_{\mathrm{E}}^i - \mu_{\max}^i - \frac{K\left[(\hat{I}_{\mathrm{E}}^i - \mu_{\max}^i)e^{\Delta/\tau_m} - m\tau_m\left(e^{\Delta/\tau_m} - 1\right)\right]}{\sqrt{2\pi D}} \\
&\cdot \exp\left[-\frac{\left(V_T - \hat{I}_{\mathrm{E}}^i - \left(\mu_{\max}^i - \hat{I}_{\mathrm{E}}^i\right)e^{\Delta/\tau_m} - m\left(\tau_m e^{\Delta/\tau_m} - (\Delta + \tau_m)\right)\right)^2}{2D}\right]
\end{aligned}
\tag{61}
$$

For constant drive ($m = 0$, $I_{\mathrm{E}}(t) \equiv \hat{I}_{\mathrm{E}}^i$) we recover Eq (37), and thus the asymptotic solution $\mu_{\max}^\infty$. For $m \neq 0$, we can solve Eq (61) for small $m$. To this end, we insert the perturbation series $\mu_{\max}^i = \mu_{\max}^\infty(\hat{I}_{\mathrm{E}}^i) + \hat{\mu}m + O(m^2)$ and only keep the terms linear in $m$. Using Eq (37), we obtain to first order in $m$

$$
\mu_{\max}^i \approx \mu_{\max}^\infty(\hat{I}_{\mathrm{E}}^i) + m\hat{\mu}
$$

with

$$
\hat{\mu} \quad \approx \frac{\tau_m(1 - e^{-\Delta/\tau_m})}{[I_{\mathrm{E}} - V_T]\sqrt{\frac{2}{D}\ln\left[\frac{K}{\sqrt{2\pi D}}e^{\Delta/\tau_m}\right]}} + 2\ln\left[\frac{K}{\sqrt{2\pi D}}e^{\Delta/\tau_m}\right] - \underbrace{[\tau_m - (\Delta + \tau_m)e^{-\Delta/\tau_m}]}_{>0}
\tag{62}
$$

Since the drive is strongly superthreshold ($\hat{I}_{\mathrm{E}}^i \gg V_T$) the first order deviation $m\hat{\mu}$ of $\mu_{\max}^i$ from its asymptotic value $\mu_{\max}^\infty(\hat{I}_{\mathrm{E}}^i)$ is generally very small. For biologically plausible parameters (*e.g.* a synaptic delay $\Delta$ that is not too small), the second term of Eq (62) is slightly larger than the first. Thus, $\mathrm{sgn}(m\hat{\mu}) = -\mathrm{sgn}(m)$, *i.e.* $\mu_{\max}^i$ is slightly smaller than its asymptotic value for linearly *in*creasing drive, and slightly larger otherwise (see Fig 6C, cyan dots vs line).

Since already the zeroth-order approximation $\mu_{\max}^\infty(\hat{I}_{\mathrm{E}}^i)$ is close to the numerical solution $\mu_{\max}^i$ of Eq (61), and the reset mechanism remains unchanged, this implies that also $\mu_{\mathrm{reset}}^i$ is close to $\mu_{\mathrm{reset}}^\infty(\hat{I}_{\mathrm{E}}^i)$.

The duration of the upstroke $t_{\mathrm{off}}^i$ can be obtained from Eq (59) taking into account that $I_{\mathrm{E}}(0) = \hat{I}_{\mathrm{E}}^i - mt_{\mathrm{off}}^i$:

$$
t_{\mathrm{off}}^i = -\tau_m \mathrm{W}\left(\frac{\hat{I}_{\mathrm{E}}^i - m\tau_m - \mu_{\max}^i}{m\tau_m}\exp\left(-1 + \frac{\hat{I}_{\mathrm{E}}^i - \mu_{\min}^i}{m\tau_m}\right)\right) + \frac{\hat{I}_{\mathrm{E}}^i - m\tau_m - \mu_{\min}^i}{m}
\tag{63}
$$

where W is the Lambert W function, which has solutions for arguments $> -\exp(-1)$. For positive slope $m > 0$ and low drive ($\hat{I}_{\mathrm{E}}^i - \mu_{\max}^i < m\tau_m$) this introduces a constraint on the initial value:

$$
\mu_{\min}^i > \hat{I}_{\mathrm{E}}^i - m\tau_m \log\left(-\frac{m\tau_m}{\hat{I}_{\mathrm{E}}^i - m\tau_m - \mu_{\max}^i}\right)
\tag{64}
$$

(cf. Figs 6B and 7Ai).

The instantaneous frequency of the cycle follows as

$$
f_{\mathrm{net}}^{\mathrm{inst},i} = \left(t_{\mathrm{off}}^i + \Delta\right)^{-1} .
\tag{65}
$$

Note that for linearly changing drive, the asymptotic initial value $\mu_{\min}^{\infty}(\hat{I}_E^i)$ no longer implies the asymptotic frequency $f_{net}^{inst,i} = f_{net}^{\infty}(\hat{I}_E^i)$. If the drive increases linearly with a positive slope $m > 0$, the correct initial value $\mu_{\min}^i$ leading to $f_{net}^{\infty}(\hat{I}_E^i)$ (Fig 6B, left, white line) is larger than $\mu_{\min}^{\infty}(\hat{I}_E^i)$ (Fig 6B, left, black line). Conversely, if the drive decreases linearly ($m < 0$), the correct initial value $\mu_{\min}^i$ leading to $f_{net}^{\infty}(\hat{I}_E^i)$ is smaller than $\mu_{\min}^{\infty}(\hat{I}_E^i)$ (Fig 6B, right, white vs black line). The correction of the initial condition can be understood as follows: if *e.g.* the slope is positive, $m > 0$, the driving current $I_E(t)$ is smaller than $\hat{I}_E^i$ during the upstroke of $\mu(t)$, which lasts until time $t_{off}^i$ (cf. Eq (12)) and Fig 6A, left). A smaller driving current results in a slower increase of $\mu(t)$, (cf. Eq (8a)), and hence a longer rise time from $\mu_{\min}^i$ to $\mu_{\max}^i$. In order to match the asymptotic frequency, the initial value must therefore be chosen larger than $\mu_{\min}^{\infty}(\hat{I}_E^i)$ so as to compensate the slower increase of $\mu(t)$ during the rising phase. A similar argument holds when the driving current is decreasing ($m < 0$).

**Step 2.** We will now compute the mean membrane potential $\mu_{\min}^{i+1}$ that is reached at the end of a cycle $i$. Note that this step is technically not necessary to understand the instantaneous frequency $f_{net}^{inst,i}$ of an isolated cycle $i$, which we already derived above as a function of the reference drive $\hat{I}_E^i$, the slope $m$, and an arbitrary initial condition $\mu_{\min}^i$ (Eq (65)). It is however of interest to demonstrate that the mean membrane potential $\mu_{\min}^{i+1}$ at the end of the cycle is close to the asymptotic reference $\mu_{\min}^{\infty}(\hat{I}_E^i)$, since it serves as the initial condition of the next cycle. It is this property, together with the dependence of $f_{net}^{inst,i}$ on the initial condition $\mu_{\min}^i$, that implies the emergence of IFA under a piecewise linear drive that first increases, and then decreases over multiple consecutive ripple cycles (cf. Results, Fig 6C).

We find $\mu_{\min}^{i+1}$ by integrating the total current during the downstroke:

$$\mu_{\min}^{i+1} = \mu_{reset}^i e^{-\Delta/\tau_m} + \frac{1}{\tau_m} \int_0^{\Delta} I(s + t_{off}^i) e^{-(\Delta-s)/\tau_m} ds \qquad (66)$$

The total current $I$ can be split into two parts: $I^{stat}$, which is approximately equal to the feedback current for constant drive $I_E \equiv \hat{I}_E^i$ (Eq (41), except for slight deviations in $\mu_{\max}^i$), and an additive new term $I^m$ caused by the linear change in the external drive:

$$
\begin{aligned}
I(t + t_{off}^i) &= I^{stat}(t + t_{off}^i) + I^m(t + t_{off}^i) \\
I^{stat}(t + t_{off}^i) &= \hat{I}_E^i - K p(V_T, t_{off}^i - (\Delta - t))\left(\hat{I}_E^i - \mu_{\max}^i\right) e^{(\Delta-t)/\tau_m} \\
&\quad + K^2 p(V_T, t_{off}^i - (\Delta - t)) p(V_T, t_{off}^i - (2\Delta - t))\left(\hat{I}_E^i - \mu_{\max}^i\right) e^{(2\Delta-t)/\tau_m} \\
I^m(t + t_{off}^i) &= m\Big(t + K p(V_T, t_{off}^i - (\Delta - t))(\Delta - t) \\
&\quad - K^2 p(V_T, t_{off}^i - (\Delta - t)) p(V_T, t_{off}^i - (2\Delta - t))(2\Delta - t)\Big)
\end{aligned}
$$

For constant drive ($m = 0$) we recover the asymptotic solution $\mu_{\min}^{\infty}(\hat{I}_E^i)$ (Eq (50)). For $m \neq 0$ we integrate Eq (66) numerically and find that $\mu_{\min}^{i+1}$ is indeed close to the asymptotic solution $\mu_{\min}^{\infty}(\hat{I}_E^i)$. One can infer intuitively, that the sign of the (small) deviation of $\mu_{\min}^{i+1}$ from $\mu_{\min}^{\infty}(\hat{I}_E^i)$ equals the sign of the slope $m$: If the drive *increases* linearly, the drive during the upstroke of the mean membrane potential ($0 \leq t \leq t_{off}^i$) is *below* the reference drive $\hat{I}_E^i = I_E(t_{off}^i)$ (Eq (12)). Thus $\mu$ rises towards a slightly smaller maximum $\mu_{\max}^i$ (Eq (62)) and with slightly reduced speed, which decreases the resulting population rate and thus the inhibitory feedback (Eqs (29) and (23b)). Furthermore, the excitatory drive during the feedback-induced downstroke of $\mu$ ($t \in [t_{off}^i, t_{off}^i + \Delta]$)) is stronger than the reference drive (Eq (12)). It follows that the

downstroke of the mean membrane potential is reduced, and thus $\mu_{\min}^{i+1} \gtrsim \mu_{\min}^{\infty}(\hat{I}_E^i)$. The opposite argument can be made for linearly *decreasing* drive.

At this point it is clear that a piecewise linear drive, that first increases and then decreases over multiple ripple cycles (Eq (17)) will inevitably induce IFA. It is also clear, that this IFA asymmetry will vanish in the limit of infinitely slow drive ($|m| \to 0$).

Showing a concrete example of IFA under linear drive (Fig 6C) requires a forward integration of the network response over multiple cycles. Note that our analytical ansatz centered around the reference point $\hat{I}_E = I_E(t_{\text{off}}^i)$ (Eq (56)) focuses on isolated cycles. For any *individual* cycle *i*—characterized by the drive $\hat{I}_E^i$, the slope *m* and the initial condition $\mu_{\min}^i$—we can (semi-)analytically calculate the frequency difference $\Delta f_i = f_{\text{net}}^{\text{inst},i} - f_{\text{net}}^{\infty}(\hat{I}_E^i)$ (Eqs (65) and (32)). The colorplots in Fig 6B show $\Delta f_i$ at a fine resolution on two hyperplanes in this three-dimensional parameter space (drive with slope $m = \pm 0.4$/ms). In retrospect, the drive at the beginning and end of any individual cycle can easily be inferred as $I_E(t_0^i) = \hat{I}_E^i - mt_{\text{off}}^i$ and $I_E(T^i) = \hat{I}_E^i + m\Delta$. Thus, to approximate the ripple dynamics over multiple consecutive cycles under linear drive (Fig 6B and 6C, circular markers) we search the previously explored parameter space and match cycles self-consistently such that the drive at the end of one cycle equals the drive at the beginning of the next: $\hat{I}_E^i + m\Delta = \hat{I}_E^{i+1} - mt_{\text{off}}^{i+1}$. For the double ramp examples shown in Figs 6 and 7 we further restricted the search for trajectories during the upwards-ramp to trajectories where the drive in the last cycle ends at the plateau level $I_E^{\text{full}}$.

**Comparing theory and simulation for piecewise linear drive.** Since the theory provides a discrete, cycle-wise estimate of the instantaneous network frequency, we compare the result to the discrete estimate of instantaneous frequency in the simulations, based on the inverse of the distances between consecutive peaks in the oscillatory population rate. In the spiking network, SPW-like drive is modeled as a piecewise linear double ramp with an intermediate plateau phase of 20 ms (Eq (17)). The Gaussian-drift approximation is used to estimate the instantaneous network frequencies separately for the rising and falling phases of the drive (linear increase or decrease with slope $\pm m$). The plateau phase is ignored, since the network frequencies rapidly converge to the asymptotic frequency associated to the drive during the plateau phase. In both simulation and theory IFA is quantified by computing a linear regression slope over the instantaneous frequencies. The theoretically estimated instantaneous frequencies are shifted in time to account for a hypothetical plateau phase of 20 ms in between up- and downstroke and allow full comparability with the simulation results.

For every theoretical instantaneous frequency estimate $(t_i, f_{\text{theory}}^{\text{inst}}(t_i))$ an error is calculated relative to the average instantaneous frequencies observed in the spiking simulation around the same time point ($t_i \pm 1.5$ ms):

$$f_{\text{sim}}^{\text{inst}}(t_i) \approx \left\langle \{f_{\text{sim}}^{\text{inst}}(t)\}_{t \in [t_i - 1.5, t_i + 1.5]} \right\rangle$$

The average relative error of the theoretical estimate, compared to the simulations, is then computed as

$$\epsilon := \frac{1}{n} \sum_i \frac{\left| f_{\text{theory}}^{\text{inst}}(t_i) - f_{\text{sim}}^{\text{inst}}(t_i) \right|}{f_{\text{sim}}^{\text{inst}}(t_i)} \quad \in [0, 1] \ , \tag{67}$$

see Table 1.

## Stationary solution and linear stability analysis of the Fokker-Planck Equation

In what follows, we briefly summarize how we approximated the spiking network's bifurcation point in the mean-field limit as well as the network frequency and the mean unit firing rate in the bifurcation (red triangle and circular marker in Fig 1B). This involves finding the stationary solution of the Fokker-Planck Equation (Eq (23)) and analyzing its linear stability with respect to the external drive. These are standard procedures described in the literature [34, 51, 53, 90]. Here, for an easier reference, we reproduced these findings using the mathematical symbols used throughout the manuscript.

**Stationary solution.** The stationary solution $p_0(V)$ of the FPE has been derived by [34] (see also [51]). The constant population rate $r_0$ and resulting total drive $I_0$ in the stationary state can be inferred self-consistently by solving:

$$I_0 = I_E - K\tau_m r_0$$
$$r_0 = f_{LIF}(I_0)$$

where

$$f_{LIF}(I) = \left( \tau_m \sqrt{\pi} \int_{(I-V_T)/\sqrt{2D}}^{(I-V_R)/\sqrt{2D}} e^{x^2} \mathrm{erfc}(x) \mathrm{d}x + \tau_{ref} \right)^{-1}$$

is the firing rate of an uncoupled LIF neuron receiving constant drive $I$ and Gaussian white noise of intensity $D$ (f-I curve, [91]).

**Linear stability analysis.** For a given external drive $I_E$ one assumes a weak, periodic perturbation of the population rate around its stationary value:

$$r(t) = r_0(I_E) + \epsilon r_1(t) = r_0(I_E) + \epsilon e^{i\omega t + \lambda t} \tag{68}$$

The bifurcation where the stationary state loses stability corresponds to $\lambda = 0$. In the recurrently coupled network this perturbation of the rate translates into a perturbation of the input current:

$$I(t) = \underbrace{I_E - K\tau_m r_0(I_E)}_{=:I_0} - \underbrace{\epsilon K\tau_m r_1(t-\Delta)}_{=:-I_1(t)} \tag{69}$$

The linear response of the LIF units to this weakly modulated drive $I(t)$ under Gaussian white noise is given by convolution with the linear response function $G$:

$$r(t) = r_0(I_E) + \epsilon \int_0^\infty G(s, I_E) I_1(t-s) s$$

This output rate must match the weakly periodically modulated rate $r(t)$ that we assumed in the beginning (Eq (68)):

$$r_1(t) = \int_0^\infty G(s, I_E) I_1(t-s)\mathrm{d}s \overset{(69)}{=} -K\tau_m \int_0^\infty G(s, I_E) r_1(t-\Delta-s)\mathrm{d}s$$

At the bifurcation ($r_1(t) = e^{i\omega t}$, Eq (68)) this self-consistent condition is equivalent to

$$1 = -K\tau_m \tilde{G}(\omega, I_E) e^{-i\omega\Delta} \quad \Leftrightarrow \quad \begin{cases} 1 = K\tau_m |\tilde{G}|, & \text{amplitude condition} \\ 0 = \pi + \arg(\tilde{G}) - \omega\Delta, & \text{phase condition} \end{cases} \tag{70}$$

where $\tilde{G}$ denotes the Fourier transform of the linear response function (*susceptibility*). We use

the exact expression for the susceptibility of an LIF unit under Gaussian white noise (more specifically, the complex-conjugated of the expression derived by [53]):

$$\tilde{G}(\omega) = \frac{r_0}{\sqrt{D}} \frac{i\omega}{i\omega + 1} \frac{\mathcal{D}_{-i\omega-1}\left(\frac{I_0-V_T}{\sqrt{D}}\right) - e^{\delta}\mathcal{D}_{-i\omega-1}\left(\frac{I_0-V_R}{\sqrt{D}}\right)}{\mathcal{D}_{-i\omega}\left(\frac{I_0-V_T}{\sqrt{D}}\right) - e^{\delta}e^{-i\omega\tau_{\text{ref}}}\mathcal{D}_{-i\omega}\left(\frac{I_0-V_R}{\sqrt{D}}\right)} \ .$$

where $\mathcal{D}$ are parabolic cylinder functions and $\delta = \frac{V_R^2 - V_T^2 + 2I_0(V_T-V_R)}{4D}$ (for an alternative expression in terms of confluent hypergeometric functions, see [92]). We solve the amplitude and phase condition (Eq (70)) numerically to find the critical drive $I_{\text{E}} =: I_{\text{E}}^{\text{crit}}$ (at which the stationary state loses stability) and the corresponding frequency $\omega$ of the emerging oscillation. The network frequency and mean unit firing rate at the bifurcation are thus given by $f_{\text{net}}^{\text{crit}} = \omega/2\pi$ and $f_{\text{unit}}^{\text{crit}} = r_0(I_{\text{E}}^{\text{crit}})$. This approach is equivalent to the derivation by [34] via linear expansion of the FPE solution (see also [71]).

## Supporting information

**S1 Appendix. Performance evaluation of the Gaussian-drift approximation for constant drive.** The Gaussian-drift approximation under constant drive is compared to spiking network simulations for a range of noise intensities $D$ and inhibitory coupling strengths $K$ (Fig A). The parameter dependencies of the point of full synchrony are captured well by the theory (Fig B).
(PDF)

**S2 Appendix. The influence of network architecture and the shape of the external drive on the asymptotic and instantaneous ripple oscillation dynamics.** A covariation of network architecture and stimulus profile demonstrates that IFA is modulated by, but occurs largely independent of, the shape of the asymptotic network frequency as a function of the external drive (Fig A). Furthermore, we illustrate that a simple square pulse cannot elicit IFA in the feedback-based inhibition-first model.
(PDF)

**S1 Fig. Transient bimodality in the membrane potential distribution can affect instantaneous ripple frequency dynamics in the spiking network.** (A) Same layout as in Fig 2A: Spiking network response to an isolated *downwards* ramp stimulus with the same slope as in Fig 2D, middle, after time $t > 10$ ms ($N = 10,000$). Note that units that participate in the third population spike tend *not* to spike in the fourth population spike (red lines in raster plot), which is an indication of a residual bimodality in the membrane potential distribution from one cycle to the next (only faintly visible in voltage plot, see red square). (B) Same layout as Fig 2B: instantaneous network frequencies pooled together from 50 such ramp-down-only simulations. What appears as a continuous non-monotonic "wiggle" in the instantaneous frequencies of Fig 2D, middle (gray dots) is now clearly identifiable as a single outlier cycle (marked in red).
(TIF)

## Acknowledgments

We thank José R. Donoso, Nikolaus Maier, Naomi Auer, and Gaspar Cano for valuable discussions and comments on the manuscript.

## Author Contributions

**Conceptualization:** Natalie Schieferstein, Tilo Schwalger, Benjamin Lindner, Richard Kempter.

**Formal analysis:** Natalie Schieferstein, Tilo Schwalger, Benjamin Lindner, Richard Kempter.

**Funding acquisition:** Benjamin Lindner, Richard Kempter.

**Investigation:** Natalie Schieferstein.

**Methodology:** Natalie Schieferstein, Tilo Schwalger, Benjamin Lindner, Richard Kempter.

**Project administration:** Richard Kempter.

**Software:** Natalie Schieferstein.

**Supervision:** Tilo Schwalger, Benjamin Lindner, Richard Kempter.

**Visualization:** Natalie Schieferstein.

**Writing – original draft:** Natalie Schieferstein.

**Writing – review & editing:** Natalie Schieferstein, Tilo Schwalger, Benjamin Lindner, Richard Kempter.

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
