## [Decision Letter · Decision Letter 0]

18 Jun 2023

Dear Ms Schieferstein,

Thank you very much for submitting your manuscript "Intra-ripple frequency accommodation in an inhibitory network model for hippocampal ripple oscillations" for consideration at PLOS Computational Biology.

As with all papers reviewed by the journal, your manuscript was reviewed by members of the editorial board and by several independent reviewers. In light of the reviews (below this email), we would like to invite the resubmission of a significantly-revised version that takes into account the reviewers' comments.

We cannot make any decision about publication until we have seen the revised manuscript and your response to the reviewers' comments. Your revised manuscript is also likely to be sent to reviewers for further evaluation.

Sincerely,

Jonathan David Touboul

Academic Editor

PLOS Computational Biology

Lyle Graham

Section Editor

PLOS Computational Biology

Reviewer's Responses to Questions

**Comments to the Authors:**

Reviewer #1: In this paper, Schieferstein et al., look at the theoretical mechanisms of intra-frequency accommodation (IFA) of hippocampal ripple oscillations (>150 Hz) in a fully connected network of N inhibitory cells. Using the integrate-and-fire passive model, they develop an analytical mean-field approach for the large N condition to derive a drift-based approximation. This approximation is used to explain how IFA continuously emerges in response to symmetrical fast input ramps accompanying sharp-waves. They elegantly show that under these conditions, a hysteresis emerges in the trajectory of membrane potential from the minimal level to the spike threshold that controls cycle duration in response to fast symmetric changing drive.

The paper is well written, and in spite of the strong theoretical background, it is easy to read. Results are clearly presented and contribute to a better understanding of the behavior of inhibition-first models of ripple oscillations developed by the senior author’s previous work. There is value in adopting an analytical solution of the model operating in a regular oscillatory regime. Apart from some specific comments, my only concern is with providing a more balanced view of the implications of results when it comes to the real biological phenomenon.

Below comments and suggestions that I hope can be useful to improve the paper.

1- The motivation of using symmetrical inputs is justified, since the authors look to explain an asymmetric phenomenon, IFA, from the intrinsic properties of the bifurcation-based model. However, IFA emerges directly along the fast rising phase of an input after the initial transitory response, not apparently requiring the descending phase (fig.5 and 6, eq 50 for toff, and 35 for fnet). As long as the input is fast enough to drive the instantaneous cycle above the expected asymptotic frequency, IFA will emerge. While the non-trivial hysteresis effect to a symmetric input emphasizes IFA below fnet, in real case scenarios sharp-waves are not necessarily symmetric. Thus, the emphasis on the hysteresis drags the focus out to the real speed-dependent mechanism which may actually operate. The paper would benefit by better considering these aspects (the authors partially touch this point only at the level of discussion, lines 490-495).

2- At introduction, there are references to various computational models of ripple generation, basically, inhibition-first and excitation-first models. The authors then refer to the experimental evidence and debate, and propose that looking at transient properties such as IFA can advance model selection. Arguing that only inhibition-first models reproduces IFA, it is implicitly assumed that they are best posed to explain ripple generation. However, all this requires some corrections. First, other examples of excitation-first models, including excitatory-inhibitory networks, are proposed to explain a range of frequency behavior during ripples (e.g. PMID: 21452258, PMID: 30482688, PMID: 2112357, PMID: 21452258). By bringing the focus to inhibition-first models (based only on recurrently connected inhibitory cells) the introduction reads unbalanced. Actually, some of the cited models consider a network of excitatory and inhibitory cells to generate transient ripple oscillations (e.g., ref 32). This is important, since ripples generation requires excitatory-inhibitory interactions as experiments suggest: optogenetic activation of PV cells does not elicit ripples, while their activation by pyramidal cells does (Stark et al., 2014). Second, some of the cited references are not about computational models of inhibition-first ripple oscillations; rather they are either about up-down slow transitions, stationary gamma oscillations and/or they reflect experimental data. Finally, some of these problems drag into results and discussion (for example: line 46, line 512).

3- Fig.4 and text: it could be useful to specify that simulation data refers to the N=10000 model, where the lim to inf applies. The limit of applicability of the mean-field solution seems to draw two different scenarios above and below simulations.

4- Line 440: The experimental results of optogenetic stimulation of PV cells generating ripples is shown at CA3 in vitro, which represents a reduced experimental model (the biological complexities explaining why that experiment worked are far from the focus of the paper). The basic pyr-inh microcircuit motif was demonstrated experimentally by Bazelot & Miles in CA3 (PMID: 26728572) and confirmed in vivo by Stark in CA1 (ref 40). Importantly here, the statement “The speed-dependence of IFA is an important prediction of the bifurcation based model….” may be confusing. The whole paper is about showing how the mean-field solution for the IFA can be applied far from the bifurcation regime (IE >> IE min; fig.4) and so this makes the intra-ripple accommodation phenomena not necessarily linked to the bifurcation point (which would be rather obvious) nor to inhibition-first model.

5- Line 487: The statement that decreases/increases of ripple frequency are observed experimentally refers to non-physiological data, such as in ref 41 in response to treatment with GABAa receptor agonists and antagonists. This should be clarified to avoid the reader making a wrong impression.

6- Discussion will benefit from inserting a short paragraph on the limitations of the study/model. Apart from the applicability range of the mean-field approximation, other aspects to be considered are: a) Real LFP signals are contributed by spatially aligned currents from pyramidal cells. This means that inhibitory potential dynamics are strongly influenced by the integration properties in post-synaptic pyramidal cells (in contrast to statement in line 525 regarding frequency structure of IPSPs), which are not considered by the model. b) only passive leaky currents are considered in eq 1, but GABAergic cells exhibit different degree of firing rate accommodation. c) short-term plasticity of inhibitory potentials are not included. Both facilitation and depression are characteristic of heterogeneous populations of basket cells. Finally, the mean-field approximation applies to any other regular oscillatory regime with different fnet (such as for gamma and ref 34 &35) and so results may not uniquely applied to the ripple phenomenon.

7- For the pathological fast ripple regime, other models have been proposed based on excitation-first and/or excitatory-inhibitory interactions. The paper misses the opportunity to comment this further (e.g. lines 523, 727), although not strictly necessary. Importantly, note that experimental evidence demonstrates that physiological ripples always span the 100-250Hz frequency range and anything >250 Hz is considered pathological. Given that ripple frequency in inhibitory-first models is determined by interneuron high-frequency firing rates, the mean ripple frequency in these models is not informative. Rather, it is the dynamics of the intra-ripple adaptation what really matter. This comment is just a side note for the authors to consider if appropriate.

Suggestions on the paper structure and readability

8- Many equations are numbered indisctinly in the results and method sections. For example, eq. 1 and 13 are the same, eq. 2 and 14, etc.. (though some terms are written differently). This is possibly inevitable but the reader can get lost easily and it requires attention moving all along the results and method section.

9- The method section is rather long, especially for the mean-field approximation sections. One the one hand this is a major part of the paper, on the other it reads very dense and to get the main point requires digging into the underlying math. A suggestion would be to provide a short summary (also considering the previous point) which integrates the figures, and moving a large part of the mathematical derivation to supplementary material (or annex).

Reviewer #2: Reviewer’s report for PLoS CB on

“Intra-ripple frequency accommodation in an inhibitory network model for hippocampal

ripple oscillations”

by Natalie Schieferstein, Tilo Schwalger, Benjamin Lindner, Richard Kempter

Also attached as a pdf.

Overview.

This paper reports on the modeling and analysis of IFA. It provides a follow-up to an earlier paper from the Kempter lab: Donoso, Schmitz, Maier, Kempter (2018) J Neurosci

The spiking model involves all-to-all coupled LIF units; coupling is by an inhibitory current pulse with propagation delay of 1.2 ms. The application is SWRs in CA1 driven by excitatory drive IE from CA3; IE is represented by a linear ramp-up then plateau then linear ramp-down with total duration of 10s ms. The goal is to provide and to analyze a MF-like approximation for the experimental and simulated observations: generally, the intra-SWR cycle frequency shows ‘accommodation’, decreasing throughout the stimulus but with more or less increasing frequency during the late down-ramp, depending on the speed of the down-ramp.

The treatment proceeds in a well-organized hierarchy to describe the response to increasingly detailed features of the transient input. The authors begin with strictly steady input and approximating the steady response in terms of network frequency and cell frequency vs IE (Fig 4). This analytic part alone is quite nice and of value. Then, they provide math analysis of the ramping up and ramping down in stepwise fashion, with an IE incremental step occurring at the end of each response cycle; finally they mathematically treat the linear ramp case.

The response during ramping deviates from close tracking of the asymptotic frequency, depending on the speed and direction of the changing IE. There is asymmetry (hysteresis) between increasing and decreasing IE. A substantial portion of the paper involves accounting for the deviations with simulation and math analysis; an attempt with various approximations to describe/dissect the response to transient stimuli.

The math analysis is insightful with clever approximations/assumptions, based on a Fokker-Planck description of the noisy response of the LIF network. The authors provide a valuable and novel analytic treatment for responses to a transient input. But some readers may prefer a more direct demonstration of the gaussian approximation without the detailed analysis (see our suggestion below, just before Major Concerns).

The paper is well-organized and clearly written, with generally understandable figures and captions. It’s long -- with Methods being longer than the main text (20 pages compared to 19 pages) and 4 pages of Supp Info.

Curiously, the authors’ I-only model shows decreasing asymptotic network frequency with increasing IE. This behavior contrasts with the network frequency vs IE relation seen in many cell-based models for an I-only network, including that in Donoso et al. (2018), Fig 1D, and, for example, in models for ING-type gamma oscillations the frequency typically increases with IE (Fig 12A in Wang & Buzsáki 1996; Fig 4C in Bartos, Vida & Jonas 2007).

We offer the following suggestion before listing our major/minor concerns.

The approximations, simplifying assumptions, and analysis are impressive for arriving at an analytical solution. A key factor is the gaussian profile assumption and by-passing of the cellular-threshold resetting. The DDE by itself is a valuable reduction and can be integrated numerically for demonstrating key features of IFA. Figures 3 and 4 are quite good for the case of steady input drive. Eventually, after going through much analysis in the paper, I realized that simulations of IFA for the direct numerical integration of the DDE (as in Fig 6) could be shown earlier, say, just after Fig 4. I suggest a comparison of the DDE simulations with the LIF network simulations to directly illustrate IFA (by simulation for a piecewise linear IE(t) ) with time courses like those in, say, Figs 2A, 3A, 3B. Many readers may be satisfied with these demonstrations and the text could offer them the option to by-pass the results and details of the analytical approach (pages 12-16 and in Methods, pages 27-33). You could include the time course of the mean of V across cells from the spiking network, likewise for Fig 1. The other figures could remain pretty much as they are.

Major Concerns:

1. In this LIF I-only network model the synaptic current Isyn is a delta-pulse, decreasing the post-synaptic V(t) by J/N=65 mV/10000. There is not a finite time course or decay time constant for Isyn. This representation leads to stronger post-synaptic hyperpolarization (decreased µmin) with increased IE. The hyperpolarization can be 2-3 x 13 mV below Vrest (see Fig 2A), say to -100 mV or more negative. I urge you to make this point very explicitly in the text. Please provide intuitive explanation for this decrease of µmin with increasing IE. Is it that, larger IE tends to speed the passage through threshold, enabling more spikes to occur in a firing episode (greater saturation, larger value of s) and thereby more inhibitory current pulses, decreased µmin? Is this the basis for decreased asymptotic frequency with IE? Please give the reader an intuitive explanation for the decrease of µmin with IE.

2. With conductance-based inhibitory synaptic current there could be saturation of the synaptic activation variable, w(t), according to Isyn = Gsyn w(t) (V-Vsyn) (with modestly hyperpolarizing Vsyn and 0 � w � 1) and recovery with time scale, τsyn. For GABAA inhibition this time scale can be 3-10 ms. Would a Gsyn-based Isyn allow for the dynamic effects and brief intra-SWR events that you find for pulse-coupling in the asymptotic frequency vs IE. How would the dynamics of IFA be altered? Reference to the earlier paper (Donovo et al., 2018) and synaptic filtering is barely adequate. I would like to see also a simulation of the authors’ LIF network with gsyn(t) = Gsyn w(t) for your typical ramping IE(t) with, say, two illustrative values for τsyn and Gsyn. It could be in Supp Info. It’s surprising to us why decreasing Gsyn (GABA blockers) in Donoso et al. (2018) did not affect the network frequency.

Our simulations of your model, but with conductance-based synaptic current, suggested that there is a longer transient after a firing episode before the distribution of membrane potential becomes Gaussian-like. With large noise/small external excitatory drive, bimodal or trimodal distribution can exist for more than 500 ms.

3. The idea of no more than one spike per neuron per cycle is clever to avoid the resetting. Can you provide references for exploiting this feature in other studies for network firing events? Is the methodology of FPE and approximations for ‘disregarding reset’ generalizable to some other network cases: E-only or to E-I networks, or other architectures (say, none all-to-all)?

4. I see that you concluded from the analytic formulae in line 794 (according to Eq 41) that K needs to be sufficiently large. That’s fine but I still wonder if further direct interpretation are gleaned from the formulae themselves, as they are quite complex.

For example, do you see directly from the analysis about limits for synaptic strength? Maybe you can point to why there is a lower bound for the synaptic strength instead of an upper bound? I will accept that for µmax to be large enough, the recurrent inhibition needs to be weak.

Minor Concerns:

5. I did not find a listing of individual contributions of the co-AUs to the project. I suppose you have provided this during the submission process.

6. I appreciate that you address, early on, for the naive reader, the lack of a precise definition of IFA (lines 115-124). Can you maybe say explicitly what features are not part of IFA? For example, some “nots” might be “network frequency decreases with IE in the asymptotic case” (True for your network but IFA may not depend on this), and another “not” is “network frequency changes slowly (decreases/increases) with time during steady input” (There is no explicit slow time scale or adaptation in your formulation.).

My impression is that IFA refers to the dynamic effect about deviations from steady (adiabatic) tracking of the asymptotic frequency for the instantaneous value of a dynamically changing IE. IFA decreases with slower dynamic changes in IE. IFA is not ‘adaptation of network frequency’ for steady input. “accommodation” in IFA is specifically a term applied to responses to dynamic (e.g., ramping) stimuli as used historically to describe behavior at the individual neuron level.

7. Is it feasible under steady or slowly changing inputs (e.g. optogenetics) to obtain experimental data to support the modeling result that CV decreases with increased IE? Is this mostly due to decreased standard deviation or increased period?

Also from experimental data, what is the variability of local period/frequency during a SPW? Is there correlation between durations of successive SPWs?

8. I suggest that for Fig 3 the caption should explicitly describe the shaded representation of the gaussian distribution as, say, “painted on” and state explicitly that the black curves in Fig 3B are time courses of the numerical integration of the DDE, according to the equation numbers (8a, b).

9. Can you briefly describe the phenomenological ‘resetting’ for the DDE early on, around equations (9-10), intuitively without the math details?

10. Include horizontal lines at Vthr, Vreset for all time course plots (e.g., Fig 1A, Fig 2A, Fig 3Ai, Bi).

11. Fig 6C bottom. I think the y-axis label should be for both V and for IE.

12. On page 3 (line 62) it’s mentioned that a refractory period is ignored for simplicity. What effects are we overlooking?

13. In the middle panel of Fig 2D there’s evidence of a small wiggle (gray dots) during decreasing phase of ramp. What is that about?

14. Re: the stability analysis that leads to Hopf bifurcation, HB, here and in Brunel & Hakim (1999). The authors state that the HB is supercritical at a few points in the text. It appears to be supercritical but does the analysis really distinguish that the HB is super- or sub-critical? My impression is that the determination of “direction” would require looking at higher order terms, beyond linear. Please confirm or restate.

15. Re: caption Fig 6. “Thin grey line: mean membrane potential trajectory μ(t) in response to SPW-like drive IE(t) (numerical integration of DDE Eq (8)).” Please state explicitly if this integration of DDE also includes the resetting.

16. The classification of inhibition-first models into categories that are named, perturbation-based and bifurcation-based, is not meaningful enough for a general reader. Since your operating range for the DDE is for strong coupling and strong drive it is far from the HB regime of weakly modulated spiking of Brunel-Hakim; why call it bifurcation-based? You are describing distinctly large-amplitude episodic behavior. Also, what is meant by perturbation-based for the other category?

Actually, I’m wondering why you go through the HB derivation if it already is in the literature?

I suppose, in your view, that an essential feature of a ‘bifurcation-based’ model is that the onset of oscillations occurs with weakly modulated (sparse) spiking. Increasing IE leads to increased saturation “s” and therefore more negative µmin which, in turn, overwhelms the increasing drive to produce slower network frequency. But for the Donoso et al. model (also bifurcation-based) the asymptotic frequency changes little while ‘s’ increases to 0.8 (almost its maximum) with increasing input drive and then both ‘s’ and frequency increase with IE. What is the more important effect in your model of increasing drive: that frequency decreases or that ‘s’ increases from near zero?

17. What is the initial condition for the numerical integration of the DDE, i.e. µ(t) for -Δ<t<0? dde="" infinite-dimensional="" is="" since="" the="">

18. Along the lines of the just-previous query, what is the initial condition for the LIF network? We found in our simulation that if the membrane potentials are initialized uniformly on [Vreset, Vthr], there will be a longer initial transient with multiple clusters in the distribution of membrane potentials. Will these clusters eventually merge into one Gaussian-like cluster or is it possible that for some parameter values, there are more than one cluster (e.g., two stable clusters) as time gets large? The raster may look like ABABAB where there are two different ISIs. This may also happen when the noise level D is large.

19. Re: line 496. “wrt” should be spelled out.

20. Can you mention/summarize, say in Discussion, the features of your analytic treatment that could generalize to other applications? And highlight the advantages of the analytic approximations, beyond having the DDE derivation, over, say, just integrating the DDE.</t<0

Reviewer #3: Overview

In the present MS, the authors suggest that the intra-ripple frequency adaptation observed in experiments is due to the inhibitory interneuron-based network model. In the inhibitory-based ripple model, the instantaneous frequency is inversely related to the magnitude of the instantaneous excitatory input. The basic idea of the present MS is that the instantaneous frequency depends not only on the instantaneous excitatory input, but also on the membrane potential at the end of the previous ripple cycle, endowing the system with hysteresis. Thus, when the input magnitude increases, the actual instantaneous frequency decreases more rapidly than expected, whereas when the input magnitude decreases, the instantaneous frequency increases more slowly than expected. When the input is a sufficiently short symmetric ramp, this hysteresis automatically translates to a largely monotonously decreasing instantaneous frequency.

General

I find that the MS is by and large clearly written and that the analytical and computational methods are rigorous and well explained. My main problem with the work is that the starting point for the development of the analytical model, the integrate and fire network model, yields asymptotic behavior that is inconsistent with previous theoretical and experimental findings. If the basis is inconsistent, then the entire work may be an interesting mathematical analysis but the relevance for the neurophysiology of ripples would remain unclear.

If the authors can modify the starting point for consistency with previous findings, then I think the MS warrants further consideration.

Major comments

(1) A main inconsistency is between the present integrate-and-fire model and previous spiking and firing rate models. In the present implementation, the steady-state instantaneous frequency decreases with increased input magnitude (Fig. 1B, top; Fig. 4), whereas in the noise-driven implementation, the correlation is clearly positive (see Fig. 6 of Brunel and Hakim, 1999) or by and large flat (Fig. 2 of Brunel and Wang, 2003). It is unclear whether the opposite dependencies of frequency on input magnitude are due to (1) structural differences in the model, (2) differences in parameters, (3) different working domains, or (4) the nature of the input. This inconsistency appears for both the analytical and the spiking models. Please explain the exact differences between the present integrate and firing simulation and the Brunel and Hakim simulation, and between the present firing rate model and the 1999 one.

(2) During spontaneous ripples, the mean ripple frequency exhibits a positive correlation with the SPW amplitude, which is consistent with the Brunel and Hakim model. Moreover, when the excitatory input is controlled experimentally using optogenetics, ripple frequency increases (see Fig. 1 of Stark et al., 2014). These inconsistencies (also see Comment #1 above) cast doubt on the relevance of the starting point of the MS (Figs. 1-4).

(3) In experiments, the frequency accommodation is clearly apparent also when the input is a constant optogenetic pulse – for long and short pulses, in CA1, DG, and neocortex. This critical feature of in vivo ripples appears to already test the predictions of the present MS but is neither accounted for nor discussed. The authors state that the “simplest form” of a SPW model is a symmetric piecewise linear ramp, but a pulse is certainly simpler from a theoretical point of view. Can the authors replicate the IFA phenomenon with a pulse?

(4) The authors arrive at the hysteresis insight via an elaborate large-signal approximation to the firing rate model of Brunel and Hakim (1999), but it is not clear to me that the effort is actually warranted. The basic logic is that the latency to the spike depends on the initial membrane potential, and this can already be seen in a single neuron spiking model. Can the authors demonstrate the hysteresis effect on the membrane potential of integrate and fire neurons?

Minor comments

(5) Please extend Fig. 1 to the regime where s > 1: give examples (in Fig. 1A) and extend the summary (Fig. 1B) to include the second bifurcation point.

(6) Please add raster plots to Fig. 1A for every regime (e.g., 100 random spike trains out of the 10,000).

(7) Please add spectra of the membrane potentials to Fig. 1A. Do the Vm spectra also exhibit harmonics at full synchronization?

(8) Please report the exact value of s at the first bifurcation, around 0.2 nA

(9) Please add estimates of the dispersion of the frequency estimation to Fig. 2A, i.e., add error bars to the white dots.

(10) Please explain why the asymptotic steady state frequency dips at the end of the first ramp and the beginning of the second ramp in Fig. 2B

(11) In Fig. 2B, the grey dots at the plateau of the input appear to decrease in frequency. Certainly the first cluster (around -10 ms) is above the line. The resolution is not perfect, but the second cluster (around -5 ms) appears to be mostly above the line, whereas the third cluster (around 5 ms) appears to be on the line. Thus, it appears that there is some accommodation even when the input is constant – can the authors comment on this? See also Comment #3 above. If what I wrote is correct, then it seems that a main conclusion of the MS, namely that a symmetric ramp is the “minimal input” that is required to replicate the IFA phenomenon in the inhibitory network model, should be reassessed.

(12) Please write all partial derivatives explicitly in all equations.

(13) What is the source of the “higher harmonics” in Fig. 1A? Are these harmonics due to the fitting of integer multiple sine waves to the sequence of “delta functions”, or is there rhythmicity at 400 Hz (as in Fig. 2A)?

(14) In Fig. 5A, the legend (and Fig. 5B) refers to seven examples, but only three are shown.

(15) Several of the equations are repeated in both the Results and Methods sections. Please refrain from the repetition, or state explicitly next to every repeated equation which other equation it replicates (in the Methods).

**Have the authors made all data and (if applicable) computational code underlying the findings in their manuscript fully available?**

Reviewer #1: **No: **They argue data will be available upon publication

Reviewer #2: **No: **The reviewer did not find a statement in the paper about uploading their simulation code(s) to a public depository.

Reviewer #3: **No: **The authors state that the DOI will be provided after acceptance

PLOS authors have the option to publish the peer review history of their article (what does this mean?). If published, this will include your full peer review and any attached files.

Reviewer #1: No

Reviewer #2: No

Reviewer #3: No
---

## [Decision Letter · Decision Letter 1]

31 Dec 2023

Dear Ms Schieferstein,

Thank you very much for submitting your manuscript "Intra-ripple frequency accommodation in an inhibitory network model for hippocampal ripple oscillations" for consideration at PLOS Computational Biology. As with all papers reviewed by the journal, your manuscript was reviewed by members of the editorial board and by several independent reviewers. The reviewers appreciated the attention to an important topic. Based on the reviews, we are likely to accept this manuscript for publication, providing that you modify the manuscript according to the review recommendations.

All Reviewers and Editors handling the manuscript are happy with the revision and we are ready to accept your paper. Reviewer #3 made further suggestions for you. After discussion between the Editors, we decided to give you an opportunity to consider these suggestions and include any changes relative to these two points (or leave your revision unchanged), so the manuscript management system considers this as a minor revision.

Sincerely,

Jonathan David Touboul

Academic Editor

PLOS Computational Biology

Lyle Graham

Section Editor

PLOS Computational Biology

All Reviewers and Editors handling the manuscript are happy with the revision and we are ready to accept your paper. Reviewer #3 made further suggestions for you. After discussion between the Editors, we decided to give you an opportunity to consider these suggestions and include any changes relative to these two points (or leave your revision unchanged), so the manuscript management system considers this as a minor revision.

Reviewer's Responses to Questions

**Comments to the Authors:**

Reviewer #1: The authors have addressed extensively my comments and I found the new version is improved. So, I endorse publication.

Reviewer #2: Your detailed responses to reviewers' concerns were appreciated, as were many of the modifications in your revised paper.

Reviewer #3: The paper has been revised and improved. Two key points remain open; both were raised in my previous assessment as well as by other reviewers in the previous round. The concerns did receive detailed replies in the response letter, but insufficient clarification in the paper itself. I believe that once clarified, the paper could make an interesting contribution to the literature.

First, the physiological observation that work aims to explain/replicate remains insufficiently clear. There are within-ripple changes in frequency, sure. But it is unclear whether it is an outcome of the system, the input to the system, etc. Furthermore, it is unclear whether there is only decrease (“adaptation”) or also an increase of frequency during the beginning of the ripples (see countless wideband traces in the literature, e.g., Fig. 1 of Buzsaki et al., 1992; Fig. 2A of Csicsvari et al., 1999). One way to address this experimentally is to take control over the input. While I agree that light power applied to pyramidal cells does not map 1:1 to the current input to the interneurons, the two are likely to be positively correlated, in particular for weak inputs and even for recurrent networks. Looking at the spectrograms in the experimental papers once again in detail, it seems clear that when the input was constant, there was prominent frequency adaptation of the LFP frequency (e.g., Fig. 2C, Fig. 4B, and Fig. S3C of Stark et al., 2014; Fig. S4A of Fernández-Ruiz et al., 2019). It remains unclear whether the present model replicates this observation?

Second, even after adding Supplementary Figure C, it is still unclear whether the model does or does not predict frequency accommodation for a pulse input to the interneurons. In Fig. 2B and in their reply to my previous Comment 11 the authors do report frequency accommodation during fixed input. However, when addressing the previous Comment 3 the opposite is stated, while minimizing the importance of the 1 cycle caveat. This does not receive consistent and direct treatment in the manuscript either. Because the issue is key, it should not be limited to discussions during the peer review process. Rather, the behavior with respect to a pulse input should be shown clearly in a main figure and discussed heads-on in the paper.

**Have the authors made all data and (if applicable) computational code underlying the findings in their manuscript fully available?**

Reviewer #1: Yes

Reviewer #2: Yes

Reviewer #3: Yes

PLOS authors have the option to publish the peer review history of their article (what does this mean?). If published, this will include your full peer review and any attached files.

Reviewer #1: No

Reviewer #2: No

Reviewer #3: No

Figure Files:

Data Requirements:

Reproducibility:

References:

---

## [Editor Report · Decision Letter 2]

1 Feb 2024

Dear Ms Schieferstein,

We are pleased to inform you that your manuscript 'Intra-ripple frequency accommodation in an inhibitory network model for hippocampal ripple oscillations' has been provisionally accepted for publication in PLOS Computational Biology.

Best regards,

Jonathan David Touboul

Academic Editor

PLOS Computational Biology

Lyle Graham

Section Editor

PLOS Computational Biology

---

## [Editor Report · Acceptance letter]

14 Feb 2024

PCOMPBIOL-D-23-00433R2 

Intra-ripple frequency accommodation in an inhibitory network model for hippocampal ripple oscillations

Dear Dr Schieferstein,

I am pleased to inform you that your manuscript has been formally accepted for publication in PLOS Computational Biology. Your manuscript is now with our production department and you will be notified of the publication date in due course.

With kind regards,

Anita Estes
